# Migration of solidification grain boundaries and prediction

Hongmei Liu[1,2,3] ✉, Shenglu Lu [◐][4], Yingbo Zhang [◐][1], Hui Chen[1], Yungui Chen[2] & Ma Qian [◐][4] ✉

Solidification processing is essential to the manufacture of various metal products, including additive manufacturing. Solidification grain boundaries (SGBs) result from the solidification of the last liquid film between two abutting grains of different orientations. They can migrate, but unlike normal GB migration, SGB migration (SGBM) decouples SGBs from solidification microsegregation, further affecting material properties. Here, we first show the salient features of SGBM in magnesium-tin alloys solidified with cooling rates of 8–1690 °C/s. A theoretical model is then developed for SGBM in dilute binary alloys, focusing on the effect of solute type and content, and applied to 10 alloy systems with remarkable agreement. SGMB does not depend on cooling rate or time but relates to grain size. It tends to occur athermally. The findings of this study extend perspectives on solidification grain structure formation and control for improved performance (e.g. hot or liquation cracking during reheating, intergranular corrosion or fracture).

Grain boundaries (GBs) are interfaces where crystals of different orientations meet[1]. They are an integral part of polycrystalline materials and exert a profound influence on almost all of their useful properties[2]. Solidification is an essential manufacturing process or step for various metal products, including metal additive manufacturing (AM). During solidification, an interface will form between two coalescing crystals of different orientations. The basic principles for the formation of such an interface or solidification grain boundary (SGB) in the last-stage solidification have been delineated by Rappaz[3,4] and other researchers[5–7]. It results from the solidification of the last liquid film (-1 nm thick) between two grains with solute composition $X_{L(f_s \to 1)}$[3–7], where $f_s$ denotes the solid fraction. In this last stage, the liquid film composition and thickness remain little changed until the required coalescence or bridging undercooling is reached for the liquid film to solidify as a SGB[3–7]. As illustrated in Fig. 1a[1], SGBs normally coincide with the peripheries of the coalescing crystals of different orientations. However, exceptions do occur, where SGBs can migrate to decouple themselves from their initial as-solidified boundaries, as shown schematically in Fig. 1b, referred to as SGB migration (SGBM).

Biloni[8] first pointed out the prospect of SGBM in 1961 to elucidate the grain structures observed in as-cast pure aluminium (Al, 99.99 wt.%). Chernyshova reported SGBM in both as-cast niobium-tantalum alloys and weld seams of niobium metal in 1967 according to ref. 9 (co-authored by Chernyshova). More SGBM phenomena were subsequently observed in as-cast, welded, wire-deposited (AM-fabricated) metals or alloys. Examples include Al-4.6 wt.% Mg[10], Al-0.2 wt.%Cu[11], Cu (Cu-0.006% wt.%O)[12,13], Cu-3 wt.%Sn[13,14], Ti-6 wt.% Cr[15], Zr-2 wt.%Mo[16], commercial magnesium alloy AM50 (Mg-4.9 wt.%Al-0.34 wt.%Mn)[17], maraging steels (both as-cast and wire-deposited)[9], stainless steels including both commercial (304, 309, 310, 347, 321, 316, 430)[18–25] and non-commercial grades[21,22,24,25], Ni-based filler alloys for welding[26], and high entropy alloys[27]. In general, SGBM is less often observed in commercial cast alloys that contain a noticeable presence of second-phase particles, due to the particle pinning effect (note that GB particles can migrate along with moving GBs, known as 'mobile particles'[28], especially at elevated temperatures, depending on their properties, sizes, and interactions with GBs).

[1]School of Materials Science and Engineering, Southwest Jiaotong University, 610031 Chengdu, China. [2]School of Materials Science and Engineering, Sichuan University, 610065 Chengdu, China. [3]Department of Chemical and Materials Engineering, The University of Auckland, Auckland 1010, New Zealand. [4]Centre for Additive Manufacturing, School of Engineering, RMIT University, Melbourne, VIC 3000, Australia. ✉e-mail: lhm@home.swjtu.edu.cn; ma.qian@rmit.edu.au

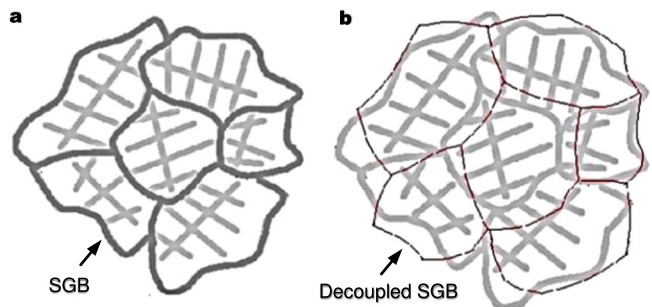

**Fig. 1 | Schematic correspondence between dendrite cells of different orientations and solidification grain boundaries (SGBs). a** Typical correspondence. **b** Decoupled SGBs due to SGBM.

Unlike GBM during recrystallisation or grain growth, SGBM decouples SGBs from solidification microsegregation of solutes and/or impurities. Consequently, it offers prospects for mitigating the risk or adverse effect of a variety of undesired developments: (i) GB-microsegregation-related hot or liquation cracking due to reheating during welding or AM[23,29]; (ii) thermal or residual stress at GB triple junctions[29], which are stressed regions conducive to fracture initiation[30]; (iii) intergranular corrosion due to SGB microsegregation[19,31], and (iv) intergranular fracture due to the microsegregation-induced second-phase formation (e.g. by slow cooling or reheating during welding or AM)[32]. In this regard, decoupling SGBs from solidification microsegregation is desired.

Information on SGBM scattered in the literature since 1961 provides the following basic understanding: (i) SGBM occurs near the solidus temperature ($T_{solidus}$) of the alloy[9,21,22,25,33]; (ii) the strain generated during cooling after solidification has a minor or negligible influence based on experiments using specially designed samples[20] – this clarifies an important concern about the underlying mechanism; (iii) tortuous or irregular SGBs are apt to migrate[20,34], but SGBM is not due to grain growth[34]; and (iv) the main driving force for SGBM is the reduction in the total GB energy[18,21,25,35], and the GB triple junctions approach equilibrium (straight GBs in 120°–120°–120°) after SGBM[18]. However, compared to the GBM in recrystallisation or grain growth, the fundamental factors affecting SGBM, including alloy composition (solute type and content), cooling rate, solidification characteristics and grain size, remain essentially unexplored. No theoretical model exists that allows for the prediction and control of SGBM during casting, welding or AM.

In this work, we first investigate SGBM in dilute Mg-Sn alloys and identify the influence of solute type and content, grain size, and cooling rate. Then we show the thermal stability of the migrated SGBs after annealing at an elevated temperature. On this basis, a theoretical model is developed for SGBM in dilute binary alloys, using the basic Gibbs equation to estimate the specific SGB energy and the Won-Thomas version of the Brody-Flemings model to describe the last-stage solidification. The model is validated with four dilute Mg-X (X = Sn, Pb, Al, Zn) alloys and six other alloys, by focusing on the effect of solute type and content. Finally, the kinetics of SGBM is briefly discussed.

## Results

### SGBM in as-cast Mg-0.63 at.%Sn alloy

The Mg-0.63 at.%Sn (3 wt.%Sn) alloy is a promising Mg alloy for biomedical and automotive applications[36,37]. Figure 2a, b shows its representative grain structures solidified in a preheated grey cast-iron mould. The boundaries of each growing cell are discernible by chemical etching due to solute microsegregation. After solidification, a SGB is expected to form at the boundary between every two adjacent grains of different orientations according to Fig. 1a. However, a significant number of the SGBs have migrated to other locations, as exemplified in Fig. 2a, akin to Fig. 1b.

Two types of SGBM are prevalent in this alloy: (i) large departures up to about 50 μm (arrows 1 and 2 in Fig. 2a), accompanied by the migration of SGB triple junctions, indicative of the extent of SGBM, and (ii) local small deviations (arrows 1 and 2 in Fig. 2b). In both cases, the migration decoupled SGBs from solidification microsegregation.

Figure 2c shows a forescatter detector (FSD) image of a selected area with migrated SGBs from the Mg-0.63 at.%Sn alloy. The network-like bright regions are due to atomic number contrast, confirmed to be Sn-enriched but Mg-depleted, Fig. 2e, f. They correspond to the dark dendrite cell regions in Fig. 2a and contain ~5–15 wt.%Sn (Supplementary Fig. 1), compatible with the solubility of Sn in α-Mg (15 wt.%) at the eutectic temperature. Figure 2d is an electron backscatter diffraction (EBSD) inverse pole figure (IPF) map of Fig. 2c. The misorientation angle/axis pairs are determined to be 32.5°/<52$\bar{7}$0> between the blue and purple grains, 32.9°/<54$\bar{1}$6> between the blue and green grains, and 35.5°/<2$\bar{7}$56> between the purple and green grains, consistent with SGBs being usually high angle GBs (>30°) in cast alloys[38]. The GB triple junction (130°–115°–115°) is close to the expected equilibrium state (120°–120°–120°)[39], indicative of its stable state after SGBM.

Also observed are small pockets of eutectic α-Mg(Sn)+Mg₂Sn phases in certain interdendritic regions, Fig. 2g, h, which are predictable using the subsequent Eq. (7). Note that a minimum of 3.35 at.% Sn is required for eutectic formation in binary Mg-Sn alloys by the phase diagram (equilibrium solidification). The eutectic presence increased with increasing Sn content over 0.63 at.% (Supplementary Fig. 2). It is, however, negligible in the Mg-0.21 at.%Sn and Mg-0.3 at.% Sn alloys, due to their low solute (Sn) content and the back-diffusion of Sn atoms from the remaining liquid into the solid α-Mg(Sn).

### The effect of solute content on SGBM in Mg-Sn alloys

We first investigate the effect of solute content on SGBM and then the effect of a solute type after the formulation of our model. To understand the influence of solute content, four more dilute binary Mg-$x$Sn alloys ($x$ = 0.21, 0.42, 1.07, 1.52, in at.%) were prepared and cast under the same conditions as for Mg-0.63 at.%Sn. Figure 3 depicts their representative grain structures, where the left-hand-side observations provide an overview of the grain structures while the right-hand-side images focus on selected SGBM features. Both solute microsegregation regions and SGBs are discernible in each field of view. SGBM occurred in each alloy but became less prevalent with increasing Sn content. For example, it was extensive in the Mg-0.21 at.%Sn (1 wt.%Sn) alloy, involving noticeable migration of the GB triple junctions, Fig. 3a, b. The trend is similar in the Mg-0.42 at.%Sn (2 wt.%Sn) alloy, Fig. 3c, d. At the solute content of 1.07 at.%Sn (5 wt.%), SGBM was still clear but had eased off, Fig. 3e, f. In contrast, SGBM was limited in the less dilute Mg-1.52 at.%Sn (7 wt.%Sn) alloy, Fig. 3g, h.

The formation of eutectic phases in a dilute alloy renders the possibility of particle pinning. Indeed, this was observed in the Mg-0.63 at.%Sn alloy (red arrows in Fig. 2g, Mg₂Sn particles). However, it is also clear from Fig. 2g that the two SGB segments that are about 30 μm away from the Mg₂Sn-pinning particles still exhibited distinct migration (yellow arrows in Fig. 2g). This implies that it may require a prominent presence of second-phase particles to completely suppress SGBM. This inference is supported by the observation in the Mg-1.52 at.%Sn (7 wt.%Sn) alloy, which had a marked presence of eutectic α-Mg+Mg₂Sn (Supplementary Fig. 2) and therefore exhibited only localised SGBM, Fig. 3g, h. As mentioned earlier, the particle pinning effect is related to particle properties, sizes, interactions with GBs, and temperature[28].

### SGBM at different cooling rates

SGBM at different cooling rates was investigated by changing the cooling rate from ~8 to ~1690 °C/s. Figure 4a shows an as-cast cone-shaped sample of the Mg-0.63 at.%Sn alloy. Its representative grain

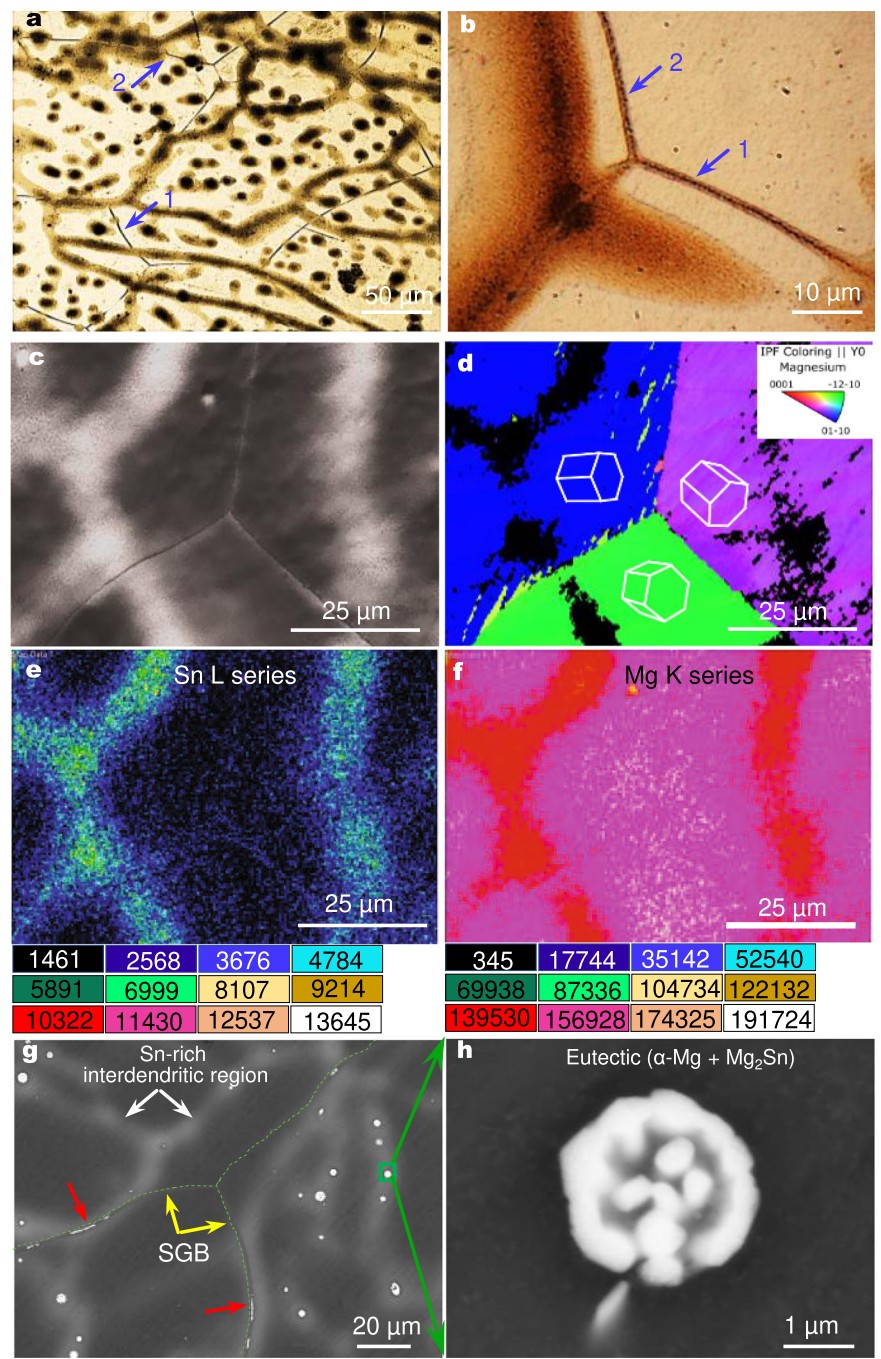

**Fig. 2 | SGBM in as-cast Mg-0.63 at.%Sn (3 wt.%) alloy. a, b** Optical micrographs showing large SGBM (arrows 1 and 2 in **a**) and localised small SGBM (arrows 1 and 2 in **b**). **c** A forescatter detector (FSD) image showing migrated SGBs and Sn-enriched interdendritic regions (bright). **d** EBSD inverse pole figure (IPF) map of **c**, showing different orientations of each grain at the triple junction in **c**, confirming the observed GB triple junction, where the dark regions are zero-solution regions, which cannot be indexed as Mg by the AZtec software due to Sn enrichment. **e, f** Energy dispersive spectroscopy (EDS) mapping of Sn and Mg in **c**, showing the enrichment of Sn (**e**) and depletion of Mg (**f**) in interdendritic regions. The unit for the colour scale is counts/s. **g** Backscattered electron (BSE) image showing migrated SGBs (yellow arrows), SGB particles (red arrows), Sn-enriched interdendritic regions (white arrows). **h** Eutectic α-Mg+Mg$_2$Sn from a Sn-enriched interdendritic region in **g**.

structures at sections **b**, **c** and **d** (Fig. 4a) are displayed in Fig. 4b–d. The cooling rate ($\dot{T}$) at each position was estimated from its secondary dendrite arm spacing (SDAS, $\lambda$) via $\lambda(\mu m) = A\dot{T}^{-n}$[40], where $A = 105.33$ and $n = 0.44$ for Mg-Sn alloys[41]. Table 1 lists the values of $\lambda$ and the calculated $\dot{T}$, together with the four Mg-Sn alloys studied earlier.

Section **b** in Fig. 4a underwent an average cooling rate of -54 °C/s and section **c** of -166 °C/s, while section **d**, close to the apex, was cooled at -1690 °C/s, which is -7.5 times the cooling rate of brine

quenching (220 °C/s[42]). As shown in Fig. 4e–g, SGBM occurred at each cooling rate. The maximum SGBM distance ($\Delta_{max}$) corresponding to each cooling rate was measured along with the grain size ($d$) and included in Table 1. The ratio of $\Delta_{max}/d$ varied between 30% and 40%, which is essentially independent of the range of cooling rate studied (8–1690 °C/s) or cooling time. This observation is in line with the influence of grain size on the migration of non-solidification GBs based on limited experimental data[43]. No change in grain number was observed after SGBM at each cooling rate, in agreement with the

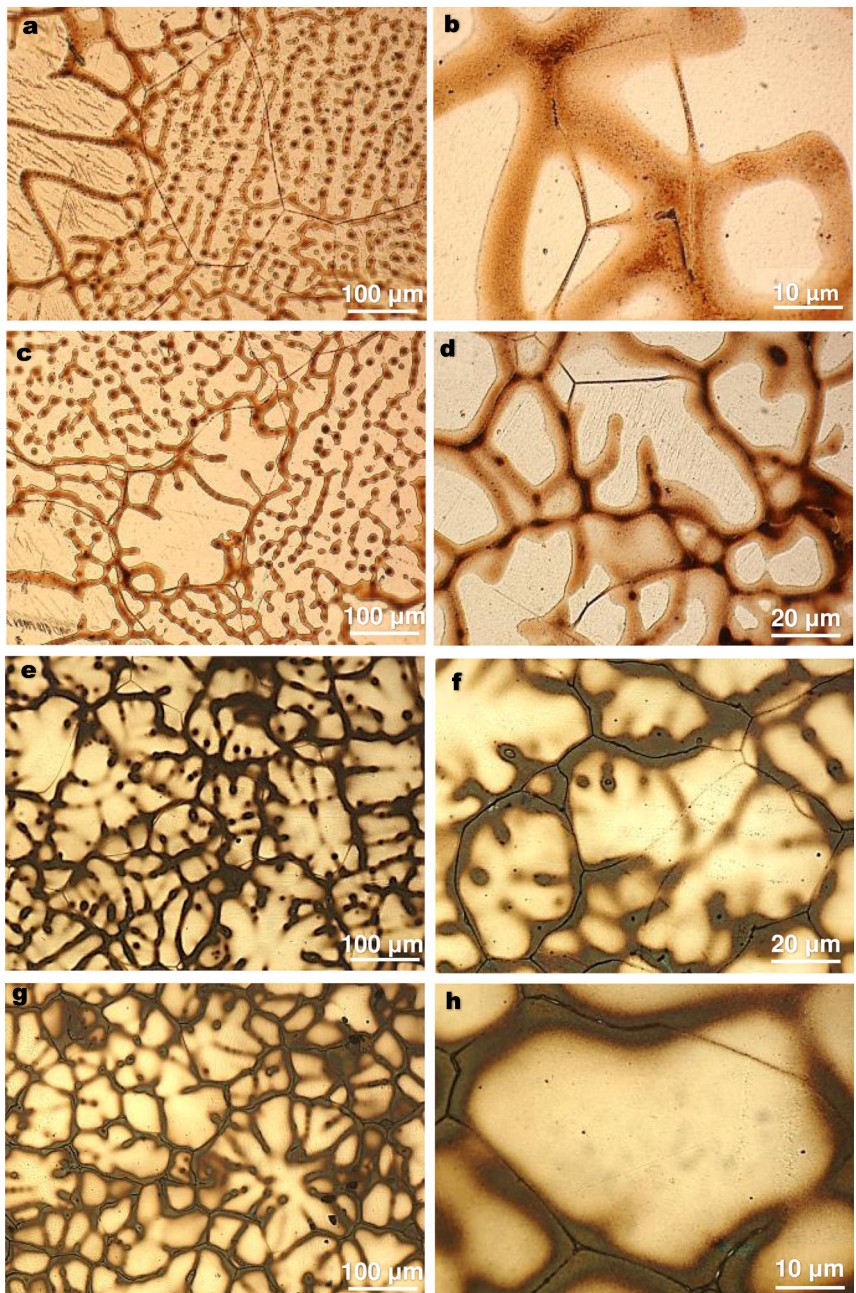

**Fig. 3 | SGBM in as-cast Mg-Sn alloys with different solute contents.** Left-hand-side: overviews; right-hand-side: closer views of SGBM features. **a, b** Mg-0.21 at.%Sn (1 wt.%Sn), extensive SGBM. **c, d** Mg-0.42 at.%Sn (2 wt.%Sn), widespread SGBM. **e, f** Mg-1.07 at.%Sn (5 wt.%Sn), clear SGBM. **g, h** Mg-1.52 at.%Sn (7 wt.%Sn), localised SGBM.

finding of Shibata and Watanabe that SGBM is not due to grain growth[34] as previously described.

## Thermal stability of the migrated SGBs

To assess the thermal stability of the migrated SGBs, samples of the Mg-0.63 at.%Sn alloy were annealed at 350 °C or 0.7T$_{solidus}$ (T$_{solidus}$: 898 K) of the alloy for 48 h. This annealing condition was chosen to allow for substantial diffusion. Figure 5a, b shows the grain structures in the same field of view before and after annealing, in which arrow 1 was selected as the reference point. No further migration was observed of those already migrated SGBs (arrows 2–4), demonstrating their thermal stability. However, non-migrated tortuous SGBs may migrate during subsequent reheating[44], confirmed in Fig. 5 by referring to arrow 5.

## Discussion
### A model for SGBM

The integrated driving force (often called driving force[45]) for GBM is clear. "The deviation from equilibrium at the interface provides the driving force for the interface migration"[45], leading to a reduction in the total free energy. As observed, after pronounced migration (Fig. 2, Supplementary Fig. 3), SGBs approach their equilibrium state (near 120°-triple junctions plus $d/a \rightarrow \sqrt{3}$, $d$: grain size; $a$: grain side length).

The Gibbs free energy ($G$) of a polycrystalline material, without considering its external surface free energy (constant at constant temperature and pressure), can be written as

$$G = G_B + \sigma A \qquad (1)$$

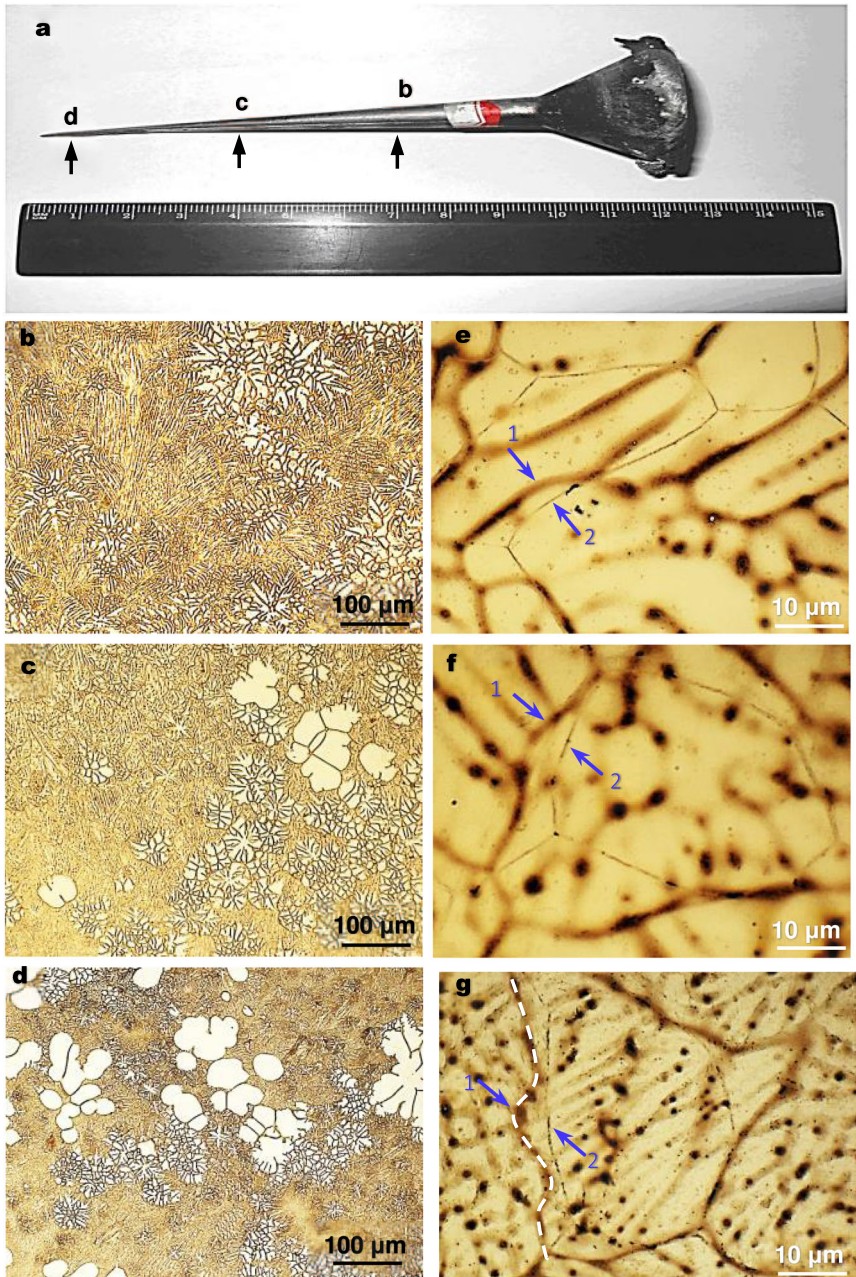

**Fig. 4 | SGBM at different cooling rates. a** An as-cast cone-shaped Mg-0.63 at.%Sn (3 wt.%Sn) specimen. **b**–**d** Grain structures at each specified position in the cone-shaped specimen (**a**). **e**–**g** Closer views of the SGBM in the grain structures at each specified position. Arrow 1 in each micrograph points to the initial SGB position while Arrow 2 indicates the observed SGB position. The cross-sectional diameters are 6.5 mm at **b**, 3.0 mm at **c** and 0.5 mm at **d**. The corresponding cooling rates are 54 °C/s at **b**, 166 °C/s at **c** and 1690 °C/s at **d** (Table 1).

where $G_B$ is the Gibbs free energy of all bulk grains, $A$ is the total GB area, and $\sigma$ is the average specific GB energy. Equation (1) encompasses the contribution of all components to $G$ in both the bulk grains and the GBs. This is reflected in the definition of $\sigma$ via $\sigma A = \sum_{i=1}^{m} n_i^{\Phi}(\mu_i^{\Phi} - \mu_i)$[46], where $n_i^{\Phi}$ is the number of moles of $i$ in the GBs, $\mu_i$ is the chemical potential of $i$ in the bulk grains, and $\mu_i^{\Phi}$ is that in the GBs ($\mu_i \neq \mu_i^{\Phi}$ unless $\sigma = 0$; they are defined differently[46]).

For most metallic materials, their GB thickness (~1 nm) can be assumed to be infinitesimal versus their grain size ($d$)[3,6] (typically $d \geq 10^4$ nm, excluding nano-grains). Therefore, $G_B$ can be treated as being effectively independent of both $n_i^{\Phi}$ (GB composition) and $A$ (GB area)[47,48] or as a constant at constant temperature and pressure.

The elastic strain energy as a source of the driving force (DF) for SGBM is negligible compared to the SGB energy (see Supplementary

Section 4). Applying Eq. (1) to the as-solidified initial SGB state ($I$) and its final or equilibrium state ($E$) allows for the determination of the deviation in free energy from equilibrium, i.e. the DF by definition[45], denoted as $\Delta G_{\text{DF}} = G_E - G_I$. As pointed out earlier, since $G_B$ can be regarded as a constant (excluding nano-grains), we have

$$\Delta G_{\text{DF}} \approx \sigma_E A_E - \sigma_I A_I \tag{2}$$

where the subscript $E$ refers to the equilibrium state and the subscript $I$ refers to the initial state. Equation (2) is consistent with the current understanding of the main DF for SGBM being the reduction in the total GB energy[18,21,25,35].

Since the migrated SGBs always end up inside the grains (Fig. 2), for dilute alloys, $\sigma_E$ can be taken as the average specific GB energy for

the solvent metal grains ($\sigma_0$). Rearranging Eq. (2) with $\sigma_E \approx \sigma_0$ gives

$$\frac{\Delta G_{DF}}{\sigma_0 A_I} \approx -\frac{(A_I - A_E)}{A_I} + \frac{\sigma_0 - \sigma_I}{\sigma_0} \tag{3}$$

As a necessary condition, the occurrence of SGBM requires $\Delta G_{DF} < 0$, i.e. $(A_I - A_E)/A_I > (\sigma_0 - \sigma_I)/\sigma_0$, where $(A_I - A_E)/A_I$ may be regarded as the driving factor and $(\sigma_0 - \sigma_I)/\sigma_0$ as the inhibiting factor for SGBM. It is noteworthy that $\sigma_0$ can vary significantly, depending on the solvent system (Supplementary Tables 1 and 2). Accordingly, $\Delta G_{DF}$ may vary significantly for a similar $A_I$.

The next step is to determine the variations of $(\sigma_0 - \sigma_I)$ for SGBs versus solute type and content. The basic Gibbs equation that connects the specific interfacial energy ($\sigma$) between two phases with their interfacial composition, at constant temperature and pressure, is given by refs. [46], [48]

$$\sum_{i=1}^{m} n_i^\Phi d\mu_i + A \, d\sigma = 0 \tag{4}$$

where $n_i^\Phi$ means the same as above, i.e. the number of moles of component $i$ in the interface ($\Phi$). Equation (4) is applicable to both interfaces and GBs[46,48], providing that the interfacial or GB thickness is infinitesimal versus the size of each phase or grain (excluding nano-grains).

For a dilute binary alloy, both the activity coefficient and the activity of the solvent can approximately be taken as unity by adopting the Raoultian standard state[46] (i.e. the pure solvent substance state), while for the solute ($X_i$) we can write $\mu_i \approx \mu_i^0 + RT\ln X_i$, where $\mu_i^0$ refers to the Henrian standard state[46]. The use of the Raoultian standard state

## Table 1 | Secondary dendrite arm spacing (SDAS, $\lambda$), cooling rate ($\dot{T}$), average grain size ($d$), maximum SGBM distance ($\Delta_{max}$) and ratio of $\Delta_{max}/d$ for dilute Mg-Sn alloys[a]

| Alloy (at.%) | SDAS ($\lambda$, μm) | Cooling rate ($\dot{T}$, °C/s) | Average grain size ($d$, μm) | Maximum GBM ($\Delta_{max}$, μm) | $\Delta_{max}/d$ (%) |
|---|---|---|---|---|---|
| Mg-0.21Sn | 42.5 ± 17.2 | ~8 | 241.5 ± 47.2 | 86.8 | 35.9 |
| Mg-0.30Sn | 42.7 ± 16.5 | ~8 | 213.4 ± 40.1 | 71.2 | 33.4 |
| Mg-0.42Sn | 41.8 ± 15.6 | ~8 | 190.3 ± 38.8 | 60.6 | 31.8 |
| Mg-1.07Sn | 41.5 ± 14.5 | ~8 | 108.1 ± 18.3 | 31.2 | 28.9 |
| Mg-0.63Sn | 41.6 ± 14.8 | ~8 | 144.8 ± 28.0 | 45.8 | 31.6 |
| Mg-0.63Sn | 18.2 ± 5.4 | ~54 | 73.3 ± 13.1 | 23.1 | 31.5 |
| Mg-0.63Sn | 11.1 ± 3.7 | ~166 | 61.9 ± 10.6 | 18.8 | 30.4 |
| Mg-0.63Sn | 4.0 ± 1.4 | ~1690 | 40.8 ± 6.1 | 12.1 | 29.7 |

[a]Raw Experimental Data: Supplementary Information.

for the solute, i.e. the pure solute substance, will just add an extra term $RT\ln\gamma_i^\infty$ to $\mu_i^0$ without affecting the formulation ($\gamma_i^\infty$: the activity coefficient of $i$ at infinite dilution, i.e. $\gamma_i^\infty \equiv (\gamma_i)_{X_i \to 0}$, which is a constant at constant temperature and pressure[46]). Equation (4) can then be written as

$$\frac{d\sigma}{dX_i} \approx -\frac{RT}{X_i}\frac{n_i^\Phi}{A} \tag{5}$$

which is effectively the same as Eq. (3) of Liu and Kirchheim[48] in their formulation of $\sigma$ for solute-segregated non-solidification GBs. The quantity $n_i^\Phi/A$, often denoted as $\Gamma_i$[48], can be expressed through the GB thickness ($\Delta$), GB atomic density ($\rho$, moles/m³) and GB solute composition $X_i^\Phi$ as follows[48,49]

$$\Gamma_i = \frac{n_i^\Phi}{A} = \Delta\rho X_i^\Phi \tag{6}$$

Since the formation of a SGB results from the solidification of the last liquid film (~1 nm thick) with solute content $X_{L(f_s \to 1)}$[3–7], where $f_s$ denotes the solid fraction, we have $X_i^\Phi = X_{L(f_s \to 1)}$ immediately after solidification. The next step is to determine $X_{L(f_s \to 1)}$.

For equilibrium solidification (complete diffusion in both liquid and solid), the remaining liquid composition $X_L$ is defined by the Lever Rule or phase diagram, while for non-equilibrium solidification that involves no solute back-diffusion into the solid (still complete diffusion in the liquid), $X_L$ can be described by the Scheil-Gulliver model (breaking down when $f_s \to 1$)[50]. Both models describe extreme cases, deviating from reality. Consequently, numerous models[51–56] have been proposed to predict solute microsegregation (i.e. $X_L$) during solidification since Scheil[50]. Among these, the Brody-Flemings model (1966)[51] and its five subsequent variations[52–56] have been assessed in detail for various alloy systems. The Won-Thomas version[56], which may be referred to as the six-generation Brody-Flemings model, considered both solute back-diffusion and dendrite coarsening based on previous efforts by Clyne and Kurz[52], Ohnaka[53], Voller[54], and Voller and Beckermann[55]. The model, which retains the basic Brody-Flemings expression, is given by

$$X_L = X_0\left[1 - (1 - \beta k)f_s\right]^{\frac{k-1}{1-\beta k}} \tag{7}$$

$$\beta = 2\alpha^+\left[1 - \exp\left(-\frac{1}{\alpha^+}\right)\right] - \exp\left(-\frac{1}{2\alpha^+}\right) \tag{8}$$

$$\alpha^+ = \alpha + 0.1 = \frac{4D_s\Delta T}{\lambda^2 \dot{T}} + 0.1 \tag{9}$$

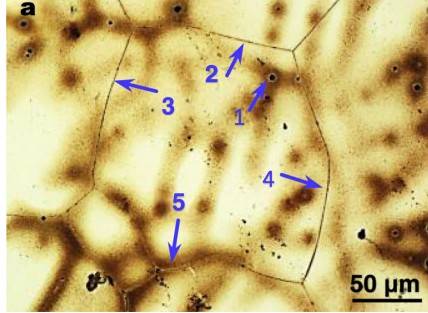
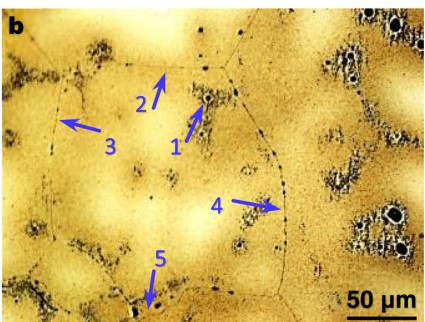

**Fig. 5 | Thermal stability of migrated SGBs in the Mg-0.63 at.%Sn alloy. a** As-cast grain structures with migrated and non-migrated normal SGBs, where arrow 1 was chosen as the reference point; arrows 2–4 indicate the four SGBs that underwent migration; and arrow 5 points to a normal curved SGB that did not exhibit migration. **b** After annealing at 350 °C or 0.7T$_{solidus}$ for 48 h. No further migration was observed of the already migrated four SGBs (arrows 2–4), indicative of their high thermal stability. In contrast, the non-migrated curved SGB (arrow 5 in **a**) displayed clear migration during this annealing treatment.

where $X_L$ (remaining liquid composition), $\lambda$ (SDAS) and $\dot{T}$ (cooling rate) have the same definitions as defined earlier, $X_0$ is alloy composition, $k$ solute partition coefficient, $f_s$ solid fraction, $\triangle T$ actual solidification temperature range ($X_L$ related), and $D_s$ solute diffusion coefficient in the solid ($\triangle T$ related). Equation (7) reduces to the Lever Rule when $\beta = 1$ and the Scheil-Gulliver model when $\beta = 0$. The model offers good predictions for solidification of experimental alloys[57].

Substituting $X_{L(f_s \to 1)}$ according to Eq. (7) for $X_i^{\Phi}$ in Eq. (6) to obtain $n_i^{\Phi}/A$ for Eq. (5) and noting that $X_i$ stands for $X_0$ in Eq. (5), then integrating the resulting Eq. (5) from $\sigma_0$ to $\sigma_1$ for $d\sigma$ and 0 to $X_0$ for $dX_i$, we obtain for $f_s \to 1$

$$\int_{\sigma_0}^{\sigma_1} d\sigma \approx \int_0^{X_0} -RT \triangle \rho \left[1 - (1-\beta k)f_s\right]^{\frac{k-1}{1-\beta k}} dX_i \qquad (10)$$

As detailed in Supplementary Section 6, our systematic assessments of the dependence of $\beta$ on $X_0$, $k$, $\dot{T}$, $\lambda$, $\triangle T$, and $D_s$ for 10 dilute alloy systems, over a wide range of cooling rates (up to 2160 °C/s), have uncovered that $\beta$ can be treated as a constant for a dilute alloy system. In addition, for the sake of simplicity, similar to ref. 48, $\rho$ is taken as the solvent atomic density for dilute alloys (see Supplementary Section 7 for justification). It follows that

$$\sigma_0 - \sigma_1 \approx RT\triangle\rho X_0 \left[1 - (1-\beta k)f_s\right]^{\frac{k-1}{1-\beta k}} \qquad (11)$$

where $f_s \to 1$. Equation (11) provides an estimate of the specific SGB energy ($\sigma$) in a dilute alloy as a function of the solute content ($X_0$), solute type (via $k$), solute back-diffusion and cooling rate (via $\beta$). The value of $\beta$ is listed in Supplementary Table 4 for the alloy systems assessed, which can be similarly determined for any other dilute alloy systems.

It should be noted that the composition of the last liquid film $X_{L(f_s \to 1)}$, which will be inherited by the SGB, arises from the cumulative increase of solute in the remaining liquid (for $k < 1$) with increasing $f_s$, due to the change in solute composition from the liquid state to the solid state (solute partition)[50,58], minus the effect of solute back-diffusion. It is a natural development of the solidification process due to solute partition and back-diffusion. No solute enrichment or microsegregation will occur in the remaining liquid, including the last liquid film, if $k = 1$, which leads to $X_L \equiv X_0$ by Eq. (7), corresponding to no solute partition.

**Prediction of SGBM in other dilute Mg-X alloys and validation**
We first use Eq. (11) to compare the values of $(\sigma_0 - \sigma_I)/\sigma_0$ for Mg-Sn, Mg-Al, Mg-Zn, and Mg-Pb alloys to understand the effect of solute type with respect to the same solute content (e.g. 0.3 at.%). Our basic hypothesis is that since dilute Mg-Sn alloys are prone to SGBM, if any other dilute Mg-X alloys exhibit a similar $(\sigma_0 - \sigma_I)/\sigma_0$ at the same solute content, then they are potentially susceptible to SGBM for comparable values of $(A_I - A_E)/A_I$ by Eq. (3) (excluding nearly hexagon-like SGBs).

Figure 6a, b plots $(\sigma_0 - \sigma_I)/\sigma_0$ versus $X_0$ for Mg-Sn, Mg-Pb, Mg-Al and Mg-Zn alloys up to $X_0 = 0.3$ at.% by focusing on $f_s = 0.99999$ (Fig. 6a) and $f_s = 1$ (Fig. 6b), showing the combined effect of solute type and content. The value of $\sigma_0$ for Mg as a function of temperature is calculated using an established model (Supplementary Section 5)[28]. According to our hypothesis, Mg-Pb and Mg-Al should be inclined to exhibit SGBM, while Mg-Zn should be highly resistant to SGBM. To test our model, four binary Mg-X alloys, Mg-0.3 at.%Sn, Mg-0.3 at.%Pb, Mg-0.3 at.%Al, and Mg-0.3 at.%Zn, were prepared at 720 °C and cast under the same conditions (cooling rate: ~8 °C/s). Figure 6c–f shows their representative grain structures. In addition to Mg-0.3 at.%Sn (Fig. 6c), SGBM is observed in both Mg-0.3 at.%Pb (Fig. 6d) and Mg-0.3 at.%Al (Fig. 6e) but negligible in Mg-0.3 at.%Zn (Fig. 6f). Excellent agreement is obtained. To our knowledge, this should be the first successful

prediction of SGBM in dilute binary alloys and the first theory-based alloy design for SGBM.

To evaluate the relative influence of $(\sigma_0 - \sigma_I)/\sigma_0$ versus $(A_I - A_E)/A_I$, we measured the grain size $d$ and grain side length $a$ in each alloy. Then, we used a truncated octahedron, which is the only Archimedean solid that can offer full space filling (cavity-free)[59], to simulate each equiaxed grain. Table 2 summarises the measured grain size $d$, grain side length $a$, their ratio ($d/a$), and calculated values of $(A_I - A_E)/A_I$ and $(\sigma_0 - \sigma_I)/\sigma_0$. The confluence of $(A_I - A_E)/A_I$ and $(\sigma_0 - \sigma_I)/\sigma_0$ by Eq. (3) agrees well with our experimental observations in each alloy. For instance, the sum of $(A_I - A_E)/A_I - (\sigma_0 - \sigma_I)/\sigma_0$ decreases with increasing solute content from 0.21 at.%Sn to 1.07 at.% Sn, which corresponds to increasingly difficult SGBM shown in Fig. 3. In addition, the driving factor $(A_I - A_E)/A_I$ is markedly greater than the inhibiting factor $(\sigma_0 - \sigma_I)/\sigma_0$ for Mg-0.21 at.%Sn, Mg-0.42 at.%Sn, Mg-0.63 at.%Sn, and Mg-0.3 at.%Pb alloys. Accordingly, significant SGBM was observed in them (Figs. 2 and 3). The effect of solute type (Zn, Al, Sn, Pb) at the same solute content of 0.3 at.% on the inhibiting factor $(\sigma_0 - \sigma_I)/\sigma_0$ is also clear from Table 2.

After SGBM, the ratio of $d/a$ consistently approaches the expected theoretical value of $\sqrt{3}$ for each alloy (Table 2). It was found that most grains in the Mg-0.3 at.%Al alloy displayed an approximate hexagonal or pentagonal shape with $d/a = 1.66$ (close to $\sqrt{3}$) before SGBM (Fig. 6e). Therefore, only limited SGBM was observed. In contrast, although most SGBs are curved and not hexagonal in the Mg-0.3 at.%Zn alloy ($d/a = 1.58$, Fig. 6f), SGBM was negligible. The underlying reason is that the Mg-0.3 at.%Zn alloy has the highest SGBM inhibiting factor of all the alloys studied (Table 2), which is greater than the driving factor, thereby preventing the occurrence of SGBM. Again, the observations agree well with the predictions. The results in Fig. 6 and Table 2 demonstrate the combined effect of solute type and content on SGBM.

**Extension of the model to other binary dilute alloy systems**
To further test the generality of our model, we considered five other binary alloy systems which had exhibited SGBM from the literature. These include Al-4.6 wt.%Mg (Al-5.2 at.%Mg) (Fig. 2b in ref. 10), Al-0.2 wt.%Cu (Al-0.083 at.%Cu) (Fig. 5.25 in ref. 11), Cu-3 wt.%Sn (Cu-1.61 at.%Sn) (Fig. 14 and Fig.15a in ref.13 and Fig. 8 in ref. 14), Ti-6 wt.%Cr (Ti-5.55 at.%Cr) (Fig. 9 in ref. 15), and Zr-2 wt.%Mo (Zr-1.9 at.%Mo) (Fig. 2c in ref. 16). Additionally, we have assessed the Fe-C system for SGBM in stainless steels[18–25]. The inhibiting factor $(\sigma_0 - \sigma_I)/\sigma_0$ for each of these alloy systems and the four Mg-X alloy systems discussed in Fig. 6 is calculated and plotted in Fig. 7a–d by focusing on $f_s = 0.999$ to 1. The dependence of $(\sigma_0 - \sigma_I)/\sigma_0$ on both solute type and content is clear from Fig. 7. In these calculations, the value of $\beta$ for each alloy system is taken from Supplementary Table 4 (the effect of cooling rate on $\beta$ is negligible at < 2160 K/s–Supplementary Table 5), while the value of $\sigma_0$ is taken from Supplementary Table 2. All these alloy systems are predicted to be prone to SGBM based on our hypothesis, providing that their driving factor for SGB is reasonable. Indeed, SGBM is noticeable in each of these alloys[8–25]. The generality of our model is validated.

The coordinate origin in Fig. 7 corresponds to the condition of $(\sigma_0 - \sigma_I)/\sigma_0 = 0$. It should be stressed that this does not mean that SGBM will occur in all high-purity metals, though high-purity Al (99.99 wt.%)[8] and ultrahigh-purity Cu (99.9999 wt.%)[12,13] have both exhibited noticeable SGBM. The basic principle is the same as discussed for the case of dilute binary alloys or $(\sigma_0 - \sigma_I)/\sigma_0 > 0$. For example, if the SGBs in an ultrahigh-purity metal are nearly hexagonal and straight with $d/a$ close to $\sqrt{3}$ or are parallel straight lines, then no SGBM will be expected. In addition, in all the analyses discussed above, we used the average specific GB energy ($\sigma$) without considering the GB structure, including possible changes in GB planes during migration. These factors may become influential for SGBM in certain metal or alloy systems.

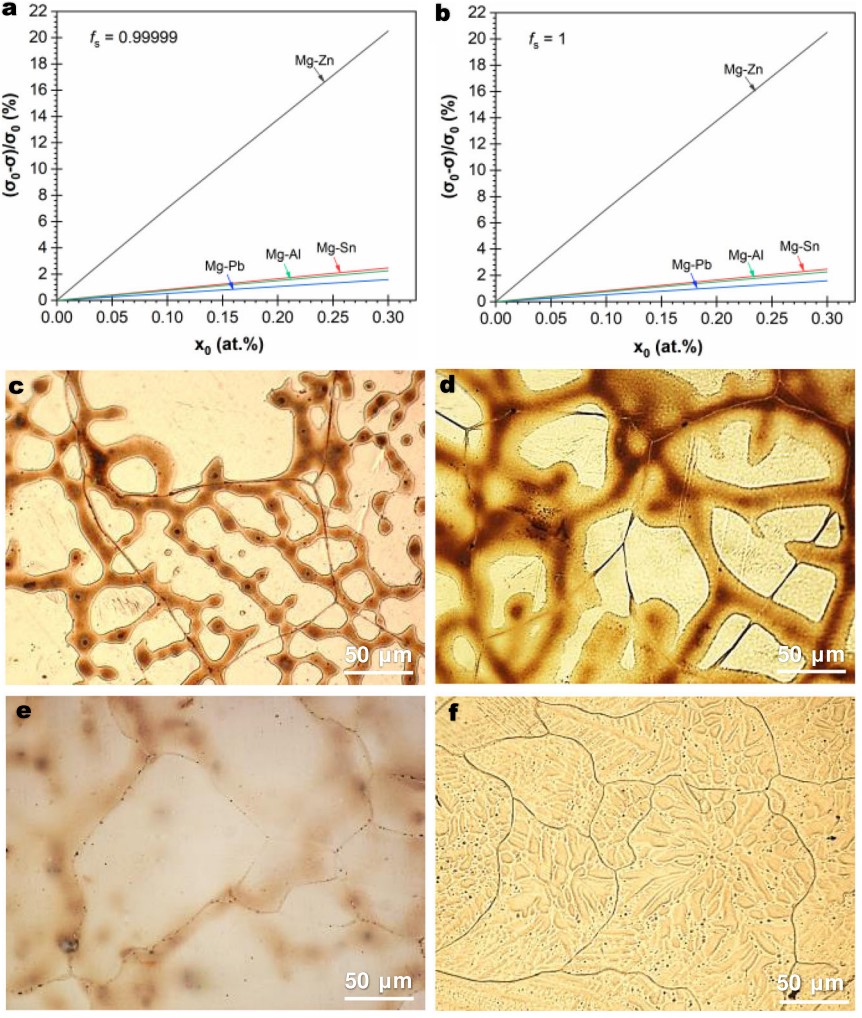

**Fig. 6 | SGBM in Mg-X (X = Sn, Al, Zn) alloys with the same solute content of 0.3 at.%. a, b** Calculated values of $(\sigma_0 - \sigma_l)/\sigma_0$ versus $X_0$ for binary Mg-X alloys using Eq. (11) by focusing on $f_s = 0.99999$ (**a**) and $f_s = 1$ (**b**), showing the influence of solute type on $(\sigma_0 - \sigma_l)/\sigma_0$. **c** Mg-0.3 at.%Sn, profound SGBM. **d** Mg-0.3 at.%Pb, significant SGBM. **e** Mg-0.3 at.%Al, localised SGBM. **f** Mg-0.3 at.%Zn, little SGBM.

**Table 2 | Grain side length (*a*), grain size (*d*), driving factor $(A_I - A_E)/A_I$ and inhibiting factor $(\sigma_0 - \sigma_I)/\sigma_0$ in dilute binary Mg-X alloys (X = Sn, Pb, Al, Zn)[a]**

| Binary Mg-X alloy (at.%) | Average grain side length *a* (μm) | | Average grain size (*d*, μm) | *d*/*a* before SGBM | *d*/*a* after SGBM | Driving factor[b] (%) | Inhibiting factor (%) |
|---|---|---|---|---|---|---|---|
| | Before SGBM | After SGBM | | | | | |
| 0.21Sn | 170.7 ± 31.3 | 137.2 ± 26.8 | 241.5 ± 47.2 | 1.41 ± 0.15 | 1.76 ± 0.11 | 33.57 | 1.73 |
| 0.42Sn | 130.2 ± 25.1 | 111.3 ± 22.7 | 190.3 ± 38.8 | 1.46 ± 0.13 | 1.71 ± 0.12 | 28.78 | 3.44 |
| 0.63Sn | 94.0 ± 23.6 | 84.2 ± 17.9 | 144.8 ± 28.0 | 1.54 ± 0.14 | 1.72 ± 0.11 | 20.76 | 5.13 |
| 1.07Sn | 69.3 ± 10.2 | 63.2 ± 8.1 | 108.1 ± 19.3 | 1.56 ± 0.12 | 1.71 ± 0.09 | 18.69 | 8.60 |
| 0.30Zn | 83.1 ± 16.8 | 83.1 ± 16.8 | 131.3 ± 25.2 | 1.58 ± 0.13 | 1.58 ± 0.13 | 16.59 | 20.51 |
| 0.30Al | 102.3 ± 20.3 | 99.4 ± 15.2 | 170.0 ± 28.3 | 1.66 ± 0.12 | 1.71 ± 0.08 | 7.93 | 2.26 |
| 0.30Sn | 150.3 ± 30.1 | 124.1 ± 22.3 | 213.4 ± 40.1 | 1.42 ± 0.14 | 1.72 ± 0.10 | 32.63 | 2.47 |
| 0.30Pb | 155.1 ± 37.9 | 127.5 ± 29.4 | 221.8 ± 41.5 | 1.43 ± 0.13 | 1.74 ± 0.10 | 31.67 | 1.59 |

[a]Raw Experimental Data: Supplementary Information.
[b]$A \propto a^2$ for a truncated octahedron[59]. Note that $(d/a)_{Equilibrium} = \sqrt{3}$. Hence, $\frac{(A_I - A_E)}{A_I} \approx \frac{3 - (d/a)_I^2}{3}$ if *d* remains unchanged.

For non-high purity metals, it should further be noted that impurity solutes are likely to accumulate in the last liquid and end up at the SGBs along with the intentional solute(s) to affect SGBM. The potential influence of these factors needs to be clarified by detailed studies in the future.

As observed from Fig. 7a–d, once $f_s \geq 0.9999$, the plots look identical. The underlying reason is that the compositon of the last liquid film, $X_{L(f_s \to 1)}$, remains little changed once $f_s \geq 0.9999$ (Supplementary Fig. 4), consistent with the atomistic theory proposed for SGB formation[3].

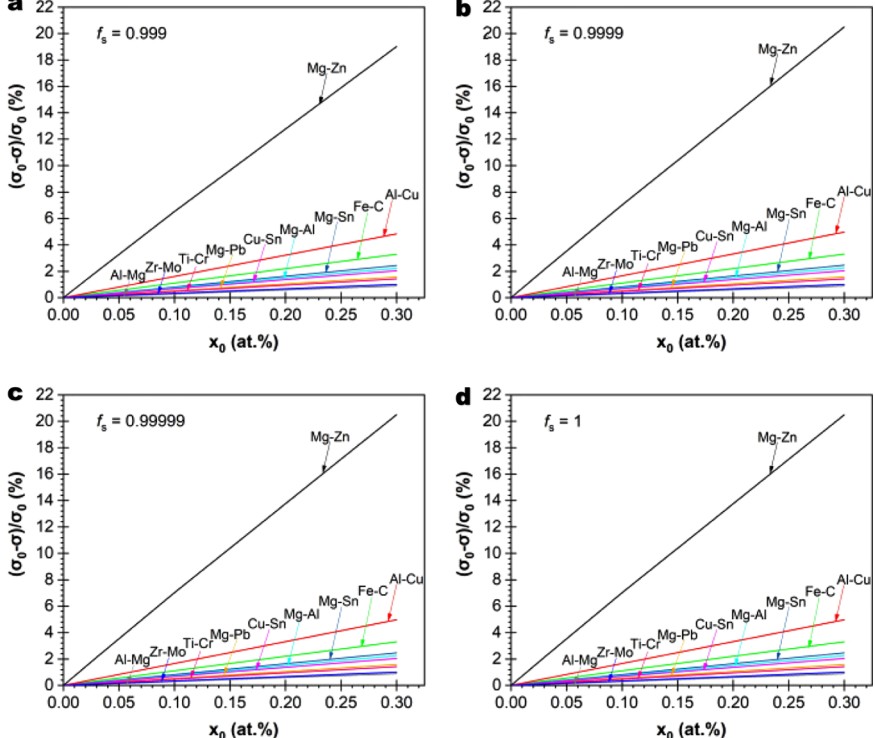

**Fig. 7 | Effect of solute type and content on SGBM assessed through the inhibiting factor $(\sigma_0 - \sigma_I)/\sigma_0$.** The values of $(\sigma_0 - \sigma_I)/\sigma_0$ are calculated using Eq. (11), with the solid fraction $f_s$ being varied from 0.999 to 1. **a** $f_s$ = 0.999 (~100 nm thick liquid film left). **b** $f_s$ = 0.9999 (~10 nm thick liquid film left). **c** $f_s$ = 0.99999 (~1 nm thick liquid film left). **d** $f_s$ = 1 (complete solidification). Once $f_s$ reaches 0.99999, $(\sigma_0 - \sigma_I)/\sigma_0$ remains virtually unchanged (**b–d**).

### Kinetics of SGBM

Immediately after solidification, the SGBs are associated with solute atoms inherited from the last liquid film. It is equivalent to having solute clouds attached to them. Owing to the irregular peripheries of the abutting dendric grains (Fig. 2a), SGBs are usually tortuous. Therefore, they lean towards migrating to their equilibrium or final state. At low temperatures, solute clouds tend to move along with GBs[58,60–62], affecting GBM (solute dragging). However, at high temperatures[58,60,61], when the driving force is sufficiently large[60,61] (versus solute dragging, related to solute content and type[58,60]), GBs cannot be held by solute atoms, leading to solute detachment or breakaway[58,60,61].

The breakaway frees the GBs and can result in athermal GB motion[60,61] until solute reattachment occurs[60,61]. Athermal motion has been studied both theoretically and experimentally[28,46,58,62–67]. It can be regarded as a barrierless process[62] and does not depend on time as there is no need to wait for any thermal activation. It is diffusionless[45] and occurs through a "co-operative movement"[45,58,61,64] or "collective shuffling" mechanism[28,66], by which many atoms can move co-operatively at the same time, with a velocity approaching that of sound in solid[28,58,66]. More specifically, atoms only need to traverse just the interatomic spacing[28,58,64,66,67] to transfer themselves from one side of the GB to the other side, by some form of deformation and/or rotation to realise significant collective movement[45]. In situ observations of atomic-scale GBM in ultra-pure gold films at $0.5–0.74T_m$ using high-resolution transmission electron microscopy have confirmed this collective shuffling GB motion mechanism[66].

As shown in Fig. 4d, g, noticeable SGBM occurred even at the cooling rate of 1690 °C/s when the molten Mg-0.63 at.%Sn alloy was cooled from 720 °C. This is substantially faster than water quenching (diffusionless). Also, SGBM does not depend on cooling time over the broad cooling rate range of 8–1690 °C/s (but related to grain size, Table 1), further confirming its diffusionless nature. The SGBM

distance observed in different Mg alloys in this study ranges from 1 to 87 μm (Table 1, Figs. 2–4, Supplementary Fig. 3). Applying the speed of sound in solid Mg (5770 m/s)[68], the estimated SGBM time ranges from 170 ps to 15 ns, in line with the athermal GB motion investigated using molecular dynamics simulations (nano or picoseconds)[60,69,70]. Therefore, we propose that the SGBM observed in this work has most likely occurred athermally.

## Methods

### Sample preparation

Pure metals of Mg (99.95 wt.%), Sn (99.98 wt.%), Pb (99.93 wt.%), Al (99.95 wt.%), and Zn (99.99 wt.%) were used as feedstock materials to make each alloy, including Mg-(0.2–1.52) at.%Sn (1–7 wt.%Sn), Mg-0.3 at.%Pb (2.5 wt.%Pb), Mg-0.3 at.%Al (0.33 wt.%Al), and Mg-0.3 at.%Zn (0.8 wt.%Zn). A charge of ~500 g of pure metals was melted in a low-carbon steel crucible (80 mm inner diameter and 10-mm wall thickness) at 750 °C, heated in an electrical resistance furnace under the cover gas of $CO_2$-1.0 vol.%$SF_6$. Manual stirring was applied at 30 strokes/min for 3 min during alloying. Plate samples with the dimensions of $140 \times 110 \times 20$ mm³ were cast by pouring the melt at 720 °C into a grey cast-iron mould (30 mm wall thickness) preheated to 250 °C.

To investigate the effect of grain size and cooling rate on SGBM, the Mg-0.63 at.%Sn alloy was cast at 720 °C into a non-preheated conical copper mould (Fig. 4a). The pouring basin was a truncated cone with a base diameter of 40 mm and a depth of 30 mm. The secondary dendrite arm spacing (SDAS) in different locations of the conical sample was measured to estimate the cooling rate. Each set of SDAS data was averaged from 50 measurements.

To investigate the thermal stability of the migrated SGBs, polished and etched samples of the Mg-0.63 at.%Sn alloy were heated to and held at 350 °C for 48 h in a muffle furnace. The samples were placed in a sealed graphite crucible in which clean magnesium swarf was used to

scavenge oxygen. Special marks were made on each sample surface in order to track the same field of view before and after the annealing treatment.

## Microstructure characterisation

Samples for metallographic characterisation were all cut from the central region of each cast plate. They were ground, polished and etched with a solution of 5 vol.% nitric acid in ethyl alcohol. The microstructures were characterised using optical microscopy (Olympus BX60M) and JEOL JSM-7200F scanning electron microscopy (SEM) with a Bruker QUANTAX energy dispersive spectroscopy (EDS) detector. Electron backscatter diffraction (EBSD) was used to identify grain orientations using a JEOL JSM-7200F SEM system, operated at 20 kV with step size 0.3 μm, and a sample-tilt angle of 70°. The samples ($15 \times 10 \times 10$ mm³) were first carefully polished and then subjected to sufficient argon ion etching. The EBSD data were analysed using AztecCrystal, version 1.1 software.

## Data availability

The authors declare that all the original data are available within this article and its Supplementary Information file.

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

## Acknowledgements

We thank Prof. Pavel Lejcek (Czech Academy of Science) and Prof. Johan Du Plessis (RMIT University) for discussions on grain boundary segregation in the solid state. M.Q. acknowledges support from the Australian Research Council (ARC) through DP180103205 and the ARC Industrial Transformation Training Centre IC180100005 (Surface Engineering for Advanced Materials).

## Author contributions

H.L. and M.Q. conceived the idea, designed the experiments, analysed the data and wrote the manuscript. H.L. initiated the study on magnesium alloys and carried out experiments and characterisation. M.Q. developed the theoretical model and extended it to other alloys. S.L. performed EBSD and SEM analyses. Y.Z. prepared specimens and plotted data. H.C. and Y.C. analysed the data and facilitated the experimental work. All authors reviewed and commented on the manuscript.

## Competing interests

The authors declare no competing interests.
