## [Peer Review File · Nature Communications]

Title: Migration of solidification grain boundaries and predictionREVIEWER COMMENTS

Reviewer #1 (Remarks to the Author):

Solidification grain boundary migration is studied in dilute binary alloys. The paper uses the solute-corrected grain boundary energy from reference 68 to predict the resistance to grain boundary migration for a range of alloy systems. This equation shows reasonable correlation with qualitative microstructural observations of the susceptibility of an alloy to gb migration in castings.

The experiments provide good evidence for the phenomenon. The model appears to be consistent with the experiments, at least in this alloy system. But I have some questions:

(1) On the right hand side of Eq (12), the first term (involving the solvent specific GB energy) is the driving force and is negative, and the second term is the resistance to SGBM and is positive. But the authors only consider the second term, and extend it to other alloy systems only based on the comparison of the second term, according to Fig 7. But the first term (the solvent specific GB energy) can vary significantly between different systems. The authors should discuss this in the manuscript, otherwise it might be misleading.

(2) The grain boundary area (grain size and morphology) plays a big role in determining whether SGBM can happen (Eq 12). And the authors chose to change the cooling rate to get different grain size, and thus different gb area, as mentioned from line 338-345. But as shown in Fig 4, there is no difference in terms of the grain size or SGBM distance between the three regions of the cone-shaped sample even though the difference between their cooling rate, as calculated by authors, is over 30 times. So there is little point in studying the effect of cooling rate. Instead, it would be more direct if the authors studied the effect of grain size. The reason why the SGBM distance is different between Fig 2a and Fig 4a (line 340-342) seems to be the grain size, not the cooling rate. The authors should change the wording.

(3) According to the authors, it is all up to the last term in Eq.12 to determine whether SGBM can happen or not, and they did experiments to confirm: Mg-0.3 at.% Al,Pb,Sn and Zn. Based on Fig 6(a), Mg-0.3 at% Pb has lower ($\gamma_0-\gamma$) than Mg-0.3 at% Sn, so SGBM can happen in Mg-0.3at% Pb. While for Mg-0.3 at.% Al and Mg-0.3at% Sn, they have higher ($\gamma_0-\gamma$) than Mg-0.3at % Sn, so it has lower chance to have SGBM. But, in Fig 3(e)-(f), there is clear SGBM for Mg-1.07at% Sn, whose ($\gamma_0-\gamma$) is much higher (twice) than Mg-0.3at.% Al and very close to the Mg-0.2at.% Zn in Fig 6(a). Why Mg-0.2Al has so small SGBM and no SGBM in Mg-0.2at% Zn ?

(4) There are a range of issues with Figure 2 and the associated text:

(4a) On line 88, discussing Fig. 2(c), the authors state “The bright network like regions (Arrows 1 and 4) are Sn-rich dendritic regions”. I think you mean ‘interdendritic’ regions? Also, are the Sn-rich bright regions in Fig. 2(c) Mg₂Sn or are they alpha-Mg with a high Sn content? This should be explained in the text. It would be useful to show the EBSD phase map of this region to help readers understand.

(4b) Do the dark regions in the optical micrographs of Fig. 2(a) and (b) correspond to the bright regions in the BSE-SEM image in Fig. 2(c)? If so, it would be useful to state it in the text.

(4c) In Figure 2(d) (the ebsd orientation map), please provide a color scale to make the figure quantitative. What are the small particles inside each Mg grain? If they are a second phase (e.g. Mg₂Sn), it would be better to show the orientation map for the alpha-Mg phase only. If they are Mg, are they subgrains, recrystallized grains or something else? If they are Mg, can the authors comment on their formation? And does their presence provide any insight into processes that have occurred that may affect gb migration? Also, the misorientation angle between each unit cell does not equal the angle between the <0001> direction; that plotting appears to be wrong- please check.

(4d) Please provide a color scale for the EDS maps in Figure 2(e)-(f). It is not obvious which colors indicate more counts. That is potentially confusing.

(5) on lines 113-118, the authors point out that Mg₂Sn particles at SGBs inhibit migration and imply that they only saw Mg₂Sn at 7wt% Sn and higher. This latter point seems surprising. For the Scheil approximation, we would expect nonequilibrium Mg+Mg₂Sn eutectic at all Sn compositions and, in practice, eutectic seems likely at most of your Mg-Sn compositions given the cooling rates. One might imagine the dark regions in Fig. 3 are Mg+Mg₂Sn eutectic, especially as the area fraction of dark regions increases with Sn content. Could the authors talk about this in the text and, potentially show higher magnification micrograph(s) to convince readers of what the dark regions are in Fig. 2(a)-(b) and Fig.3, and the bright region in Fig. 2(c)?

(6) Can the authors please be more precise about their definition of (i) $X\Phi$, “the solute content from a binary phase diagram beyond which intermetallic phases or compound phases start to dominate”, and (ii) X^* , the “solid solubility”. What definitions are you using to extract $X\Phi$ and X^* from a binary phase diagram? This seems important for determining ΔG_{seg} and $\Gamma\Phi$ in Eq. 13 and, therefore, Fig. 6a.

(7) Lines 218-222 and 232-234: For the dilute compositions discussed in this paper, the authors are assuming equilibrium solidification when they take the last solid to form to have composition X_0 and for grain boundaries to form at the solidus temperature. Surely, limited diffusion in the solid at your cooling rates will cause the final composition of the alpha-Mg to be higher and, for most compositions, to be the solid solubility at the eutectic tie line? (i.e. solidification will be closer to Scheil than Lever). This would result in a bigger difference in composition between the GB and the interior. Does assuming Scheil conditions instead of Lever rule make a significant difference to your analysis?

(8) In Eq.9: why is phi is not capitalized here?

(9) In Eq.11: it is potentially confusing to use k_s for the solidus slope. m_s is more common.

(10) On lines 133-134: the authors use an equation for the SDAS from ref. 66, which is for an Mg-Al-Zn

alloy. The SDAS depends on the phase diagram (k , mL , C_0), interface properties (Γ), and transport properties (DL), (e.g. Mortensen A (1991) Metall Mater Trans 22A:569) and will, therefore, be different for different alloys. Given this, is it reasonable to use this equation for Mg-Sn alloys? This may not be important given point (2).

(11) It is odd to use the words “instant” or “immediately” for processes that occur rapidly.

Optional points to consider

(X) impurity solute (as well as intentional solute) is likely to accumulate in the last liquid to solidify and end up at SGBs. Is this likely to have a significant impact on the analysis.

(X) the authors consider two factors: the change in interface area and the change in specific interfacial energy due to solute. To what extent is the change in specific interfacial energy due to the change in grain boundary plane during migration also important?

(X) In discussing SGBM, is it important to also consider how and when grain boundaries form at the end of solidification. For example, see: Rappaz, M., Jacot, A., & Boettinger, W. J. (2003). Last-stage solidification of alloys: Theoretical model of dendrite-arm and grain coalescence. Metallurgical and Materials Transactions A, 34(3), 467-479.

Reviewer #2 (Remarks to the Author):

The objective of this paper is to study an important grain boundary migration phenomenon occurring during solidification process, SGBM, which is known but less researched. The authors first demonstrated the existence of grain boundary migrations in different Mg alloys under different solidification conditions. The experimental results are interesting and valuable. A major focus of this paper has been put on a theoretical interpretation about the migration mechanism of grain boundaries, in terms of driving force and resistance from the thermodynamic point of view. Based on the resistance term, the authors tried to derive a general criterion to assess the possibility of SGBM in different alloys. However, the deduction process has significant flaws and therefore not convincing. The major problems are as follows,

1. Solute atoms originally position in the GBs have different Gibbs free energy before and after SGBM. In equation (1), the authors did not consider this energy change.
2. The application of Equation (5) in the deduction is wrong. This equation is originally used for calculation of solute concentration at GBs. It cannot be used to deduce ΔG_{seg} , especially with the description of X_c and X_b in the manuscript. With the description of X_c in Equation 8, and the description of X_b with equations (9) and (10), the ΔG_{seg} value will be dependent of X_0 , nominal composition of alloys. This is not correct, because ΔG_{seg} should be a constant value for a GB in a fixed alloy system. It only changes with the atomic structure of GB.

3. Equation (9) is questionable. By insert equation (10) into Eq. (9), one can get $X_b = X_0 \cdot (k+1)/4 + X_\omega / 2$. It means, in any case, the concentration of X_b is larger than $X_\omega / 2$. For Mg-Sn system, $X_b > 16.6$ at%, which is already larger than the eutectic concentration (10.7 at%) and thus impossible.

4. Equation (7) is too rough an estimation about the atomic density at GB. In principle, a substitutional solute atom can also be positioned in the interstitial atomic sites in the GB, if the solute atom is much smaller than the matrix atom, for example Cu in Al alloy.

5. If the criterion proposed in the work is correct, according to Fig. 6 and Fig. 7, one can expect all binary alloys with extremely low solute content will have significant SGBM, which is not reasonable.

Based on above problems, this manuscript is not acceptable for publication in the present form. A substantial improvement for the theoretical part is needed.

Reviewer #3 (Remarks to the Author):

This is an interesting paper about a topic that is not widely described in the open literature and occasionally speculated upon. Thus it is a valuable contribution. The phenomenon termed SGBM in this work is however not often observed in cast alloys, as these alloys commonly form secondary phases that pin grain boundaries. This aspect has not even been mentioned in the manuscript.

Generally I find the approach and especially the experimental work appealing and interesting. I like the fact that alloys were produced to test the model and hypothesis.

Overall, I fail to see from the manuscript the authors argument why this paper is relevant to an audience beyond the metallurgical and materials community.

There are aspects of the model however that I do not agree with.

- I would like to see a more rigorous definition of the driving force for GB segregation. My understanding is that this is a driving force that drives atoms of alloying elements from the bulk to the grain boundary (GB). That is for a stationary GB. It is also accounted for in theories of solute drag. I therefore do not agree with the way this is used in the present context.

- The statements in line 262 - 265 seems to contradict themselves. If Mg-Pb and Mg-Sn alloys show similar resistance, why are Mg-Pb alloys more prone to SGBM? Maybe this is just a problem for formulation and the sentence just needs some re-phrasing.

- line 273 ff: Well, it seems like some alloys do not show any SGBM at all, but I have my doubts on the author's arguments as to why that is the case. The difference in driving force reduction is not that massive but we still see a massive step change in behaviour! I suspect pinning effects.

- line 318 ff: I am not convinced by the diffusion argument developed here. Diffusion leading to GB

motion is not the same as solute diffusion over long distances, but involves a large number of small hops, over low activation energy barriers (due to loose structure in GB) covering short distances of about 1 atomic spacing. The estimate in the manuscript suggests grain boundaries behave like atoms and diffuse by vacancy exchange mechanisms. This would of course be quite slow.

Response Letter

We are grateful to each respected reviewer for their invaluable comments, suggestions, and questions, which have re-shaped a number of important aspects of this work (the intellectual input from each reviewer richly deserves a co-authorship if permitted). We have acknowledged this fact in the Acknowledgements. We would be pleased to further revise the work to make it more meaningful and useful if needed. This study focuses on dilute binary alloys, but we have recently noticed that concentrated alloys or high entropy alloys (HEAs) can show clear solidification grain boundary migration (SGBM) as well (e.g., Senkov et al., Refractory high-entropy alloys, *Intermetallics*, 2010:18;1758, Fig. 5, now included in the revised Introduction).

Briefly, we have **(i)** replaced our original model with a conceptually clearer and simpler one formulated by using the basic Gibbs equation to estimate the specific GB energy and the Scheil model to describe the last-stage solidification; **(ii)** clarified the driving and inhibiting factors for SGBM and assessed their relative importance using a truncated-octahedron grain model based on experimental data; **(iii)** identified the effect of grain size; and **(iv)** considered the particle pinning effect.

A point-to-point response is detailed below, followed by a list of Detailed Changes to the manuscript after this Response Letter (RL).

Reviewer #1 (Remarks to the Author):

Solidification grain boundary migration is studied in dilute binary alloys. The paper uses the solute-corrected grain boundary energy from reference 68 to predict the resistance to grain boundary migration for a range of alloy systems. This equation shows reasonable correlation with qualitative microstructural observations of the susceptibility of an alloy to gb migration in castings.

The experiments provide good evidence for the phenomenon. The model appears to be consistent with the experiments, at least in this alloy system. But I have some questions:

(1) On the right-hand side of Eq (12), the first term (involving the solvent specific GB energy) is the driving force and is negative, and the second term is the resistance to SGBM and is positive. But the authors only consider the second term, and extend it to other alloy systems only based on the comparison of the second term, according to Fig 7. But the first term (the solvent specific GB energy) can vary significantly between different systems. The authors should discuss this in the manuscript, otherwise it might be misleading.

Response: We totally agree with you and thank you for these comments. Indeed, it always depends on the combined effects of both terms. Please refer to our Table 2 below on page 3.

Our original Eq. (12) has now been rearranged as Eq. (3) in our revised manuscript as follows:

$$\frac{\Delta G}{\sigma_0 A_1} \approx - \frac{(A_1 - A_2)}{A_1} + \frac{\sigma_0 - \sigma}{\sigma_0} \quad (3)$$

where A_1 and A_2 are the GB areas before and after SGBM, respectively, σ is the specific GB energy before SGBM (i.e. σ_1), and σ_0 is the specific GB energy after SGBM (i.e. σ_2), taken as the specific solvent GB energy for a dilute alloy. Besides, since γ is often used to denote activity coefficient, we have used σ to denote specific GB energy in the revised manuscript.

SGBM requires $\Delta G < 0$, i.e. $(A_1 - A_2)/A_1 > (\sigma_0 - \sigma)/\sigma_0$, where $(A_1 - A_2)/A_1$ may be regarded as the driving factor for SGBM while $(\sigma_0 - \sigma)/\sigma_0$ as the inhibiting factor. Note that the extent of SGBM is further related to the magnitude of ΔG and hence to both A_1 and σ_0 , where σ_0 can vary significantly between different solvent systems (e.g. see Supplementary Table S1).

To quantify $(A_1 - A_2)/A_1$ in Eq. (3), to a first approximation, we treat each equiaxed grain as a truncated octahedron (see **Fig. RL1** below), which is the only Archimedean solid that can offer cavity-free full space filling (S. Torquato and Y. Jiao, “Dense packings of polyhedra: Platonic and Archimedean solids”, *Physical Review E*, 2009, 80, 041104). The surface area of a truncated octahedron (A), which consists of eight hexagonal planes and six squares of the same side length (a), equals $(6 + 12\sqrt{3})a^2$, and its volume (V) is given by $8\sqrt{2}a^3$ [1].

Total surface: $A = (6 + 12\sqrt{3})a^2$
 Total volume: $V = 8\sqrt{2}a^3$

[1] Robert Williams (1979). *The Geometrical Foundation of Natural Structure: A Source Book of Design*, page 78, New York, Dover Publications, Inc.

Fig. RL1 A truncated octahedron with side length a (see also <https://mathworld.wolfram.com/ArchimedeanSolid.html>)

Table 2 (next page) summarizes the grain side length (a), grain size (d), ratio of d/a , and calculated values of $(A_1 - A_2)/A_1$ and $(\sigma_0 - \sigma)/\sigma_0$, where the value of $(\sigma_0 - \sigma)/\sigma_0$ was calculated using our new model (please see our response to Q6). As pointed out earlier, the occurrence of SGBM requires $\Delta G < 0$, which depends on both $(A_1 - A_2)/A_1$ and $(\sigma_0 - \sigma)/\sigma_0$ by Eq. (3). The results in **Table 2** can nicely explain our observations in each alloy.

In addition, as shown in **Table 2**, after SGBM, the ratio of d/a has consistently approached the expected theoretical value of 1.732 or $\sqrt{3}$. In contrast, before SGBM the ratio of d/a is markedly lower than $\sqrt{3}$. This interprets the SGBM from a different perspective.

We have revised our manuscript. Please refer to the second part of this RL and the manuscript.

Table 2 Grain side length (a), grain size (d), driving factor $(A_1-A_2)/A_1$ and inhibiting factor $(\sigma_0-\sigma)/\sigma_0$ in dilute binary Mg-X alloys ($X = \text{Sn, Pb, Al, Zn}$)

Binary Mg-X alloys	Average grain side length ^{1*} (μm)		Average grain size (d , μm)	d/a_1	d/a_2	Driving factor $(A_1-A_2)/A_1$ (%) ^{2*}	Inhibiting factor $(\sigma_0-\sigma)/\sigma_0$ (%) ^{3*}
	Before SGBM (a_1)	After SGBM (a_2)					
0.21at.%Sn	170.7 ± 31.3	137.2 ± 26.8	241.5 ± 47.2	1.41 ± 0.15	1.76 ± 0.11	35.41	2.17
0.42at.%Sn	130.2 ± 25.1	111.3 ± 22.7	190.3 ± 38.8	1.46 ± 0.13	1.71 ± 0.12	26.90	4.48
0.63at.%Sn	94.0 ± 23.6	84.2 ± 17.9	144.8 ± 28.0	1.54 ± 0.14	1.72 ± 0.11	19.76	6.95
1.07at.%Sn	69.3 ± 10.2	63.2 ± 8.1	108.1 ± 18.3	1.56 ± 0.12	1.71 ± 0.09	16.83	12.80
0.30at.%Zn	83.1 ± 16.8	83.1 ± 16.8	131.3 ± 25.2	1.58 ± 0.13	1.58 ± 0.13	0 ^{4*}	15.71
0.30at.%Al	102.3 ± 20.3	99.4 ± 15.2	170.0 ± 28.3	1.66 ± 0.12	1.71 ± 0.08	5.59	4.26
0.30at.%Sn	150.3 ± 30.1	124.1 ± 22.3	213.4 ± 40.1	1.42 ± 0.14	1.72 ± 0.10	31.82	3.77
0.30at.%Pb	155.1 ± 37.9	127.5 ± 29.4	221.8 ± 41.5	1.43 ± 0.13	1.74 ± 0.10	32.42	2.39

1*: The average grain side length was calculated by dividing the measured average grain circumference by 6.

2*: The surface area (A) of a truncated octahedron is equal to $(6 + 12\sqrt{3})a^2$ [1].

3*: $\sigma_0 = 0.52 \text{ J/m}^2$ for pure Mg from 640°C to 650°C (Supplementary Table S1).

4*: Although $(A_1-A_2)/A_1 = 0$ due to the absence of migration, the driving force for migration still exists because of their low average value of d/a (1.58, less than 1.732) and non-hexagonal shapes.

[1] Williams, R. *The Geometrical Foundation of Natural Structure: A Source Book of Design*. 78 (Dover Publications, Inc., New York, 1979).

(2) The grain boundary area (grain size and morphology) plays a big role in determining whether SGBM can happen (Eq 12). And the authors chose to change the cooling rate to get different grain size, and thus different gb area, as mentioned from line 338-345. But as shown in Fig 4, there is no difference in terms of the grain size or SGBM distance between the three regions of the cone-shaped sample even though the difference between their cooling rate, as calculated by authors, is over 30 times. So there is little point in studying the effect of cooling rate. Instead, it would be more direct if the authors studied the effect of grain size. The reason why the SGBM distance is different between Fig 2a and Fig 4a (line 340-342) seems to be the grain size, not the cooling rate. The authors should change the wording.

Response: Thank you for another very insightful comment. Accordingly, we have measured the average grain size (d) and the maximum SGBM distance (Δ_{max}) observed in each alloy with respect to its cooling rate. The results are summarized in **Table 1** below (next page). **Indeed**, the ratio of Δ_{max}/d is **effectively independent of the cooling rate** from 8 °C/s to 1690 °C/s. This seems to be in line with the effect of grain size on the migration of non-solidification GBs based on *limited* experimental data: “the average migration distance appears to scale approximately with the grain diameter if all other conditions are held constant” [1].

Table 1 Secondary dendrite arm spacing (SDAS, λ_2), cooling rate (\dot{T}), average grain size (d), maximum SGBM distance (Δ_{\max}) and ratio of Δ_{\max}/d for dilute Mg-Sn alloys

Alloy	SDAS (λ_2 , μm)	Cooling rate (\dot{T} , $^{\circ}\text{C/s}$)	Average grain size (d, μm)	Maximum GBM (Δ_{\max} , μm)	Δ_{\max}/d (%)
Mg-0.21at.%Sn	42.5 \pm 17.2	~8	241.5 \pm 47.2	86.8	35.9
Mg-0.30at.%Sn	42.7 \pm 16.5	~8	213.4 \pm 40.1	71.2	33.4
Mg-0.42at.%Sn	41.8 \pm 15.6	~8	190.3 \pm 38.8	60.6	31.8
Mg-1.07at.%Sn	41.5 \pm 14.5	~8	108.1 \pm 18.3	31.2	28.9
Mg-0.63at.%Sn	41.6 \pm 14.8	~8	144.8 \pm 28.0	45.8	31.6
Mg-0.63at.%Sn	18.2 \pm 5.4	~54	73.3 \pm 13.1	23.1	31.5
Mg-0.63at.%Sn	11.1 \pm 3.7	~166	61.9 \pm 10.6	18.8	30.4
Mg-0.63at.%Sn	4.0 \pm 1.4	~1690	40.8 \pm 6.1	12.1	29.7

We have changed the original subtitle “The effect of cooling rate on SGBM” to “The effect of grain size on SGBM through changing cooling rate” and rephrased that section as follows:

The effect of grain size on SGBM through changing cooling rate. Grain size is likely to affect SGBM. This influence was investigated through changing cooling rate from 8 $^{\circ}\text{C/s}$ to 1690 $^{\circ}\text{C/s}$ The maximum SGBM distance (Δ_{\max}) observed at each cooling rate was measured along with grain size (d). The results are included in Table 1. The ratio of Δ_{\max}/d varies between 30% and 40%, which is essentially independent of the range of cooling rate investigated (8-1690 $^{\circ}\text{C/s}$). This observation is in line with the influence of grain size observed on migration of non-solidification GBs based on limited experimental data [1]. No change in grain number was observed after SGBM at each cooling rate.

[1] King, A. H. Diffusion induced grain boundary migration. *Inter. Mater. Rev.* **32**, 173-189 (1987).

(3) According to the authors, it is all up to the last term in Eq.12 to determine whether SGBM can happen or not, and they did experiments to confirm: Mg-0.3 at.% Al, Pb, Sn and Zn. Based on Fig 6(a), Mg-0.3 at.% Pb has lower ($\gamma_0-\gamma$) than Mg-0.3 at.% Sn, so SGBM can happen in Mg-0.3at.% Pb. While for Mg-0.3 at.% Al and Mg-0.3at.% Sn, they have higher ($\gamma_0-\gamma$) than Mg-0.3at.% Sn, so it has lower chance to have SGBM. But, in Fig 3(e)-(f), there is clear SGBM for Mg-1.07at.% Sn, whose ($\gamma_0-\gamma$) is much higher (twice) than Mg-0.3at.% Al and very close to the Mg-0.3at.% Zn in Fig 6(a). Why Mg-0.3Al has so small SGBM and no SGBM in Mg-0.3at.% Zn?

Response: Thank you again. As appointed out earlier and shown by our new Table 2, it should always depend on the combined effects of $(A_1-A_2)/A_1$ (the driving factor) and $(\sigma_0-\sigma)/\sigma_0$ (the inhibiting factor). We have re-interpreted our experimental observations as follows:

Mg-xSn (x = 0.21, 0.42, 0.63, at.%) and Mg-0.3at.%Pb: According to our **Table 2** above,

$(A_1-A_2)/A_1$ is markedly greater than $(\sigma_0-\sigma)/\sigma_0$ for each of these four alloys, hence significant SGBM was observed in each of them.

Mg-0.3at.%Al: Its $(A_1-A_2)/A_1$ (5.59%) is higher than $(\sigma_0-\sigma)/\sigma_0$ (4.26%) by **Table 2**. Hence, SGBM occurs. However, since most grains in this alloy (see **Fig. 6c in the manuscript**) show an approximate hexagonal shape with $d/a_1=1.66$ (close to $\sqrt{3}$), SGBM is less significant.

Mg-1.07at.%Sn: Its $(A_1-A_2)/A_1$ is 16.8%, greater than its $(\sigma_0-\sigma)/\sigma_0$ (12.8%) (**Table 2**). Therefore, clear migration occurred, despite its relatively high inhibiting factor.

Mg-0.3at.%Zn: Few hexagonal grains were observed in this alloy (**Fig. 6d in the manuscript**). In addition, its grains exhibit a low value of d/a_1 ($1.58 < 1.732$). Both observations are indicative of the absence of SGBM. However, the noticeable departure of its d/a_1 ratio (1.58) from the equilibrium ratio (1.732) implies that thermodynamically its grains should move towards reaching their equilibrium shapes. The underlying reason for the absence of SGBM is that this alloy has the highest SGBM-inhibiting factor of all the alloys studied (**Table 2**), which prevented the occurrence of SGBM. Again, the observations agree with predictions.

Table 2 also reveals that the value of $[(A_1-A_2)/A_1 - (\sigma_0-\sigma)/\sigma_0]$ decreases with increasing content of Sn for the five Mg-xSn ($x = 0.21, 0.30, 0.42, 0.63, 1.07$) alloys studied, which is consistent with the increasingly difficult SGBM observed in them.

We have accordingly revised our manuscript. Please refer to the second part of this RL and our revised manuscript.

(4) There are a range of issues with Figure 2 and the associated text:

(4a) On line 88, discussing Fig. 2(c), the authors state “The bright network like regions (Arrows 1 and 4) are Sn-rich dendritic regions”. I think you mean ‘interdendritic’ regions? Also, are the Sn-rich bright regions in Fig. 2(c) Mg₂Sn or are they alpha-Mg with a high Sn content? This should be explained in the text. It would be useful to show the EBSD phase map of this region to help readers understand.

Response: Our amended **Fig. 2** is shown below (next page). Yes, we meant Sn-enriched interdendritic regions. Thank you. The bright network-like regions are confirmed to be α -Mg(Sn) containing 5-15 wt.%Sn (see **Fig. RL2 after Fig. 2**), compatible with the solubility of Sn in α -Mg (15 wt.%) at the eutectic temperature. We have added a forescatter detector (FSD) image, **Fig. 2c**, which provides atomic number contrast information, and an EBSD IPF map (**Fig. 2d**). Unexpectedly, the Sn-rich interdendritic regions appeared as zero-solution regions (i.e. they cannot be indexed as Mg by the AZtec software, the same for the EBSD phase maps), due to Sn-induced lattice interference. We have rephrased the original text about **Fig. 2**.

Fig. 2 SGBM in the Mg-0.63at.%Sn (3wt.%Sn) alloy. **(a, b)** Optical micrographs showing massive SGBM **(a)** and localized SGBM **(b)**. **(c)** FSD image showing migrated SGBs (arrows 1, 2, 3) and Sn-enriched interdendritic regions (bright). **(d)** EBSD IPF map of **(c)**, where the dark regions are zero-solution regions, i.e. they cannot be indexed as Mg by the AZtec software due to Sn enrichment. **(e, f)** Energy dispersive spectroscopy (EDS) mapping of Sn and Mg of **(c)**. **(g)** Backscattered electron (BSE) image showing migrated SGBs (blue arrows), SGB particles (red arrows), Sn-enriched interdendritic regions (white arrows) and eutectic α -Mg+Mg₂Sn **(h)**.

Fig. RL2 Line scans showing the composition profiles of Mg and Sn across the bright network-like regions and migrated GBs in the Mg-0.63at.%Sn alloy. The bright regions are confirmed to be Sn-rich and contain 5-15 wt.%Sn, compatible with the solubility limit of Sn in α -Mg (15 wt.%) at the eutectic temperature. **(Supplementary Information)**

(4b) Do the dark regions in the optical micrographs of Fig.2(a) and (b) correspond to the bright regions in the BSE-SEM image in Fig. 2(c)? If so, it would be useful to state it in the text.

Response: Yes, they do. We have specified this point in the revised text. Thank you.

(4c) In Figure 2(d) (the ebsd orientation map), please provide a color scale to make the figure quantitative. What are the small particles inside each Mg grain? If they are a second phase (e.g. Mg_2Sn), it would be better to show the orientation map for the alpha-Mg phase only. If they are Mg, are they subgrains, recrystallized grains or something else? If they are Mg, can the authors comment on their formation? And does their presence provide any insight into processes that have occurred that may affect gb migration? Also, the misorientation angle between each unit cell does not equal the angle between the $\langle 0001 \rangle$ direction; that plotting appears to be wrong- please check.

Response: Please see our detailed response below point by point.

Fig. 2(d): a color code triangle has been added to indicate grain orientations.

The small particles inside each Mg grain: we have added **Fig. 2 (g, h)** (see above) to show that they are eutectic α -Mg(Sn)+ Mg_2Sn or devolved eutectic Mg_2Sn . They exist in those Sn-

enriched bright network-like regions (last liquid to solidify) as shown in **Fig. 2g**, predictable by the Scheil model. **Fig. RL3** (next page) shows more observations (see also Response to Q5).

Insight into processes that may affect gb migration: Thank you again! The **first** insight is that **the Scheil model**, rather than the lever rule, should have been used. For instance, the Mg-Sn phase diagram suggests a minimum of 3.35at.%Sn for eutectic formation in Mg-Sn alloys but the Mg-0.63at.%Sn alloy has already shown eutectic phases (**Fig. 2g**).

The **second** insight is that eutectic Mg₂Sn particles can inhibit local SGBM through their GB-pinning effects (see red arrows in **Fig. 2g** above). However, as indicated by the blue arrows in **Fig. 2g**, the two GB segments that are ~20-30 μm away from the pinning particles (Mg₂Sn) still showed substantial migration. In fact, we have found that SGBM can occur even if there is noticeable eutectic formation (see **Fig. RL3** next page). This suggests that a prominent presence of particles is necessary to completely suppress SGBM (our response to Q1 of Reviewer #3 provides more details about GB particle pinning effects).

The **last** insight deals with the selection of the cut-off value (f_L) for the Scheil model in predicting SGBM. As can be seen from **Fig. 2** above, even though there has been clear formation of eutectic phases in the Mg-0.63at.%Sn alloy, the alloy has still shown massive SGBM (**Fig. 2a** and **2b**). This has important implications for the selection of f_L .

Wonderful comments and suggestions indeed! We have amended our manuscript accordingly. Please see the second part of this RL and our revised manuscript.

Misorientation angle between each unit cell does not equal the angle between the <0001> direction: Thank you. We have re-done our EBSD characterization. In **Fig. 2d**, the misorientation angle/axis pairs are now characterized to be 32.5°/<5 $\bar{2}$ 70> between the blue and purple grains, 32.9°/<5 $\bar{4}$ 16> between the blue and green grains, and 35.5°/<2 $\bar{7}$ 56> between the purple and green grains. As expected, they are all high-angle GBs. After migration, the GB triple junction in **Fig. 2d** (130°-115°-115°) is close to its 120°-120°-120° equilibrium condition (assuming that the specific GB energy is the same), indicative of the stable state of the migrated GBs or why migration has occurred.

(4d) Please provide a color scale for the EDS maps in Figure 2(e)-(f). It is not obvious which colors indicate more counts. That is potentially confusing.

Response: A color scale has been added to **Fig. 2(e, f)**. The unit is counts/second (cps).

(5) on lines 113-118, the authors point out that Mg₂Sn particles at SGBs inhibit migration and imply that they only saw Mg₂Sn at 7wt% Sn and higher. This latter point seems surprising. For the Scheil approximation, we would expect nonequilibrium Mg+Mg₂Sn eutectic at all Sn compositions and, in practice, eutectic seems likely at most of your Mg-Sn compositions given the cooling rates. One might imagine the dark regions in Fig. 3 are Mg+Mg₂Sn eutectic, especially as the area fraction of dark regions increases with Sn content. Could the authors talk about this in the text and, potentially show higher magnification micrograph(s) to convince readers of what the dark regions are in Fig. 2(a)-(b) and Fig.3, and the bright region in Fig. 2(c)?

Response: Thank you. Our original statements were problematic. As shown earlier in **Fig. 2(g, h)** above, eutectic α -Mg(Sn)+Mg₂Sn phases existed in the dark regions in the Mg-0.63at.%Sn alloy and their presence increased with increasing Sn content beyond 0.63at.%Sn. Please see **Fig. RL3** below about the eutectic phases observed in the Mg-1.52at.%Sn (7wt.%Sn)

Fig. RL3 BSE image of Mg-1.52at.%Sn (7wt.%Sn) showing eutectic α -Mg(Sn)+Mg₂Sn.

We have also studied the two low Sn content alloys, Mg-0.21at.%Sn and Mg-0.3at.%Sn. Both alloys contained negligible eutectic phases, although the classical Scheil model still predicts limited eutectic formation. Then, we considered the Scheil model modified by Kurz and Fisher [1] (with back-diffusion), which predicts no eutectic formation in Mg-0.21at.%Sn even by setting $f_L = 0$. As for Mg-0.3at.%Sn, the modified Scheil model predicts limited eutectic formation only if $f_L \leq 0.001$. We have thus rephrased our statements in the revised manuscript as follows:

Another observation is that limited eutectic α -Mg(Sn)+Mg₂Sn phases formed in certain of the interdendritic regions as exemplified in Fig. 2 (g, h), predictable by the Scheil model (a minimum of 3.35at.%Sn is, however, required by the lever rule). Their presence increased with increasing Sn content beyond 0.63at.%Sn (Supplementary Fig. S2) but is negligible in Mg-0.21at.%Sn and Mg-0.3at.%Sn alloys due to their low solute (Sn) content and the likely back-diffusion in the last-stage solidification [2].

[1] Kurz, W. & Fisher, D. J. *Fundamentals of solidification*. 127-134 (Trans Tech Pub. 1989).

[2] Rappaz, M., Jacot, A. & Boettinger, W. J. Last-stage solidification of alloys: theoretical model of dendrite-arm and grain coalescence. *Metall. Mater. Trans. A* **34A**, 467-479 (2003).

(6) Can the authors please be more precise about their definition of (i) $X\Phi$, “the solute content from a binary phase diagram beyond which intermetallic phases or compound phases start to dominate”, and (ii) X^* , the “solid solubility”. What definitions are you using to extract $X\Phi$ and X^* from a binary phase diagram? This seems important for determining ΔG_{seg} and $\Gamma\Phi$ in Eq. 13 and, therefore, Fig. 6a.

Response: Inspired by your comments and suggestions in conjunction with those from Reviewers #2 and #3, as mentioned earlier, we have developed a conceptually clearer and simpler model by using the basic Gibbs equation and the Scheil model. Consequently, both $X\Phi$ and X^* are no longer needed. The formulation of our new model is detailed below by following the same equation numbering sequence in the revised manuscript.

The basic Gibbs equation that connects the specific interfacial energy (σ) between two phases α and β with their interfacial composition at constant temperature and pressure is given by [1, 2]

$$\sum_{i=1}^m n_i^\Phi d\mu_i + A d\sigma = 0 \quad (4)$$

where n_i^Φ means the same as above, i.e., the number of moles of component i in the interface (Φ). Eq. (4) is applicable to both interfaces and GBs [1, 2], providing that the interfacial or GB thickness is infinitesimal vs. the size of each phase or each grain.

For a dilute binary alloy, both the activity coefficient and the activity of the solvent can approximately be taken as unity by adopting the Raoultian standard state (i.e. the pure solvent substance state) [1], while for the solute (X_i) we can write $\mu_i \approx \mu_i^0 + RT \ln X_i$ where μ_i^0 corresponds to the Henrian standard state [1]*. Accordingly, Eq. (4) can be written as

$$\frac{d\sigma}{dX_i} \approx - \frac{RT}{X_i} \frac{n_i^\Phi}{A} \quad (5)$$

which is effectively the same as Eq. (3) of Ref. [3] used by Liu and Kirchheim in their formulation of σ for solute-segregated non-solidification GBs. The quantity n_i^Φ/A , often denoted as Γ_i [1], can be expressed through the GB thickness (Δ), GB atomic density (moles/m³) and GB solute composition X_i^Φ as follows [3, 4]

$$\Gamma_i = \frac{n_i^\Phi}{A} = \Delta \rho X_i^\Phi \quad (6)$$

* The use of the Raoultian standard state for the solute, i.e., the pure solute substance state, will just add an extra term $RT \ln \gamma_i^\infty$ to μ_i^0 without affecting the formulation, where γ_i^∞ is the activity coefficient of i at the point of infinite dilution, i.e., $\gamma_i^\infty \equiv (\gamma_i)_{X_i \rightarrow 0}$, which is a constant at constant temperature and pressure [1]. 10

The formation of a SGB results from a closing liquid film between two coalescing grains of different orientations in the last-stage solidification [5]. Hence, to a first approximation, the average SGB composition (X_i^ϕ) can be assumed to be that of the last solidifying liquid (X_L) according to the Scheil model below [6]

$$X_L = X_0(f_L)^{k_0-1} \quad (7)$$

where X_0 is the alloy composition, k_0 is the solute partition coefficient on the phase diagram in molar fraction, and f_L is the volume fraction of the last liquid. Note that the notation X_0 in Eq. (7) means the same as X_i in Eq. (5). Substituting X_L or Eq. (7) as X_i^ϕ into Eq. (6) and then into Eq. (5), we obtain by integrating Eq. (5) from σ_0 to σ for $d\sigma$ and 0 to X_0 for dX_i

$$\int_{\sigma_0}^{\sigma} d\sigma \approx \int_0^{X_0} -RT\Delta\rho (f_L)^{k_0-1} dX_i \quad (8)$$

For simplicity purposes, following Ref. [3], ρ is taken as the solvent atomic density for dilute binary alloys (see Supplementary Section 4 for justification). It follows that

$$\sigma - \sigma_0 \approx -RT\Delta\rho(f_L)^{k_0-1}X_0 \quad (9)$$

Eq. (9) provides an estimate of the specific SGB energy (σ) in a dilute binary alloy. Note that when the Scheil model Eq. (7) is expressed in weight percentage, Eq. (9) changes to a complex equivalent form, see Supplementary Eq. (S3).

Fig. RL4 below (next page) shows our new predictions using Eq. (9). Together with Table 2, the results can nicely explain our experimental observations and those in the literature too.

We have revised our manuscript with the new model described above. Please see the second part of this RL and our revised manuscript.

- [1] Lupis, C. H. Chemical thermodynamics of materials. *Elsevier*, 347-392, 158-183 (1983).
- [2] Gottstein, G. & Shvindlerman, L. S. *Grain boundary migration in metals: thermodynamics, kinetics, applications. 2nd edition*, 12-13 (Equation 1.49) (CRC Press, Taylor & Francis Group, Boca Raton, 2010).
- [3] Liu, F. and Kirchheim, R. Nano-scale grain growth inhibited by reducing grain boundary energy through solute segregation. *J. Cryst. Growth* **264**, 385-391(2004).
- [4] Lejček, P. *Grain Boundary Segregation in Metals*, 143-144 (Heidelberg Univ., Heidelberg, 2010).
- [5] Rappaz, M., Jacot, A. & Boettinger, W. J. Last-stage solidification of alloys: theoretical model of dendrite-arm and grain coalescence. *Metall. Mater. Trans. A*, **34**, 467-479 (2003).
- [6] Porter, D. A. & Easterling, K. E., *Phase transformations in metals and alloys, 3rd edition*. 211-213 (CRC Press. 2009).

Figure RL4 Calculated values of $(\sigma - \sigma_0)/\sigma_0$ for 11 dilute binary alloy systems using Eq. (9). The selection of $f_L = 0.025$ is based on (i) the approximate minimum width of the Sn-enriched regions in Mg-0.63at.%Sn vs. grain size (see Fig. 2c and 2g), and (ii) the massive SGBM in Mg-0.63at.%Sn (Fig. 2c), despite clear eutectic formation (Fig. 2g and 2f). The selection of $f_L = 0.1$ for Fe-C and Fe-P is due to the use of smaller f_L values leading to excessive C and P (e.g. $X_L = 13.4\text{at.\%C}$ at $f_L = 0.025$ for Fe-0.3at.%C, which means substantial formation of eutectic carbides, unrealistic for this nearly pure Fe composition, Fe-0.065wt.%C).

(7) Lines 218-222 and 232-234: For the dilute compositions discussed in this paper, the authors are assuming equilibrium solidification when they take the last solid to form to have composition X_0 and for grain boundaries to form at the solidus temperature. Surely, limited diffusion in the solid at your cooling rates will cause the final composition of the alpha-Mg to be higher and, for most compositions, to be the solid solubility at the eutectic tie line? (i.e. solidification will be closer to Scheil than Lever). This would result in a bigger difference in composition between the GB and the interior. Does assuming Scheil conditions instead of Lever rule make a significant difference to your analysis?

Response: We must thank you for this suggestion! We have already followed it in our response to Q6 and the discussions above. It makes our predictions much closer to reality than the lever rule. Inspired by this suggestion, we have also considered the Scheil model modified by Kurz and Fisher [1] (with back-diffusion), which works nicely too for the binary Mg-X alloys studied in this work. However, the use of the modified Scheil model requires reliable experimental data

on secondary dendrite arm spacing (SDAS) and cooling rate (the predictions are sensitive to both sets of input data). This makes it somehow inconvenient for “experimentally unknown” alloys. We have thus cleaved to the classical Scheil model for our predictions.

[1] Kurz, W. & Fisher, D. J. *Fundamentals of solidification*. p127-134 (Trans Tech Pub. 1989).

(8) In Eq.9: why is phi is not capitalized here?

Response: Thank you for the careful read. We have made all the symbols consistent.

(9) In Eq.11: it is potentially confusing to use ks for the solidus slope. ms is more common.

Response: We agree and have made changes to avoid such potential confusion.

(10) On lines 133-134: the authors use an equation for the SDAS from ref. 66, which is for an Mg-Al-Zn alloy. The SDAS depends on the phase diagram (k , m_L , C_0), interface properties (Γ), and transport properties (DL), (e.g. Mortensen A (1991) *Metall Mater Trans* 22A:569) and will, therefore, be different for different alloys. Given this, is it reasonable to use this equation for Mg-Sn alloys? This may not be important given point (2).

Response: Thank you. Fortunately, we have found systematic experimental data on SDAS and cooling rate for binary Mg-Sn alloys [1]. They are indeed different. For example, the highest cooling rate was increased from the previously calculated ~ 1450 °C/s to 1690 °C/s and the slowest cooling from ~ 5.5 °C/s to ~ 8 °C/s. We have amended our Table 1. Thank you.

[1] Dev, A. & Paliwal, M. Influence of solute elements (Sn and Al) on microstructure evolution of Mg alloys: An experimental and simulation study. *J. Cryst. Growth*. **503**, 28–35 (2018).

(11) It is odd to use the words “instant” or “immediately” for processes that occur rapidly.

Response: We have replaced “instant” with rapid and “immediately” with rapidly. Thank you.

Optional points to consider

(X) impurity solute (as well as intentional solute) is likely to accumulate in the last liquid to solidify and end up at SGBs. Is this likely to have a significant impact on the analysis.

Response: This is another very useful suggestion for our future research. Thank you. We believe that unintentional impurity solutes are likely to accumulate in the last liquid to solidify, depending on their back-diffusion capabilities. The modified Scheil model by Kurz and Fisher may serve as a basic start point in that regard (need to measure the SDAS first and then to properly determine the cooling rate in order to use the modified Scheil model). We have some experimental data on SGBM in Mg-Sn-Zr and Mg-Pb-Zr (a minor addition of Zr to each binary

alloy) – Zr has noticeably promoted SGBM in both alloys. We plan to systematically investigate this issue in the future. To cover this point and your next optional question, we have added the following lines to our Discussion (also to cover the next optional question):

In addition, in all the above analyses, we have used the average specific GB energy (σ) for a polycrystalline alloy without considering the specific GB structure including possible changes in GB planes during migration. These factors may become influential on SGBM in certain systems. For non-high-purity metals, it should further be noted that impurity solutes are likely to accumulate in the last liquid to solidify along with intentional solute(s) and end up at the SGBs to affect SGBM. The potential influence of the above factors requires clarification through detailed experimental studies in the future.

(X) the authors consider two factors: the change in interface area and the change in specific interfacial energy due to solute. To what extent is the change in specific interfacial energy due to the change in grain boundary plane during migration also important?

Response: We will keep this fine fundamental question in mind for our future study. Grain boundary structure varies with boundary plane, which can then affect GB properties. In this research, we have simply used the average specific GB energy (σ) for a polycrystalline alloy without considering the specific GB structure. Both experimental and theoretical studies have indicated that the specific GB energy is affected by the GB type and structure including GB planes [1, 2]. For the time being, we have added just one sentence to cover this potential scenario – please see our response to the last optional question.

[1] McLean, M. Grain-boundary energy of copper at 1030 °C. *J. Mater. Sci.* **8**, 571-576 (1973).

[2] Tschopp, M. A. & McDowell, D. L. Asymmetric tilt grain boundary structure and energy in copper and aluminium. *Philos. Mag.* **87**, 3871-3892 (2007).

(X) In discussing SGBM, is it important to also consider how and when grain boundaries form at the end of solidification. For example, see: Rappaz, M., Jacot, A., & Boettinger, W. J. (2003). Last-stage solidification of alloys: Theoretical model of dendrite-arm and grain coalescence. *Metallurgical and Materials Transactions A*, 34(3), 467-479.

Response: We greatly appreciate this suggestion. This MMT-A paper laid out the fundamental principles for the formation of SGBs. It is highly relevant. We have emphasized it in our revised Introduction as follows and referred to it elsewhere in our revised manuscript:

In fact, the basic principles for the formation of this interface or solidification grain boundary (SGB) in the last-stage solidification have only recently been delineated by Rappaz et al. [3].

Reviewer #2

Q1. Solute atoms originally position in the GBs have different Gibbs free energy before and after SGBM. In equation (1), the authors did not consider this energy change.

Response: Thank you for this important comment. We have studied in detail the classical thermodynamic treatments of interfaces and GBs in the literature, including their underlying assumptions. In addition, we have had lengthy consultation with Professors Pavel Lejcek and Johan du Plessis (who both started publishing theoretical and experimental studies on GBs from the mid-1980s). We have removed our original Eq. (1) and replaced it with the following formulation, which we hope is fundamentally clear and acceptable.

The Gibbs free energy (G) of a polycrystalline material, without considering its external surface free energy (constant at constant temperature and pressure), can be written as

$$G = G_B + \sigma A \quad (1)$$

where G_B is the Gibbs free energy of all bulk grains, A is the total GB area, and σ is the average specific GB energy. Eq. (1) encompasses the contribution of each component i in both the bulk grains and GBs by noting the definition of σ , i.e., $\sigma A = \sum_{i=1}^m n_i^\phi (\mu_i^\phi - \mu_i)$ [1], where n_i^ϕ is the number of moles of component i in the GBs, μ_i is the chemical potential of i in the bulk grains, and μ_i^ϕ is that in the GBs ($\mu_i \neq \mu_i^\phi$ unless $\sigma = 0$; they are defined differently [1]).

For most metallic alloys, the GB thickness (≤ 1 nm) can be assumed to be infinitesimal vs. grain size (typically $\geq 10^4$ nm, excluding nano-grains). Hence, G_B can be treated as being effectively independent of both n_i^ϕ (GB composition) and A (GB area) [2, 3] or as a constant at constant temperature and pressure. Applying Eq. (1) to SGBM gives

$$\Delta G \approx \sigma_2 A_2 - \sigma_1 A_1 \quad (2)$$

where ΔG is the Gibbs free energy change before (subscript 1) and after (subscript 2) SGBM. For a dilute binary alloy, once the GBs have migrated into the bulk grains, σ_2 can be taken as the solvent specific GB energy σ_0 . Replacing σ_1 with σ and σ_2 with σ_0 and rearranging Eq. (2) gives

$$\frac{\Delta G}{\sigma_0 A_1} \approx - \frac{(A_1 - A_2)}{A_1} + \frac{\sigma_0 - \sigma}{\sigma_0} \quad (3)$$

SGBM requires $\Delta G < 0$, i.e. $(A_1 - A_2)/A_1 > (\sigma_0 - \sigma)/\sigma_0$, where $(A_1 - A_2)/A_1$ may be regarded as the driving factor for SGBM while $(\sigma_0 - \sigma)/\sigma_0$ as the inhibiting factor. Note that the extent of SGBM is further related to the magnitude of ΔG and hence to both σ_0 and A_1 , where σ_0 can vary significantly between different solvent systems (e.g. see Supplementary Table S1).

Thank you for this important comment again, which led to the above understanding.

[1] Lupis, C. H. P. *Chemical thermodynamics of materials*. (Eqs. (57-59) of Chapter XIV) 401 (North-Holland, Amsterdam, 1983).

[2] Plessis, J. D. & Van Wyk, G. N. A model for surface segregation in multicomponent alloys-part I: equilibrium segregation. *J. Phys. Chem. Solids*. **49**, 1441-1450 (1988).

[3] Lejček, P. *Grain Boundary Segregation in Metals*, 55-57 (Heidelberg Univ., Heidelberg, 2010).

Q2 The application of Equation (5) in the deduction is wrong. This equation is originally used for calculation of solute concentration at GBs. It cannot be used to deduce ΔG_{seg} , especially with the description of X_c and X_b in the manuscript. With the description of X_c in Equation 8, and the description of X_b with equations (9) and (10), the ΔG_{seg} value will be dependent of X_0 , nominal composition of alloys. This is not correct, because ΔG_{seg} should be a constant value for a GB in a fixed alloy system. It only changes with the atomic structure of GB.

Response: Thank you for all these important and insightful comments. Together with those from Reviewers #1 and #3, they have profoundly inspired us to develop a conceptually clearer and simpler model as mentioned in the beginning. Consequently, the above three parameters, ΔG_{seg} , X_c and X_b , which were central to our old model, are no longer needed. In the following, we show our new formulation based on the use of the basic Gibbs equation to estimate the specific GB energy (σ) and the Scheil model to describe the last-stage solidification by following the same equation numbering sequence in the revised manuscript.

Formulation of our new model:

The basic Gibbs equation that connects the specific interfacial energy (σ) between two phases α and β with their interfacial composition at constant temperature and pressure is given by [1, 2]

$$\sum_{i=1}^m n_i^{\Phi} d\mu_i + A d\sigma = 0 \quad (4)$$

where n_i^{Φ} means the same as above, i.e., the number of moles of component i in the interface (Φ). Eq. (4) is applicable to both interfaces and GBs [1, 2], providing that the interfacial or GB thickness is infinitesimal vs. the size of each phase or each grain.

For a dilute binary alloy, both the activity coefficient and the activity of the solvent can approximately be taken as unity by adopting the Raoultian standard state (i.e. the pure solvent substance state) [1], while for the solute (X_i) we can write $\mu_i \approx \mu_i^0 + RT \ln X_i$ where μ_i^0 corresponds to the Henrian standard state [1]*. Accordingly, Eq. (4) can be written as

* The use of the Raoultian standard state for the solute, i.e., the pure solute substance state, will just add an extra term $RT \ln \gamma_i^{\infty}$ to μ_i^0 without affecting the formulation, where γ_i^{∞} is the activity coefficient of i at the point of infinite dilution, i.e., $\gamma_i^{\infty} \equiv (\gamma_i)_{X_i \rightarrow 0}$, which is a constant at constant temperature and pressure [1].

$$\frac{d\sigma}{dX_i} \approx - \frac{RT}{X_i} \frac{n_i^\phi}{A} \quad (5)$$

which is effectively the same as Eq. (3) of Ref. [3] used by Liu and Kirchheim in their formulation of σ for solute-segregated non-solidification GBs. The quantity n_i^ϕ/A , often denoted as Γ_i [1], can be expressed through the GB thickness (Δ), GB atomic density (moles/m³) and GB solute composition X_i^ϕ as follows [3, 4]

$$\Gamma_i = \frac{n_i^\phi}{A} = \Delta \rho X_i^\phi \quad (6)$$

The formation of a SGB results from a closing liquid film between two coalescing grains of different orientations in the last-stage solidification [5]. Hence, to a first approximation, the average SGB composition (X_i^ϕ) can be assumed to be that of the last solidifying liquid (X_L) according to the Scheil model below [6]

$$X_L = X_0 (f_L)^{k_0-1} \quad (7)$$

where X_0 is the alloy composition, k_0 is the solute partition coefficient on the phase diagram in molar fraction, and f_L is the volume fraction of the last liquid. Note that the notation X_0 in Eq. (7) means the same as X_i in Eq. (5). Substituting X_L or Eq. (7) as X_i^ϕ into Eq. (6) and then into Eq. (5), we obtain by integrating Eq. (5) from σ_0 to σ for $d\sigma$ and 0 to X_0 for dX_i

$$\int_{\sigma_0}^{\sigma} d\sigma \approx \int_0^{X_0} -RT\Delta\rho (f_L)^{k_0-1} dX_i \quad (8)$$

For simplicity purposes, following Ref. [3], ρ is taken as the solvent atomic density for dilute binary alloys (see Supplementary Section 4 for justification). It follows that

$$\sigma - \sigma_0 \approx -RT\Delta\rho (f_L)^{k_0-1} X_0 \quad (9)$$

Eq. (9) provides an estimate of the specific SGB energy (σ) in a dilute binary alloy. Note that when the Scheil model Eq. (7) is expressed in weight percentage, Eq. (9) changes to a complex equivalent form, see Supplementary Eq. (S3).

Fig. RL4 below (next page) shows our new predictions using Eq. (9). Together with Table 2, the results well explain our experimental observations and those in the literature too.

We have revised our manuscript with the new model described above. Please see the second part of this RL and our revised manuscript.

- [1] Lupis, C. H. Chemical thermodynamics of materials. *Elsevier*, pp. 347-392; pp. 158-183 (1983).
- [2] Gottstein, G. & Shvindlerman, L. S. *Grain boundary migration in metals: thermodynamics, kinetics, applications. 2nd edition*, 12-13 (Equation 1.49) (CRC Press, Taylor & Francis, Boca Raton, 2010).
- [3] Liu, F. and Kirchheim, R. Nano-scale grain growth inhibited by reducing grain boundary energy through solute segregation. *J. Cryst. Growth* **264**, 385-391(2004).

- [4] Lejček, P. *Grain Boundary Segregation in Metals*, 143-144 (Heidelberg Univ., Heidelberg, 2010).
- [5] Rappaz, M., Jacot, A. & Boettinger, W. J. Last-stage solidification of alloys: theoretical model of dendrite-arm and grain coalescence. *Metall. Mater. Trans. A*, **34**, 467-479 (2003).
- [6] Porter, D. A. & Easterling, K. E., *Phase transformations in metals and alloys*, 3rd edition. 211-213 (CRC Press. 2009).

Figure RL4 Calculated values of $(\sigma - \sigma_0)/\sigma_0$ for eleven dilute binary alloy systems using Eq. (9). The selection of $f_L = 0.025$ is based on (i) the approximate minimum width of the Sn-enriched regions in Mg-0.63at.%Sn vs. grain size (see Fig. 2c and 2g), and (ii) the massive SGBM in Mg-0.63at.%Sn (Fig. 2c), despite clear eutectic formation (Fig. 2g and 2f). The selection of $f_L = 0.1$ for Fe-C and Fe-P is due to the use of smaller f_L values leading to excessive C and P (e.g. $X_L = 13.4\text{at.\%C}$ at $f_L = 0.025$ for Fe-0.3at.%C, which means substantial formation of eutectic carbides, unrealistic for this nearly pure Fe composition, Fe-0.065wt.%C).

Q3 Equation (9) is questionable. By insert equation (10) into Eq. (9), one can get $X_b = X_0 \cdot (k+1)/4 + X_0 / 2$. It means, in any case, the concentration of X_b is larger than $X_0 / 2$. For Mg-Sn system, $X_b > 16.6 \text{ at\%}$, which is already larger than the eutectic concentration (10.7 at%) and thus impossible.

Response: Again, thank you so much for the detailed analysis you have done for us. This comment has similarly inspired us to develop the new model presented above. As a result, the original Eq. (9) becomes redundant and has been removed from the manuscript.

Q4. Equation (7) is too rough an estimation about the atomic density at GB. In principle, a substitutional solute atom can also be positioned in the interstitial atomic sites in the GB, if the solute atom is much smaller than the matrix atom, for example Cu in Al alloy.

Response: Thank you for another important comment. We have investigated the selection of GB atomic density (moles/m³) in the literature.

Briefly, the relative atomic density (ρ_{GB}) of a low angle (θ) GB may be estimated using a simple analytical model such as $\rho_{GB} = (1 - \sin(\theta)/4)$ [1]. However, solidification GBs are usually high angle GBs. Priester [2] concluded that for high angle GBs, ρ_{GB} is about 5%-15% less than the bulk grain atomic density. In their treatment of the GBs in the Ni-3.6at.%P alloy, where the GBs contain up to 15at.%P (P is an interstitial segregating element at temperatures of practical importance [3]), Liu and Kirchheim [4], based on tomographic atom probe (TAP) data, used the atomic density of the solvent metal Ni to estimate the GB atomic density.

The use of the Scheil model [5] to estimate the composition of the last solidifying liquid film, i.e. the average SGB composition, has made it easier to determine the SGB atomic density. We have calculated the SGB composition (X_i^ϕ) in dilute binary alloys containing 0.5at.% of the solute. We found that X_i^ϕ is generally less than or about 5at.%, for example, 3.82at.% for Mg-0.5at.%Pb, 5.30at.% for Mg-0.5at.%Al, 4.72at.% for Mg-0.5at.%Sn, 2.81at.%C for Fe-0.5at.%C, and 2.56at.%P for Fe-0.5at.%P. The GB atomic density (ρ_{GB}) means the number of moles of all atoms per unit volume of the GB. Given the above estimates, to keep our final analytical model simple, similar to Ref. [4], we use the solvent atomic density as an approximation of ρ_{GB} in our calculations. This is equivalent to assuming that the volume of 1 mole of the solvent atoms in the liquid state is similar to the volume of 1 mole of the last liquid film that contains less than or about 5at.% of solute atoms at the same temperature.

[1] Kamachali, R. D. A model for grain boundary thermodynamics. *RSC Advances* **10**, 26728-26741 (2020).

[2] Priester, L. *Grain boundaries: from theory to engineering*. 156 (Springer, 2013).

[3] Lejček, P. & Hofmann, S. Interstitial and substitutional solute segregation at individual grain boundaries of α -iron: Data revisited. *J. Phys. Condens. Matter.* **28**, 064001:1-9 (2016).

[4] Liu, F. & Kirchheim, R. Nano-scale growth inhibited by reducing grain boundary energy through solute segregation. *J. Cryst. Growth.* **264**, 385-391 (2004).

[5] Porter, D. A. & Easterling, K. E., *Phase transformations in metals and alloys, 3rd edition*. 211-213 (CRC Press. 2009).

Q5. If the criterion proposed in the work is correct, according to Fig. 6 and Fig. 7, one can expect all binary alloys with extremely low solute content will have significant SGBM, which is not reasonable.

Response: Thank you for another insightful and important comment. We overstated the role of $(\sigma_0 - \sigma)/\sigma_0$ in our original submission and have corrected ourselves in the revised manuscript. As defined by Eq. (3) above (Response to your Q1), the occurrence of SGBM always depends on the combined effects of $(A_1 - A_2)/A_1$ (driving factor) and $(\sigma_0 - \sigma)/\sigma_0$ (inhibiting factor). This basic principle applies to SGBM in high-purity metals too.

Open literature that contains detailed grain-structure information about solidification of high-purity metals is limited. Following an extensive literature search, we have found only three such experimental studies, solidification of high-purity Al (99.99wt.%, by normal casting) [1] and solidification of ultrahigh-purity Cu (99.9999wt.%, undercooled to 40-325 °C below melting point) [2, 3]. Both metals have exhibited noticeable SGBM [1, 2, 3]. This is compatible with our model, because the inhibiting factor $(\sigma_0 - \sigma)/\sigma_0$ of each alloy is smaller than the inhibiting factor of our reference dilute Mg-Sn alloy system according to **Fig. RL4** while their grain morphology allows for a clear reduction in $(A_1 - A_2)/A_1$. Please refer to page 2 of this Response Letter about the calculation of A_1 and A_2 using a truncated octahedron grain model and Table 2 on page 3 about the calculated results using measured grain side length data.

In summary, if the grains in a high-purity or ultrahigh-purity metal show a nearly hexagonal shape with a d/a (d : grain diameter; a : grain side length) ratio close to its equilibrium value (1.732 or $\sqrt{3}$), then no SGBM or negligible SGBM will be expected. In contrast, if the grains are not in a hexagonal shape (the d/a ratio often clearly deviates from $\sqrt{3}$ in this case), then SGBM is expected as the migration is likely to lead to a clear reduction in $(A_1 - A_2)/A_1$, akin to those observed in high-purity Al and ultrahigh-purity Cu [1-3]. Once again, it depends on the combined effects of $(A_1 - A_2)/A_1$ and $(\sigma_0 - \sigma)/\sigma_0$.

To properly respond to this important comment, we have added the following paragraph to our Discussion together with our response to two optional questions from Reviewer #1):

The coordinate origin in Fig. 7 corresponds to the condition of $(\sigma_0 - \sigma)/\sigma_0 = 0$. It should be stressed that this does not mean that SGBM will occur in all high-purity metals. Literature containing detailed grain-structure information about solidification of high-purity metals is limited. However, both high-purity Al (99.99wt.% [1]) and ultrahigh-purity Cu (99.9999wt.% [2, 3]) have exhibited noticeable SGBM. Akin to the case of $(\sigma_0 - \sigma)/\sigma_0 > 0$, the basic principle is the same by Eq. (3) for SGBM in the case of $(\sigma_0 - \sigma)/\sigma_0 = 0$, i.e. $\Delta G < 0$ while the extent of SGBM is further related to the magnitude of ΔG . For example, if the grains in a high-purity metal show nearly hexagonal straight GBs with d/a close to

1.732, then no SGBM will be expected. In addition, in all the above analyses, we have used the average specific GB energy (σ) for a polycrystalline alloy without considering the specific GB structure including possible changes in GB planes during migration. These factors may become influential on SGBM in certain systems. For non-high-purity metals, it should further be noted that impurity solutes are likely to accumulate in the last liquid to solidify along with intentional solute(s) and end up at the SGBs to affect SGBM. The potential influence of the above factors requires clarification through detailed experimental studies in the future.

- [1] Biloni, H. Substructures and dislocations produced during solidification of aluminium. *Can. J. Phys.* (Revue canadienne de physique) **39**, 1501-1507 (1961).
- [2] Kalin, I. D., Robert, F. C. & Andrew, M. M. The mechanism for spontaneous grain refinement in undercooled pure Cu melts. *Mater. Sci. Eng. A* **375-377**, 479-484 (2004).
- [3] Battersby, S. E., Cochrane, R. F. & Mullis, A. M. Microstructural evolution and growth velocity-undercooling relationships in the systems Cu, Cu-O and Cu-Sn at high undercooling. *J. Mater. Sci.* **35**, 1365-1373 (2000).

Reviewer #3 (Remarks to the Author):

This is an interesting paper about a topic that is not widely described in the open literature and occasionally speculated upon. Thus it is a valuable contribution. The phenomenon termed SGBM in this work is however not often observed in cast alloys, as these alloys commonly form secondary phases that pin grain boundaries. This aspect has not even been mentioned in the manuscript.

Response: Thank you for these comments. SGBM is indeed not often observed in commercial cast alloys due to particle pinning effects. We have added the following sentences to the Introduction in our revised manuscript, together with a note on mobile GB particles discussed in the literature:

In general, SGBM is not often observed in commercial cast alloys due to particle pinning effects (note that GB particles can migrate with moving GBs, known as ‘mobile particles’ [1-5], especially at elevated temperatures, depending on their properties, sizes, and interactions with GBs).

[1] Gottstein, G. & Shvindlerman, L. S. Theory of grain boundary motion in the presence of mobile particles. *Acta Metall. Mater.* **41**, 3267-3275 (1993).

[2] Hassold, G. N. & Srolovitz, D. J. Computer simulation of grain growth with mobile particles, *Scripta Metall. Mater.* **32**, 1541-1547 (1995).

[3] Novikov, V. Y. Grain growth jointly affected by immobile and mobile particles. *Mater. Lett.* **178**, 276-279 (2016).

[4] Novikov, V. Y. On grain growth in the presence of mobile particles. *Acta mater.* **58**, 3326-3331 (2010).

[5] Gottstein, G. & Shvindlerman, L. S. *Grain boundary migration in metals: thermodynamics, kinetics, applications. 2nd edition*, 168-170 (CRC Press, Taylor & Francis Group, Boca Raton, 2010).

Generally I find the approach and especially the experimental work appealing and interesting. I like the fact that alloys were produced to test the model and hypothesis.

Overall, I fail to see from the manuscript the authors argument why this paper is relevant to an audience beyond the metallurgical and materials community.

Response: Thank you for your appreciation. We hope we have understood your second point (‘Overall ...’) correctly. Just a minor observation from us in that regard: *Nature Communications (NC)* published 5,835 papers in 2019 and 5661 in 2018 according to Scopus (only a few prestigious journals publish more than 3,000 papers p.a.). It seems common that papers published in *NC* are highly specialized, irrespective of the research field of the paper. It publishes a good number of interesting papers on metallurgy and metallic materials each year.

There are aspects of the model however that I do not agree with.

- I would like to see a more rigorous definition of the driving force for GB segregation. My understanding is that this is a driving force that drives atoms of alloying elements from the bulk to the grain boundary (GB). That is for a stationary GB. It is also accounted for in theories of solute drag. I therefore do not agree with the way this is used in the present context.

Response: *We must thank you for this critical comment, which has directly catalyzed our new model.* Indeed, SGB segregation differs from conventional GB segregation that occurs in the solid state from the bulk to the GB. The driving force is not the same. The new model we have developed is **no longer** related to solid-state GB segregation. It is formulated from using just the basic Gibbs equation to define the specific GB energy and the Scheil model to describe the last-stage solidification – simpler and clearer. We present our formulation below.

Formulation of our new model (following the same equation numbering sequence in the revised manuscript)

The basic Gibbs equation that connects the specific interfacial energy (σ) between two phases α and β with their interfacial composition at constant temperature and pressure is given by [1, 2]

$$\sum_{i=1}^m n_i^\phi d\mu_i + A d\sigma = 0 \quad (4)$$

where n_i^ϕ means the same as above, i.e., the number of moles of component i in the interface (ϕ). Eq. (4) is applicable to both interfaces and GBs [1, 2], providing that the interfacial or GB thickness is infinitesimal vs. the size of each phase or each grain.

For a dilute binary alloy, both the activity coefficient and the activity of the solvent can approximately be taken as unity by adopting the Raoultian standard state (i.e. the pure solvent substance state) [1], while for the solute (X_i) we can write $\mu_i \approx \mu_i^0 + RT \ln X_i$ where μ_i^0 corresponds to the Henrian standard state [1]*. Accordingly, Eq. (4) can be written as

$$\frac{d\sigma}{dX_i} \approx - \frac{RT}{X_i} \frac{n_i^\phi}{A} \quad (5)$$

which is effectively the same as Eq. (3) of Ref. [3] used by Liu and Kirchheim in their formulation of σ for solute-segregated non-solidification GBs. The quantity n_i^ϕ/A , often denoted as Γ_i [1], can be expressed through the GB thickness (Δ), GB atomic density (moles/m³) and GB solute composition X_i^ϕ as follows [3, 4]

$$\Gamma_i = \frac{n_i^\phi}{A} = \Delta \rho X_i^\phi \quad (6)$$

* The use of the Raoultian standard state for the solute, i.e., the pure solute substance state, will just add an extra term $RT \ln \gamma_i^\infty$ to μ_i^0 without affecting the formulation, where γ_i^∞ is the activity coefficient of i at the 23 point of infinite dilution, i.e., $\gamma_i^\infty \equiv (\gamma_i)_{X_i \rightarrow 0}$, which is a constant at constant temperature and pressure [1].

The formation of a SGB results from a closing liquid film between two coalescing grains of different orientations in the last-stage solidification [5]. Hence, to a first approximation, the average SGB composition (X_i^ϕ) can be assumed to be that of the last solidifying liquid (X_L) according to the Scheil model below [6]

$$X_L = X_0(f_L)^{k_0-1} \quad (7)$$

where X_0 is the alloy composition, k_0 is the solute partition coefficient on the phase diagram in molar fraction, and f_L is the volume fraction of the last liquid. Note that the notation X_0 in Eq. (7) means the same as X_i in Eq. (5). Substituting X_L or Eq. (7) as X_i^ϕ into Eq. (6) and then into Eq. (5), we obtain by integrating Eq. (5) from σ_0 to σ for $d\sigma$ and 0 to X_0 for dX_i

$$\int_{\sigma_0}^{\sigma} d\sigma \approx \int_0^{X_0} -RT\Delta\rho (f_L)^{k_0-1} dX_i \quad (8)$$

For simplicity purposes, following Ref. [3], ρ is taken as the solvent atomic density for dilute binary alloys (see Supplementary Section 4 for justification). It follows that

$$\sigma - \sigma_0 \approx -RT\Delta\rho(f_L)^{k_0-1}X_0 \quad (9)$$

Eq. (9) provides an estimate of the specific SGB energy (σ) in a dilute binary alloy. Note that when the Scheil model Eq. (7) is expressed in weight percentage, Eq. (9) changes to a complex equivalent form, see Supplementary Eq. (S3).

Fig. RL4 (next page) shows our new predictions using Eq. (9). Together with Table 2, the results can well explain our experimental observations and those in the literature too.

We have revised our manuscript with the new model described above. Please see the second part of this Response Letter and our revised manuscript.

- [1] Lupis, C. H. Chemical thermodynamics of materials. *Elsevier*, pp. 347-392, pp. 158-182 (1983).
- [2] Gottstein, G. & Shvindlerman, L. S. *Grain boundary migration in metals: thermodynamics, kinetics, applications. 2nd edition*, 12-13 (Equation 1.49) (CRC Press, Taylor & Francis, Boca Raton, 2010).
- [3] Liu, F. and Kirchheim, R. Nano-scale grain growth inhibited by reducing grain boundary energy through solute segregation. *J. Cryst. Growth* **264**, 385-391(2004).
- [4] Lejček, P. *Grain Boundary Segregation in Metals*, 143-144 (Heidelberg Univ., Heidelberg, 2010).
- [5] Rappaz, M., Jacot, A. & Boettinger, W. J. Last-stage solidification of alloys: theoretical model of dendrite-arm and grain coalescence. *Metall. Mater. Trans. A* **34**, 467-479 (2003).
- [6] Porter, D. A. & Easterling, K. E., *Phase transformations in metals and alloys, 3rd edition*. 211-213 (CRC Press. 2009).

Figure RL4 Calculated values of $(\sigma - \sigma_0)/\sigma_0$ for eleven dilute binary alloy systems using Eq. (9). The selection of $f_L = 0.025$ is based on (i) the approximate minimum width of the Sn-enriched regions in Mg-0.63at.%Sn vs. grain size (see Fig. 2c and 2g), and (ii) the massive SGBM in Mg-0.63at.%Sn (Fig. 2c), despite clear eutectic formation (Fig. 2g and 2f). The selection of $f_L = 0.1$ for Fe-C and Fe-P is due to the use of smaller f_L values leading to excessive C and P (e.g. $X_L = 13.4\text{at.\%C}$ at $f_L = 0.025$ for Fe-0.3at.%C, which means substantial formation of eutectic carbides, unrealistic for this nearly pure Fe composition, Fe-0.065wt.%C).

- The statements in line 262 - 265 seems to contradict themselves. If Mg-Pb and Mg-Sn alloys show similar resistance, why are Mg-Pb alloys more prone to SGBM? Maybe this is just a problem for formulation and the sentence just needs some re-phrasing.

Response: Thank you. We have re-phrased our original description to avoid such confusion. In addition, we have quantified the two factors in the right-hand side of Eq. (3) below, which was our original Eq. (12), to allow for a direct comparison

$$\frac{\Delta G}{\sigma_0 A_1} \approx - \frac{A_1 - A_2}{A_1} + \frac{\sigma_0 - \sigma}{\sigma_0} \quad (3)$$

where A_1 and A_2 are the GB areas before and after SGBM, respectively, σ is the specific GB energy before SGBM (i.e. σ_1), and σ_0 is the solvent specific GB energy, equivalent to σ_2 in a dilute binary alloy. To evaluate $(A_1 - A_2)/A_1$, to a first approximation, we treat each equiaxed grain as a truncated octahedron (see **Fig. RL1** next page), the only Archimedean solid that can offer cavity-free full space filling (Torquato et al., "Dense packings of polyhedra:

Platonic and Archimedean solids”, Physical Review E, 2009, 80, 041104). The surface area of a truncated octahedron (A), which consists of eight hexagonal planes and six squares of the same side length (a), is equal to $(6 + 12\sqrt{3})a^2$, and its volume (V) is given by $8\sqrt{2}a^3$ [1].

Table 2 lists the measured grain side length (a), grain size (d), ratio of d/a , and calculated values of $(A_1-A_2)/A_1$ and $(\sigma_0 - \sigma)/\sigma_0$. SGBM requires $\Delta G < 0$, i.e. $(A_1-A_2)/A_1 > (\sigma_0 - \sigma)/\sigma_0$. The results in **Table 2** indicate that Mg-0.3at.%Pb, Mg-0.3at.%Sn, and Mg-0.21at.%Sn have similar values of $[(A_1-A_2)/A_1 - (\sigma_0 - \sigma)/\sigma_0]$ in the range of 0.28-0.33. They have all displayed significant SGBM. Overall, the results can well explain our observations in each alloy.

Total surface: $A = (6 + 12\sqrt{3})a^2$
 Total volume: $V = 8\sqrt{2}a^3$

[1] Robert Williams (1979). *The Geometrical Foundation of Natural Structure: A Source Book of Design*, page 78, New York, Dover Publications, Inc.

Fig. RL1 A truncated octahedron with side length a (see also <https://mathworld.wolfram.com/ArchimedeanSolid.html>)

Table 2 Grain side length (a), grain size (d), driving factor $(A_1-A_2)/A_1$ and inhibiting factor $(\sigma_0-\sigma)/\sigma_0$ in dilute binary Mg-X alloys ($X = \text{Sn, Pb, Al, Zn}$)

Binary Mg-X alloys	Average grain side length ^{1*} (μm)		Average grain size (d , μm)	d/a_1	d/a_2	Driving factor $(A_1-A_2)/A_1$ (%) ^{2*}	Inhibiting factor $(\sigma_0-\sigma)/\sigma_0$ (%) ^{3*}
	Before SGBM (a_1)	After SGBM (a_2)					
0.21at.%Sn	170.7 ± 31.3	137.2 ± 26.8	241.5 ± 47.2	1.41 ± 0.15	1.76 ± 0.11	35.41	2.17
0.42at.%Sn	130.2 ± 25.1	111.3 ± 22.7	190.3 ± 38.8	1.46 ± 0.13	1.71 ± 0.12	26.90	4.48
0.63at.%Sn	94.0 ± 23.6	84.2 ± 17.9	144.8 ± 28.0	1.54 ± 0.14	1.72 ± 0.11	19.76	6.95
1.07at.%Sn	69.3 ± 10.2	63.2 ± 8.1	108.1 ± 18.3	1.56 ± 0.12	1.71 ± 0.09	16.83	12.80
0.30at.%Zn	83.1 ± 16.8	83.1 ± 16.8	131.3 ± 25.2	1.58 ± 0.13	1.58 ± 0.13	0 ^{4*}	15.71
0.30at.%Al	102.3 ± 20.3	99.4 ± 15.2	170.0 ± 28.3	1.66 ± 0.12	1.71 ± 0.08	5.59	4.26
0.30at.%Sn	150.3 ± 30.1	124.1 ± 22.3	213.4 ± 40.1	1.42 ± 0.14	1.72 ± 0.10	31.82	3.77
0.30at.%Pb	155.1 ± 37.9	127.5 ± 29.4	221.8 ± 41.5	1.43 ± 0.13	1.74 ± 0.10	32.42	2.39

1*: The average grain side length was calculated by dividing the measured average grain circumference by 6.

2*: The surface area (A) of a truncated octahedron is equal to $(6 + 12\sqrt{3})a^2$.

3*: $\sigma_0 = 0.52 \text{ J/m}^2$ for pure Mg from 640°C to 650°C (Supplementary Table S1).

4*: Although $(A_1-A_2)/A_1 = 0$ due to the absence of migration, the driving force for migration still exists because of their low average value of d/a (1.58, less than 1.732) and non-hexagonal shapes.

- line 273 ff: Well, it seems like some alloys do not show any SGBM at all, but I have my doubts on the author's arguments as to why that is the case. The difference in driving force reduction is not that massive but we still see a massive step change in behaviour! I suspect pinning effects.

Response: This is another important suggestion, which has led to the addition of some important new details to our manuscript. Having re-examined each alloy, we can confirm that particle pinning effects have indeed occurred, e.g. in Mg-0.63at.%Sn alloy shown by **the red arrows** in **Fig. RL5a** below. However, it is also clear from **Fig. RL5a** that the two GB segments that are $\sim 20\text{-}30\ \mu\text{m}$ away from the Mg_2Sn pinning particles ($3\text{-}8\ \mu\text{m}$ eutectic Mg_2Sn) still exhibited distinct migration (see blue arrows in **Fig. RL5a**). Similarly, we have observed clear SGBM in the Mg-1.07at.%Sn alloy (see Fig. 3f in the manuscript) in which noticeable eutectic phases have formed (see **Fig. RL5b**). This suggests that it may require a prominent presence of SGB particles to completely suppress SGBM. This hypothesis is supported by the observations in the Mg-1.52at.%Sn (7wt.%Sn) alloy in which the widespread eutectic phases (Fig. RL5 c and d) led to only minor SGBM (see Fig. 3h in the manuscript).

Fig. RL5 (a) Particle pinning in Mg-0.63at.%Sn (**red arrows**) with significant SGBM (**blue arrows**). (b) Eutectic $\alpha\text{-Mg}(\text{Sn})+\text{Mg}_2\text{Sn}$ in Mg-0.63at.%Sn, which still exhibited clear SGBM (see Fig. 3f – manuscript). (c) and (d) Widespread eutectic $\alpha\text{-Mg}(\text{Sn})+\text{Mg}_2\text{Sn}$ in Mg-1.52at.%Sn (7wt.%Sn), which exhibited only local SGBM (see Fig. 3h – manuscript).

We have also examined the eutectic presence in binary Mg-0.3at.%X (X = Sn, Al, Zn, Pb) alloys. It is barely detectable in Mg-0.3at.%Al, Mg-0.3at.%Zn and Mg-0.3at.%Pb alloys due to their low solute content and high eutectic composition (31at.%Al for Mg-Al; 30at.%Zn for Mg-Zn; 19.1at.%Pb for Mg-Pb; 10.8at.%Sn for Mg-Sn). This allows us to conclude that the lack of SGBM in Mg-0.3at.%Zn is not due to particle pinning. Rather, it is due to its high SGBM-inhibiting factor (Table 2, the highest of all the alloys studied). In contrast, the reason for Mg-0.3at.%Al exhibiting just localized SGBM is due to its low SGBM-driving factor related to its grain shape (nearly hexagonal, $d/a = 1.66$, close to 1.732, Table 2).

Owing to this suggestion, we have added the following paragraph to cover the particle pinning issue in our revised manuscript (thank you again – a nice suggestion):

The formation of eutectic phases in a dilute alloy renders the possibility of particle pinning. Indeed, this was observed in the Mg-0.63at.%Sn alloy (see red arrows in Fig. 2g, Mg₂Sn particles). However, it is also clear from Fig. 2g that the two GB segments that are ~20-30 μm away from the Mg₂Sn pinning particles still exhibited distinct migration (see blue arrows in Fig. 2g). This implies that it may require a prominent presence of secondary phase particles to completely suppress SGBM. This inference is supported by the observation in the Mg-1.52at.%Sn (7wt.%Sn) alloy, which had a marked presence of eutectic α-Mg+Mg₂Sn (Supplementary Fig. S2) and therefore exhibited only localised SGBM (Fig. 3 h). As mentioned earlier, particle pinning effects are related to particle properties, sizes, their interactions with GBs, and temperature [3, 4].

- [1] Kurz, W. & Fisher, D. J. *Fundamentals of solidification*. 127-134 (Trans Tech Pub. 1989)..
- [2] Dahle, A. K., Lee, Y. C., Nave, M. D., Schaffer, P. L. & StJohn, D. H. Development of the as-cast microstructure in magnesium–aluminium alloys. *Journal of light metals*, **1**, 61-72 (2001).
- [3] Gottstein, G. & Shvindlerman, L. S. Theory of grain boundary motion in the presence of mobile particles. *Acta Metall. Mater.* **41**, 3267-3275 (1993).
- [4] Hassold, G. N. & Srolovitz, D. J. Computer simulation of grain growth with mobile particles, *Scripta Metall. Mater* **32**, 1541-1547 (1995).

- line 318 ff: I am not convinced by the diffusion argument developed here. Diffusion leading to GB motion is not the same as solute diffusion over long distances, but involves a large number of small hops, over low activation energy barriers (due to loose structure in GB) covering short distances of about 1 atomic spacing. The estimate in the manuscript suggests grain boundaries behave like atoms and diffuse by vacancy exchange mechanisms. This would of course be quite slow.

Response: We totally agree with you. Having studied the literature on low-velocity and high-velocity GBM, we have removed that entire paragraph on the diffusion argument based on

vacancy exchange mechanisms. Furthermore, *Nature Materials* has just published online an exciting paper entitled “*Direct imaging of atomistic grain boundary migration*” (2021, DOI: [10.1038/s41563-020-00879-z](https://doi.org/10.1038/s41563-020-00879-z)), which supports the long-assumed cooperative shuffling mechanism for GBM. On this basis, we have added the following lines **in blue** to provide a possible explanation of the rapid SGBM observed in this study:

Kinetics of SGBM. As noted from Fig. 4 and Table 1, SGBM does not depend on cooling time over the range of cooling rate from 8 °C/s to 1690 °C/s. In general, this implies that there was “no need to wait for sufficient statistical fluctuations in some specific order parameter to overcome an activation barrier to initiate the process” [1]. In other words, it tends to fall in the regime of athermal motion of GBs, which can traverse at the speed of sound [2], compatible with the rapid SGBM observed at the cooling rate of 1690 °C/s in this work.

While the exact atomistic mechanisms for rapid GBM may differ to a certain extent between different systems, a basic hypothesis accepted by researchers is that it occurs by a group mechanism, through cooperative motion or local conservative shuffling of atoms across the boundary [3-7], rather than by diffusive jumps of single atoms. For example, according to Kopetsky et al. [3], GBs can move “by means of cooperative displacements of lattice atoms, by adjustment to the lattice of a growing grain; each atom is then shifted by a distance shorter than the lattice parameter”. A recent breakthrough in direct imaging of atomistic grain boundary migration has provided direct support to this hypothesis [8], which revealed that “the GB migration proceeds by the cooperative shuffling of atoms on GB ledges along specific routes”. Although no specific criteria or assessment parameters are yet available for this mechanism, conceptually, it provides a plausible mechanism for understanding the rapid SGBM observed in this work. Substantial efforts are expected to experimentally validate this mechanism for SGBM due to its rapid migration nature.

[1] Laughlin, D.E., Jones, N.J., Schwartz, A.J. and Massalski, T.B., Thermally activated martensite: its relationship to non-thermally activated (athermal) martensite, in *International Conference on Martensitic Transformations (ICOMAT)*, editor(s): Olson, G. B., Lieberman, D. S. & Saxena, A. 141-144 (The Minerals, Metals and Materials (2010). DOI:10.1002/9781118803592).

[2] Gottstein, G. & Shvindlerman, L. S. *Grain boundary migration in metals: thermodynamics, kinetics, applications. 2nd edition*, 118-129 (CRC Press, Taylor & Francis, Boca Raton, 2010).

[3] Kopetsky, Ch. V., Shvindlerman, L. S. & Sursayeva, V. G. Effect of athermal motion of grain boundaries. *Scr. Mater.* **12**, 953-956 (1978).

[4] Jhan, R. J. & Bristowe, P. D. A molecular dynamics study of grain boundary migration without the participation of secondary grain boundary dislocations. *Scripta Metall.* **24**, 1313-18 (1990).

- [5] Schönfelder, B., Wolf, D., Phillpot, S.R. & Furtkamp, M. Molecular-dynamics method for the simulation of grain-boundary migration. *Interface Sci.* **5**, 245–262(1997)
- [6] Babcock, S. E. & Balluffi, R. W. Grain boundary kinetics—II. In situ observations of the role of grain boundary dislocations in high-angle boundary migration. *Acta Metall.* **37**, 2367-76 (1989).
- [7] Gutkin, M. Y., Mikaelyan, K. N. & Ovid'ko, I. A. Athermal grain growth through cooperative migration of grain boundaries in deformed nanomaterials. *Scr. Mater.* **58**, 850-853 (2008).
- [8] Wei, J., Feng, B., Ishikawa, R., Yokoi, T., Matsunaga, K., Shibata, N. & Ikuhara, Y. Direct imaging of atomistic grain boundary migration. *Nature Materials*, 2021 (DOI: 10.1038/s41563-020-00879-z).

Detailed changes to the manuscript

Abstract

Lines 13-14: we have added (blue) “The GBs that form between coalescing grains in the last-stage solidification, referred to as solidification GBs (SGBs), can migrate ...”

Lines 19-20: we have added “SGBM does not depend on cooling rate (8-1690 °C/s) but is related to grain size.”

Introduction

Lines 30-32: we have added “The basic principles for the formation of this interface or solidification grain boundary (SGB) in the last-stage solidification have only recently been delineated by Rappaz et al. [3].”

Lines 46 -50: we have added “... and concentrated alloys such as high entropy alloys (HEAs) [27]. In general, SGBM is not often observed in as-cast commercial alloys due to particle pinning effects (note that GB particles can migrate along with moving GBs as ‘mobile particles’ [28-32], especially at elevated temperatures, depending on their properties, sizes, and interactions with GBs).

Lines 51 -53: owing to a copyright issue, we have replaced our original Fig. 1 with the newly drawn Fig. 1 by referring to the original literature (i.e. Ref. [1]).

Fig.1 Schematic correspondence between dendrite cells and solidification grain boundaries (SGBs). (a) Typical correspondence, based on Ref. [1]. (b) Decoupled SGBs.

Lines 79-81: we have added “using the basic Gibbs equation to estimate the specific SGB energy and the Scheil equation to describe the last-stage solidification.”

Results

Lines 96-111: we have amended our original Fig. 2 based on comments and suggestions from Reviewers # 1 and 3. Figs. 2c-2h are new results that have replaced the original Figs. 2c-2f.

Fig. 2 SGBM in the Mg-0.63at.%Sn (3wt.%) alloy. **(a, b)** Optical micrographs showing massive SGBM **(a)** and local SGBM **(b)**. **(c)** FSD image showing migrated SGBs (arrows 1, 2, 3) and Sn-enriched interdentritic regions (bright). **(d)** EBSD IPF map of **(c)**, in which the dark regions are zero-solution regions, i.e., they cannot be indexed as Mg by the AZtec software due to Sn enrichment. **(e, f)** Energy dispersive spectroscopy (EDS) mapping of Sn and Mg of **(c)**. **(g)** Backscattered electron (BSE) image showing migrated SGBs (blue arrows), SGB particles (red arrows), Sn-enriched interdentritic regions and eutectic α -Mg+Mg₂Sn **(h)**.

Lines 112-126: we have amended the following description for the content of Fig. 2:

Fig. 2c shows a forescatter detector (FSD) image of a selected region with migrated SGBs in the Mg-0.63at.%Sn alloy. The network-like bright regions are due to atomic number contrast and confirmed to be Sn-enriched but Mg-depleted according to Fig. 2(e, f). They correspond to the dark dendrite cell regions in Fig. 2a and contain ~5-15 wt.%Sn (Supplementary Fig. S1), compatible with the solubility of Sn in α -Mg (15 wt.%) at the eutectic temperature. Fig. 2d is an electron backscatter diffraction (EBSD) inverse pole figure (IPF) map of Fig. 2c. The misorientation angle/axis pairs are determined to be $32.5^\circ/\langle 5\bar{2}70 \rangle$ between the blue and purple grains, $32.9^\circ/\langle 5\bar{4}16 \rangle$ between the blue and green grains, and $35.5^\circ/\langle 2\bar{7}56 \rangle$ between the purple and green grains. The GB triple junction (130° - 115° - 115°) is close to equilibrium (120° - 120° - 120°) [73], indicating its stable state after SGBM. Another observation is that limited eutectic α -Mg(Sn)+Mg₂Sn phases formed in certain of the interdendritic regions as exemplified in Fig. 2 (g, h), predictable by the Scheil model (a minimum of 3.35at.%Sn is, however, required by equilibrium solidification). Their presence increased with increasing Sn content beyond 0.63at.%Sn (Supplementary Fig. S2) but is negligible in Mg-0.21at.%Sn and Mg-0.3at.%Sn due to their low solute (Sn) content and the likely back-diffusion in the last stage [3].

Lines 146-155: we have added the following paragraph to interpret the particle pinning effect.

The formation of eutectic phases in a dilute alloy renders the possibility of particle pinning effects. Indeed, this was observed in the Mg-0.63at.%Sn alloy (see red arrows in Fig. 2g, Mg₂Sn particles). However, it is also clear from Fig. 2g that the two SGB segments that are ~20-30 μ m away from the Mg₂Sn pinning particles still exhibited distinct migration (see blue arrows in Fig. 2g). This implies that it may require a prominent presence of secondary phase particles to completely suppress SGBM. This inference is supported by the observation in the Mg-1.52at.%Sn (7wt.%Sn) alloy, which had a marked presence of eutectic α -Mg+Mg₂Sn (Supplementary Fig. S2) and therefore exhibited only local SGBM (Fig. 3 (g, h)). As mentioned earlier, particle pinning effects are related to particle properties, sizes, interactions with GBs, and temperature [28, 29].

Lines 156-158: we have changed the original subtitle (**The effect of cooling rate on SGBM**) to the following, inspired by a comment from Reviewer #1:

The effect of grain size on SGBM through changing cooling rate. Grain size is likely to affect SGBM. This influence was investigated through changing cooling rate from ~ 8 °C/s to ~ 1690 °C/s without introducing grain refiners which change alloy composition.

Lines 166-171: we have added the following six lines on the effect of grain size on SGBM.

The maximum SGBM distance (Δ_{\max}) corresponding to each cooling rate was measured along with grain size (d). The results are included in Table 1. The ratio of Δ_{\max}/d varies from 30% to 40%, which is essentially independent of the range of cooling rate studied (8-1690 °C/s). This observation is in line with the influence of grain size on migration of non-solidification GBs based on limited experimental data [77]. No change in grain number was observed after SGBM at each cooling rate.

Line 177/178 (Table 1): Table 1 below has been supplemented with new experimental results about the influence of grain size on SGBM.

Table 1 Secondary dendrite arm spacing (SDAS, λ_2), cooling rate (\dot{T}), average grain size (d), maximum SGBM distance (Δ_{\max}) and ratio of Δ_{\max}/d for dilute Mg-Sn alloys					
Alloy	SDAS (λ_2 , μm)	Cooling rate (\dot{T} , °C/s)	Average grain size (d , μm)	Maximum GBM (Δ_{\max} , μm)	Δ_{\max}/d (%)
Mg-0.21at.%Sn	42.5 ± 17.2	~ 8	241.5 ± 47.2	86.8	35.9
Mg-0.30at.%Sn	42.7 ± 16.5	~ 8	213.4 ± 40.1	71.2	33.4
Mg-0.42at.%Sn	41.8 ± 15.6	~ 8	190.3 ± 38.8	60.6	31.8
Mg-1.07at.%Sn	41.5 ± 14.5	~ 8	108.1 ± 18.3	31.2	28.9
Mg-0.63at.%Sn	41.6 ± 14.8	~ 8	144.8 ± 28.0	45.8	31.6
Mg-0.63at.%Sn	18.2 ± 5.4	~ 54	73.3 ± 13.1	23.1	31.5
Mg-0.63at.%Sn	11.1 ± 3.7	~ 166	61.9 ± 10.6	18.8	30.4
Mg-0.63at.%Sn	4.0 ± 1.4	~ 1690	40.8 ± 6.1	12.1	29.7

Discussion

The entire Discussion has been re-written based on all the comments and suggestions received from each respected reviewer. The formulation of our new model, which constitutes the key part of our Discussion, has been presented above in response to respective comments. For the entire new Discussion, please refer to Lines 195-355 or Pages 11-18 in our revised manuscript.

Other changes

1: Methods – we have added the new EBSD characterisation conditions in detail.

2: We have included a **Supplementary Information** section to provide necessary support to a few statements made in the revised text.

REVIEWER COMMENTS

Reviewer #1 (Remarks to the Author):

My comments have been addressed satisfactorily.

Reviewer #2 (Remarks to the Author):

The present reviewer has noticed that the authors have done a thorough revision of the manuscript, to address the comments and concerns of the reviewers. The quality of the revised manuscript has been significantly improved. However, as a pity, the theoretical model for SGBM is still not convincing. The major problem is that the authors are trying to quantify the driving force of SGBM by using too rough an energy barrier model based on too much simplification and by using arbitrarily selected parameters. The specific problems are as follows,

1) In equation (9), the authors did not consider the segregation energy of solutes at grain boundary. This is because they simply assume that $X_i(\phi)$ is equal to X_L in eq. (6). The authors should be aware that different solute elements have different GB segregation energy and therefore different concentrations at GB even though they have the same matrix concentration. So, it is not proper to use eq. (9) as a general equation to evaluate the SGBM possibility of different solute atoms and different alloys. As for the GB segregation energy term, the authors can compare their equation with eq. (6) of reference [82].
2) It is not proper to use X_L to substitute $X_i(\phi)$ in Eq. (6), because the former can reach an infinite value at small enough f_L value according to Scheil equation. For the normal solidification condition, the X_L value cannot exceed the eutectic concentration at the end of solidification. Otherwise, eutectic reaction will occur.

3) The energy barrier factor $(\sigma_0 - \sigma) / \sigma_0$ in Eq. (9) is very much dependent of the f_L value. In table 2 and Fig. 7, the calculated $(\sigma_0 - \sigma) / \sigma_0$ values are based on $f_L = 0,025$. First, this is a very arbitrary value. I don't see the reason why the grain boundaries have to form at this liquid fraction, instead of other fractions. Secondly, one can calculate that the X_L value of Mg-1.07at%Sn alloy (at $f_L = 0,025$, $k=0,39$) is about 47 wt% Sn. If we select $f_L = 0,01$, the X_L value will become 82.2 wt.%, then the inhibiting (energy barrier) factor term $(\sigma_0 - \sigma) / \sigma_0$ will be much higher than that shown in Table 2. As a result, there may not be enough driving force for SGBM. It has to be mentioned, with either f_L values, the X_L value is much higher than the eutectic point (36.9%Sn) and thus not possible. It is impossible to have so high a concentration of Sn in liquid without forming Mg_2Sn particles. Lastly, the replacement of $X_i(\phi)$ with X_L , and the f_L values used by the authors have significantly overestimated the energy barrier.

4) The approach used by the authors to determine the energy driving factor of SGBM is not reasonable. The calculated values of $(A_1 - A_2) / A_1$ shown in Table 2 are based on the GBs after completion of SGBM. If we consider the real process of SGBM, the $(A_1 - A_2) / A_1$ should be very small at the beginning of SGBM, thus $(A_1 - A_2) / A_1 \ll (\sigma_0 - \sigma) / \sigma_0$. In this case, SGBM can not happen, because there is not enough driving force. This is fatal to the theoretical model proposed in the manuscript.

Since SGBM is a phenomenon known for decades, without a proper theoretical interpretation the novelty of this manuscript is much reduced. The manuscript cannot be accepted in the present form as it

is.

**

I have read through the response and the original comments by Reviewer 3. My evaluation is as follows, Comment 1. The authors have well addressed the concern by the reviewer by adding one sentence in the introduction part.

Comment 2. About “if this paper is relevant to an audience beyond the metallurgical and materials community”, I, personally, think the authors’ reply is proper.

Comment 3. The original comment by the reviewer is “I would like to see a more rigorous definition of the driving force for GB segregation. My understanding is that this is a driving force that drives atoms of alloying elements from the bulk to the grain boundary (GB). That is for a stationary GB. It is also accounted for in theories of solute drag. I therefore do not agree with the way this is used in the present context”.

My judgement is that the authors have tried to amend their theory part by introducing a revised model. However, the driving force for GB migration is still not convincing. This is because the total energy difference of GBs before and after migration is dependent of the total GB area change. However, at the beginning of GB migration, $(A_1 - A_2)/A_1$ is zero and thus less than $(\sigma_0 - \sigma)/\sigma_0$. Therefore, SGBM will never happen.

Comment 4. Refer to comment 3.

Comment 5. The response by the authors has properly addressed the pinning effect by GB particles in the revised manuscript based on the comment by the reviewer.

Comment 6. About the diffusion mechanism which controls the GB migration, the authors have revised the manuscript and discussed the mechanism in a proper way.

Response Letter

We absolutely appreciate the further significant insightful comments, as well as the expert reviewer's additional generous investment of his/her time.

Comments 2 and 3 are challenging but rewarding questions! We are pleased to report that our model can now predict solidification grain boundary migration (SGBM) **with any remaining liquid fraction or liquid film thickness down to zero**. Our systematic predictions concur well with experimental observations from 10 different dilute alloy systems, confirming the generality of our model. The breakthrough arises from switching from the Scheil model to the six-generation Brody-Flemings solidification model and the finding of a unique latent feature of the latter model for dilute alloys, which enabled us to nicely address Comments 2 and 3.

In addition, we have clarified the fundamental reason for the formation of a solute segregated or enriched SGB (Comment 1). The mechanism is clear and **not** related to solute segregation energy at GB. It is distinctly different from solid-state equilibrium GB segregation. Please refer to our response to Comment 1. Furthermore, we have clarified the integrated driving force for SGBM and elucidated the kinetic process for SGBM. Together they enabled us to respond to Comment 4.

We apologize for the **lengthy** Response Letter as the questions are quite demanding, but the revisions are not excessive (focused on Discussion). All changes made to the manuscript have been highlighted in red.

Reviewer #2 (Remarks to the Author):

The present reviewer has noticed that the authors have done a thorough revision of the manuscript, to address the comments and concerns of the reviewers. The quality of the revised manuscript has been significantly improved. However, as a pity, the theoretical model for SGBM is still not convincing. The major problem is that the authors are trying to quantify the driving force of SGBM by using too rough an energy barrier model based on too much simplification and by using arbitrarily selected parameters. The specific problems are as follows,

1) In equation (9), the authors did not consider the segregation energy of solutes at grain boundary. This is because they simply assume that $X_i(\varphi)$ is equal to X_L in eq. (6). The authors should be aware that different solute elements have different GB segregation energy and

therefore different concentrations at GB even though they have the same matrix concentration. So, it is not proper to use eq. (9) as a general equation to evaluate the SGBM possibility of different solute atoms and different alloys. As for the GB segregation energy term, the authors can compare their equation with eq. (6) of reference [82].

Response: We appreciate these comments and understand that different solute elements have different GB segregation energy values, which leads to different concentrations at GB and therefore different specific GB energy values. In fact, all the models we can find from the literature for estimating the specific interfacial energy of a GB with enriched solute atoms have always incorporated GB segregation energy. However, **the basic premise** for all these treatments or analyses is that **solute segregation to GB occurs in the solid state**, described by the Maclean model or various models based on the Brunauer–Emmett–Teller (BET) approach.

In response to the detailed comments from Reviewer #1, we realized that **the mechanism for the formation of a solute segregated or enriched SGB**, although **also** referred to as ‘solute segregation’, is fundamentally different from the mechanism for solute segregation to GB in the solid state. It is **not** related to solute segregation energy. Rather, it is just due to the change in composition from the liquid state to the solid state, referred to as solute partition.

To clarify these issues, in the following, we first elucidate the fundamental mechanism for solute microsegregation that occurs during solidification by adhering to the basic solidification theory. Then we summarize the current atomistic theory for the formation of a solute segregated or enriched SGB. We acknowledge that we did not articulate these issues well in our last revision.

A. Solute microsegregation during solidification

The fundamental reason for solute microsegregation or enrichment during solidification is straightforward – it is just due to the change in composition from the liquid state to the solid state¹⁻³. The equilibrium binary phase diagram (Fig. R1) best describes this phenomenon through the equilibrium solute partition coefficient $k^{1,3}$, namely

$$k = \frac{X_S}{X_L} \quad (\text{R1})$$

where X_S is the composition of the solid (following the solidus line in Fig. R1, $k < 1$) and X_L is the remaining liquid composition (following the liquidus line in Fig. R1).

Since $X_S < X_L$ (for $k < 1$), **the extra solute atoms arising from the formation of a new solid are always rejected into the remaining liquid**, leading to an accumulative increase of solute

atoms in the remaining liquid as solidification continues. This is the basic reason for solute microsegregation during solidification (**no** other principal reason exists).

Fig. R1 Illustration of equilibrium solidification of a binary alloy with solute concentration X_0 .

Based on Eq. (R1), the Lever Rule or phase-diagram rule fully defines solute microsegregation for equilibrium solidification (complete diffusion in both liquid and solid) as follows

$$X_L = \frac{X_0}{1 - (1-k)f_s} \quad (\text{R2})$$

where X_0 is the alloy solute composition, and f_s is the solid fraction.

According to Eq. (R2), for alloys of $k < 1$, the remaining liquid composition X_L , i.e., the extent of solute microsegregation, increases with increasing f_s until X_L reaches the maximum X_0/k at $f_s = 1$ (Fig. R1). However, **if $k = 1$** , i.e., when the liquid composition is the same as the solid composition, **X_L is always equal to X_0** by Eq. (R2) (i.e., no microsegregation). This occurs only to pure metals for equilibrium solidification. Hence, solute microsegregation that occurs during equilibrium solidification is just due to the change in composition from the liquid state to the solid state. It is **not** related to solute segregation energy at GB.

The Scheil model (1942)^{4,5}, which describes solute microsegregation for non-equilibrium solidification, where no solute back-diffusion occurs from the liquid into solid but complete diffusion occurs in liquid, is given by

$$X_L = X_0(1 - f_s)^{k-1} \quad (\text{R3})$$

Similarly, when $k = 1$, **X_L is always equal to X_0** (no microsegregation). It does **not** involve solute segregation energy. It is the same for all other solidification models (see Refs. [3, 17-19, 1, 7-9]) because the thermodynamics of solidification that leads to solute microsegregation is

clear⁶⁻⁹ (not related to solute segregation energy at GB but driven by the change in free energy per unit volume from the liquid to the solid, where below the melting point or liquidus temperature the solid has a lower Gibbs free energy than the liquid).

B. Formation of a solute segregated or enriched solidification grain boundary (SGB)

The formation of a solute segregated or enriched SGB in the last-stage solidification is a key issue of solidification research. We use Fig. R2 below to summarize the current atomistic theory for SGB formation¹⁰.

Fig. R2 Formation of a SGB in the last-stage solidification from an atomistic view¹⁰, where h denotes the last liquid film thickness while δ is the actual SGB thickness after solidification.

As elucidated above, the solute concentration in the remaining liquid X_L increases with increasing solid fraction f_s (for $k < 1$, for both equilibrium and non-equilibrium solidification). At the same time, solute back-diffusion occurs from the remaining liquid into the solid^{1,3,8}, which reduces the solute concentration. The net increase of the solute content in the remaining liquid is a result of the confluence of these two processes (the Scheil model ignores solute back-diffusion and therefore leads to unrealistic predictions). This trend continues until reaching the last-stage solidification, where the last liquid film (~ 1 nm thick) contains a high solute concentration $X_{L(f_s \rightarrow 1)}$ (solute segregated or enriched). For equilibrium solidification (complete back-diffusion into the solid), this last liquid film composition is defined by Eq. (R2) as X_0/k , which is still solute enriched or segregated for $k < 1$ (most alloys have $k < 1$).

Hence, in the last-stage solidification (for both equilibrium and non-equilibrium solidification), what remains between two abutting grains of different orientations is a thin solute-enriched liquid film with thickness h (Fig. R2a, for $k < 1$). In this stage, back-diffusion of solute atoms from the liquid into the solid continues, accompanied by continuous cooling. As a result, solidification inches forward to further reduce the liquid film thickness. The turning point is

when h reaches δ ¹⁰ ($\delta = \sim 1$ nm, Fig. R2b, typical of a GB thickness^{11,12}). At this point, both the liquid film composition $X_{L(f_s \rightarrow 1)}$ and thickness δ will remain little changed¹⁰ until the required undercooling, known as *coalescence* or *bridging* undercooling, is achieved for this last liquid film to solidify as a SGB^{10,13-16} (Fig. R2c). Accordingly, the SGB inherits the last liquid film composition $X_{L(f_s \rightarrow 1)}$ immediately after solidification, becoming a solute enriched or segregated SGB. This is the basic theory for the formation of a solute segregated or enriched SGB.

Numerous solidification models, as summarized in four major reviews^{3,17-19} and several seminal monographs^{1,7-9}, have been developed to describe the remaining liquid composition X_L down to $f_s \rightarrow 1$ for alloy solidification (the Lever Rule and the Scheil model mentioned earlier are two extreme cases deviating from reality). Among these, the Won-Thomas version²⁰ of the Brody-Flemings²¹ solidification model (the six-generation Brody-Flemings model), which considers the confluence of both solute back-diffusion and dendrite coarsening, provides a comprehensive description of X_L . This will be discussed in detail in our response to Comment 2 below.

In summary, solute microsegregation during solidification stems from the change in composition from the liquid state to the solid state and **is not related to solute segregation energy**. The formation of a SGB results from the solidification of a last liquid film with the thickness of ~ 1 nm and solute concentration $X_{L(f_s \rightarrow 1)}$. The as-solidified GB **inherits** the solute atoms in this last liquid film (~ 1 nm) immediately after solidification. Unlike the unrealistic predictions by the Scheil model due to ignoring solute back-diffusion, $X_{L(f_s \rightarrow 1)}$ is a realistic value with solute back-diffusion (see our response to Comment 2). Our treatment of $X_L^\Phi = X_{L(f_s \rightarrow 1)}$ in eq. (6) and the formulation of eq. (9) in our last submission are fully consistent with the basic solidification theory. We hope we have clarified these issues.

Owing to this comment, we have amended our manuscript with the following additions:

Page 2 Introduction

The basic principles for the formation of such an interface or solidification grain boundary (SGB) in the last-stage solidification have been delineated by Rappaz^{3,4} and other researchers⁵⁻⁷. **It results from the solidification of a last thin liquid film with the thickness**

of ~ 1 nm and solute concentration $X_{L(f_s \rightarrow 1)}$ ³⁻⁷, where f_s denotes solid fraction. In this last stage, both the liquid film composition and thickness remain little changed until the required coalescence or bridging undercooling is achieved for the last liquid film to solidify as a SGB³⁻⁷.

Page 14 Discussion

For equilibrium solidification (complete diffusion in both liquid and solid), X_L is defined by the Lever Rule or phase diagram, while for non-equilibrium solidification that involves no solute back-diffusion into the solid (still complete diffusion in the liquid), X_L can be described by the Scheil-Gulliver model^{90,91}. Both models describe extreme cases deviating from reality. Consequently, numerous models⁹⁰⁻⁹⁸ have been proposed to predict solute microsegregation during solidification (i.e., X_L) since Scheil⁹⁰ introduced the Gulliver model⁹¹ in 1942 ...

Page 15 Discussion

It should be noted that the solute concentration of the last liquid film (~ 1 nm), $X_{L(f_s \rightarrow 1)}$, which will be inherited by the SGB, arises from the cumulative increase of solute in the remaining liquid (for $k < 1$), due to the change in composition from the liquid state to the solid state (solute partition)¹⁻³, minus the effect of solute back-diffusion. It is a natural development of solute partition during solidification¹⁻³. No solute enrichment or microsegregation will occur in the remaining liquid including in the last liquid film if $k = 1$, which leads to $X_L \equiv X_0$ by Eq. (7) or the Lever Rule, or the Scheil model, corresponding to no solute partition.

References (for the response only)

1. Kurz, W.& Fisher, D. J. *Fundamentals of Solidification*. 4th ed., 51, 52, 121, (Trans Tech Publications, Switzerland, 1986).
2. Christian, J.W. *The theory of Transformations in Metals and Alloys*. 667 (Elsevier, Oxford, 2002).
3. Battle, T. P. Mathematical modelling of solute segregation in solidifying materials. *Int. Mater. Rev.* **37**, 249–270 (1992).
4. Gulliver, G. H. The quantitative effect of rapid cooling upon the constitution of binary

-
- alloys. *J. Inst. Met.* **9**, 120 (1913).
5. Scheil, E. Bemerkungen zur Schichtkristallbildung. *Zeitschrift Für Met.* **34**, 70–72 (1942).
 6. Porter, D. A., Easterling, K. E., & Sherif, M. Y. *Phase Transformations in Metals and Alloys*. 3rd. Edn, Chapter 4 Solidification (CRC Press, Taylor & Francis Group, Boca Raton, 2009).
 7. Glicksman, M. E. *Principles of Solidification: An Introduction to Modern Casting and Crystal Growth Concepts*. Chapter 2 and Chapter 5 (Springer, 2010).
 8. Dantzig, J. A. & Rappaz, M. *Solidification*. 1st edn, 401-407 (EPFL Press, Lausanne, Switzerland, 2009).
 9. Stefanescu, D.M., *Science and Engineering of Casting Solidification*. Chapter 4. (Springer. 2015).
 10. Rappaz, M., Jacot, A. & Boettinger, W. J. Last-stage solidification of alloys: theoretical model of dendrite-arm and grain coalescence. *Metall. Mater. Trans. A* **34**, 467-479 (2003).
 11. Loginova, I., Odqvist, J., Amberg, G. & Agren, J. The phase-field approach and solute drag modeling of the transition to massive $\gamma \rightarrow \alpha$ transformation in binary Fe-C alloys. *Acta Mater.* **51**, 1327–1339 (2003).
 12. Toloui, M & Militzer, M. Phase field modeling of the simultaneous formation of bainite and ferrite in TRIP steel. *Acta Mater.* **144**, 786-800 (2018).
 13. Mathier, V., Jacot, A. & Rappaz, M. Coalescence of equiaxed grains during solidification *Modelling Simul. Mater. Sci. Eng.* **12**, 479–490 (2004).
 14. Wang, N., Mokadem, S. Rappaz, M. & Kurz, W. Solidification cracking of superalloy single- and bi-crystals. *Acta Mater.* **52**, 3173–3182 (2004).
 15. Boettinger, W. J., Coriell, S. R., Greer, A. L., Karma, A., Kurz, W., Rappaz, M. & Trivedi, R. Solidification microstructures: recent developments, future directions. *Acta mater.* **48**, 43-70 (2000).
 16. Du, Q. & Jacot, A. A two-dimensional microsegregation model for the description of microstructure formation during solidification in multicomponent alloys: Formulation and behaviour of the model. *Acta Mater.* **53**, 3479–3493 (2005).
 17. Kurz, W., Fisher, D. J. & Trivedi, R. Progress in modelling solidification microstructures in metals and alloys: dendrites and cells from 1700 to 2000. *Int. Mater. Rev.* **64**, 311–354 (2019).
 18. Kurz, W., Rappaz, M., Trivedi, R. Progress in modelling solidification microstructures in metals and alloys. Part II: dendrites from 2001 to 2018. *Int. Mater. Rev.* **66**, 30–76 (2021).
 19. T. Kraft and Y.A. Chang. Predicting microstructure and microsegregation in multicomponent alloys. *JOM*, **49**, 20-28 (1997).
 20. Won, Y. M. & Thomas, B. G. Metall. Simple model of microsegregation during solidification of steels. *Mater. Trans. A.* **32**, 1755–1767 (2001).

21. Brody, H. D. & Flemings, M. C. Solute redistribution in dendritic solidification, *Trans. AIME* **236**, 615-624 (1966)

2) It is not proper to use X_L to substitute $X_i(\phi)$ in Eq. (6), because the former can reach an infinite value at small enough f_L value according to Scheil equation. For the normal solidification condition, the X_L value cannot exceed the eutectic concentration at the end of solidification. Otherwise, eutectic reaction will occur.

Response: Without these comments, without our small breakthrough below. Thank you. Although we fully realized this challenging issue in our last revision, we were unable to tackle it. In the following, we first introduce the Brody-Flemings solidification model and its five subsequent variations. Then we uncover and validate a unique feature of the six-generation Brody-Flemings model for dilute alloys, which enables us to nicely address Comment 3.

A. The Brody-Flemings solidification model and its five subsequent variations

In 1942 Scheil introduced the Gulliver model (1913)^{1,2}, i.e., Eq. (R2) above. Since then, a long list of analytical models has been developed to circumvent the deficiency of the Scheil model³⁻⁹. The four major reviews¹⁰⁻¹³ up to 2018 and several seminal solidification monographs¹⁴⁻¹⁷ have discussed most of these developments. Other recent major models include those proposed in Refs. [18, 19].

The Brody-Flemings solidification model^{3,4} (1966) from MIT is a milestone development in this long journey (see Table R1 for its expression). The model analytically treated the solute back-diffusion issue from the liquid into the solid during solidification (the Scheil model assumes no back-diffusion). It formed an important basis for subsequent further developments, most notably by Clyne and Kurz (1981)⁵, Ohnaka (1986)⁶, Voller (1999)⁷, Voller and Beckermann (1999)⁸, and Won and Thomas (2001)⁹ – see Table R1. Among these efforts, Clyne and Kurz⁵ amended the inconsistency of the Brody-Flemings model with the Lever Rule, which is an important contribution to the model.

As observed from Table R1, the basic form of the Brody-Flemings model has remained unchanged. The subsequent efforts have all centered on refining the parameter α and hence β in the model. In 2001, Won and Thomas⁹ assessed all previous major contributions and further improved the Clyne-Kurz treatment⁵ by incorporating the effect of dendrite coarsening during solidification on solute back-diffusion, based on the work of Voller⁷, and Voller and Beckermann⁸. This is another important and necessary modification as dendrite coarsening

occurs and affects solute back-diffusion. The Won-Thomas version of the Brody-Flemings model, which may be referred to as the six-generation Brody-Flemings model, is given by⁹ (Table R1)

$$X_L = X_0 [1 - (1 - \beta k) f_s]^{k-1} \quad (\text{R4})$$

$$\beta = 2\alpha^+ \left[1 - \exp\left(-\frac{1}{\alpha^+}\right) \right] - \exp\left(-\frac{1}{2\alpha^+}\right) \quad (\text{R5})$$

$$\alpha^+ = \alpha + 0.1 = \frac{4D_s\Delta T}{\lambda^2\dot{T}} + 0.1 \quad (\text{R6})$$

In Eqs. (R4-R6), X_L (remaining liquid composition), X_0 (overall solute concentration), k (phase diagram solute partition coefficient), and f_s (solid fraction) have the same definitions as in Eq. (R2). D_s is the solute diffusion coefficient in the solid, λ is the secondary dendrite arm spacing (SDAS), ΔT is the actual solidification temperature range, and \dot{T} is the cooling rate. Eq. (R4) **reduces to** the Lever Rule Eq. (R1) if $\beta = 1$ and the Scheil model Eq. (R2) if $\beta = 0$. The model completely addresses the unrealistic prediction of the Scheil model in the last-stage solidification due to the introduction of solute back-diffusion.

Table R1 The Brody-Flemings solidification model and its five subsequent versions

Solidification model	Expression	Ref.
Lever Rule	$X_L = \frac{X_0}{1 - (1 - k)f_s}$	Binary phase diagram
Scheil-Gulliver (1942) ^{1,2}	$X_L = X_0(1 - f_s)^{k-1}$	1, 2
Brody-Flemings (1966) ^{3,4}	$\beta = 2\alpha$	3, 4
Clyne-Kurz (1981) ⁵	$\beta = 2\alpha \left[1 - \exp\left(-\frac{1}{\alpha}\right) \right] - \exp\left(-\frac{1}{2\alpha}\right)$	5
Ohnaka (1986) ⁶	$\beta = \frac{2\alpha}{1 + 2\alpha} \text{ or } \frac{4\alpha}{1 + 4\alpha}$	6
Voller (1999) ⁷	$\beta = \frac{2\alpha}{(1 - f_{eutectic})^2 + 2\alpha}$ $f_{eutectic} = 0$ leads to the Ohnaka model	7
Voller-Beckermann (1999) ⁸	$\beta = \frac{2\alpha^+}{(1 - f_{eut})^2 + 2\alpha^+}$ where $\alpha^+ = \alpha + 0.1$	8
Won-Thomas (2001) ⁹	$\beta = 2\alpha^+ \left[1 - \exp\left(-\frac{1}{\alpha^+}\right) \right] - \exp\left(-\frac{1}{2\alpha^+}\right)$	9

To date, the Brody-Flemings model and its five variations in Table R1 have been assessed for various alloy systems, with > 2000 citations all together. It is another basic solidification model after the Lever Rule and the Scheil model. We mentioned this model as the Kurz-Fisher model in our last response by referring to the seminal Kurz-Fisher monograph on solidification¹⁴.

As can be inferred from Eqs. (R4-R6), it is not straightforward to use this model. In Section B below, we first provide our systematic solutions to this model for 10 dilute alloy systems. On this basis, we uncover and validate a unique feature of this model for dilute alloys, which ultimately enables us to address the challenging questions of Comment 3.

B. Solutions to the six-generation Brody-Flemings model Eqs. (R4-R6)

The use of Eq. (R4) requires experimental data on λ (secondary dendrite arm spacing), ΔT (actual solidification temperature range), \dot{T} (cooling rate) and D_s (solute back-diffusion coefficient in solid, a variable throughout the scope of ΔT). While both λ and \dot{T} can be determined experimentally, it is impractical to measure ΔT for each solid fraction f_s , where ΔT can vary in a broad range and hence D_s . As a result, following Kurz and Fisher¹⁴, researchers have generally used a fixed value, e.g., the equilibrium solidification range ΔT_0 in the phase diagram (Fig. R3 below), to replace ΔT for dilute alloys. For non-dilute alloys, there exist two scenarios. In the case when the remaining liquid composition X_L reaches the eutectic composition X_E , it is suggested that ΔT be replaced by equilibrium solidification temperature gap to the eutectic temperature, i.e., $T_L(X_0) - T_E$ in Fig. R3, where $T_L(X_0)$ is the liquidus temperature of the alloy and T_E is the eutectic temperature¹⁴. If X_L is clearly smaller than X_E , then the ΔT_0 approach can still be employed. These suggested approximations enable practical use of the model for general estimates.

Fig. R3 Schematic binary eutectic phase diagram ($k < 1$) for an alloy with solute concentration X_0 . In this diagram, ΔT_0 is the equilibrium solidification range; ΔT is the actual solidification range, $T_L(X_0)$ is the liquidus temperature of the alloy, $T_L(X_0/k)$ is the equilibrium solidification completion temperature; $T_L(X_L)$ is the actual solidification completion temperature, and T_E is the eutectic temperature.

Accurate numerical solutions to Eqs. (R4-R6) are obtainable by simultaneous iteration of ΔT and the ΔT -related D_s , with the assistance of the liquidus line equation due to the use of the phase-diagram k in Eq. (R4). The liquidus line in Fig. R3 follows (using the triangles to formulate the liquidus line slope m_l):

$$X_L = X_0 + \frac{T_L(X_0) - T_L(X_L)}{m_l} = X_0 + \frac{\Delta T}{m_l} \quad (\text{R7})$$

The iteration process comprises the following six basic steps:

- (i) for a given alloy composition X_0 , select an arbitrary solid fraction f_s of interest;
- (ii) let $\Delta T = \Delta T_0$ and calculate D_s at $T = T_L(X_0) - \Delta T$;
- (iii) calculate α^+ and β using Eqs. (R6) and (R5) with experimental data of λ and \dot{T} ;
- (iv) calculate X_L from Eq. (R4) and then ΔT from Eq. (R7);
- (v) repeat step (ii) by replacing ΔT_0 with ΔT obtained from (iv) and then repeat steps (iii) and (iv)
- (vi) continue the process until $X_{L(n)} - X_{L(n-1)} \leq 10^{-5}$, where n is the iteration number.

Table R2 lists all the experimental and phase-diagram parameters used for solving Eqs. (R4-R7) by iteration of both ΔT and D_s for the 10 dilute alloy systems considered in this study. The solutions are plotted in Fig. R4 in terms of X_L versus X_0 , with f_s being varied from 0.25 to 1.0.

Table R2 Basic parameters used to solve Eqs. (R4-R6) by simultaneous iteration of ΔT and D_s

Alloy system X_0 (at.%)	k	m_l (K/at.%)	ΔT_0 (K)	D_s (m ² /s)	SDAS	Cooling	Ref.	
					λ (μm)	rate \dot{T} (K/s)	D_s	λ, \dot{T}
Mg-0.3Zn	0.06	-7.35	38.63	$8.7 \times 10^{-5} \exp(-125073/RT)$	45	8	20	*
Mg-0.3Al	0.36	-6.63	3.54	$1.2 \times 10^{-4} \exp(-141814/RT)$	45	8	20	*
Mg-0.3Sn	0.31	-7.38	4.93	$1.1 \times 10^{-4} \exp(-140925/RT)$	45	8	20	*
Mg-0.3Pb	0.41	-9.63	4.16	$6.5 \times 10^{-6} \exp(-119400/RT)$	45	8	21	*
Al-0.3Mg	0.49	-9.05	2.82	$1.5 \times 10^{-5} \exp(-120500/RT)$	35	10	22	23
Al-0.3Cu	0.15	-7.03	11.95	$4.4 \times 10^{-5} \exp(-133900/RT)$	35	10	22	24
Cu-0.3Sn	0.44	-11.96	7.68	$8.2 \times 10^{-5} \exp(-187600/RT)$	45	20	25	26
Fe-0.3C	0.16	-15.03	24.01	$1.3 \times 10^{-6} \exp(-81398/RT)$	14	65	9	9
Ti-0.3Cr	0.71	-9.06	1.16	$1.8 \times 10^{-6} \exp(-168500/RT)$	25	120	27	28
Zr-0.3Mo	0.78	-6.65	0.63	$2.1 \times 10^{-4} \exp(-286800/RT)$	30	0.2	25	29

*: this study, experimentally measured

In all the above iterations, we have kept monitoring the value of X_L for each alloy system, which is less than its corresponding eutectic composition even down to $f_s = 1.0$. This is due to both solute back-diffusion and the initial low solute concentration X_0 considered in each alloy (0.3at.%). Their confluence dictates the formation of eutectic phases (discussed in page 7 of our manuscript in response to an earlier comment from Reviewer #1).

The consistent perfect linear relationship demonstrated in Fig. R4(a-f) for each selected f_s from 0.25 to 1 with respect to each of the 10 different alloy systems confirms the high self-consistency of the model. Furthermore, as revealed by Fig. R4(d-f), X_L remains essentially constant once $f_s \geq 0.9999$, which perfectly validates an important hypothesis of the atomistic theory for SGB formation mentioned earlier³⁰⁻³⁴. To our knowledge, this should be the first comprehensive validation of that hypothesis using 10 different alloy systems by iteration of both ΔT and D_s in Eqs. (R4-R6).

The systematic evaluation above forms an essential basis for us to reveal an important latent feature of the model for dilute alloys, to be discussed in the next section.

Fig. R4 Numerical solutions to Eqs. (R4-R6) by simultaneous c with respect to 10 dilute binary alloy systems using the parameters listed in Table R2, with the solid fraction f_s being varied from 0.25 to 1.0. A perfect linear relationship is observed in each case, demonstrating the high self-consistency of the model represented by Eqs. (R4-R6).

C. A unique latent feature of the six-generation Brody-Flemings model for dilute alloys

The parameter β , defined by Eq. (R5), is independent of f_s but depends on α^+ or α through Eq. (R6). Accordingly, researchers have mainly focused on estimating α^+ or α and treated β as a simple connecting parameter in predicting X_L using Eq. (R4). No systematic assessment of the dependence of β on both the phase-diagram parameters and experimental variables has been found in the literature. This is, of course, not a trivial effort.

Based on the numerical solutions presented in Fig. R4(a-f), Table R3 lists the values of β calculated for the 10 dilute alloy systems with X_0 being varied from 0.01 at.% (10^{-4} mole fraction) to 0.3 at.% ($k = 0.06-0.78$, covering most practical alloys). It is striking to find out that β varies in just a very narrow range for each dilute alloy system assessed, see the standard deviation (STDEV) listed in Table R3 in relation to the mean value of β in each case (the absolute value of β is different for different systems). The underlying reason is that increasing X_0 will increase ΔT but decrease D_s . Consequently, their influences will largely cancel out. Therefore, at similar cooling rates, the α^+ value calculated from Eq. (R6) will only vary in a narrow range for dilute alloys. On the other hand, β is not highly sensitive to a small variation of α^+ . The confluence results in a very narrow range of variations of β . We confirmed these reasons through step-by-step numerical analyses.

We then systematically assessed the dependence of β on \dot{T} (cooling rate) for different alloy systems by solving Eqs. (R4-R6) through iteration of both ΔT and the ΔT -related D_s with experimental data obtained on λ and \dot{T} . The results are detailed in Table R4. β is effectively independent of \dot{T} (up to 2160 °C/s, covering most practical solidification processes) for all the cases assessed. Also listed in Table R4 is the actual solidification temperature range ΔT corresponding to each cooling rate for each alloy. The influence of \dot{T} on ΔT is similarly negligible in each case, which agrees with the work of Won and Thomas⁹. The underlying reason is similar: for a given dilute alloy, increasing \dot{T} will decrease the SDAS λ (λ and \dot{T} are correlated^{9,35}). As a result, their influences will largely cancel out in Eq. (R6) for the calculation of α^+ , while β is not highly sensitive to a small variation of α^+ . As a result, their confluence results in β being essentially independent of cooling rate.

The above systematic assessments indicate that β can be effectively regarded as a constant for each dilute alloy system. To validate this finding, Fig. R5 (a-f) compares the accurate numerical solutions to Eqs. (R4-R6), obtained by iteration of both ΔT and the ΔT -related D_s , with those obtained from Eq. (R4) by simply applying a mean value of β from Table R3. The match is perfect for each of the 10 dilute alloy systems assessed – they effectively all overlap.

In summary, we can conclude that Eq. (R4) can now be used to easily predict the remaining liquid composition X_L by treating β as a constant for each of the 10 dilute alloy systems assessed (β varies for each system). This will enable us to address Comment 3 by integrating Eq. (R4) over a range of alloy composition from 0 to X_0 . For any new dilute alloy system, its mean β value can be similarly determined. The same observation is expected due to the broad ranges of the k value (0.06-0.78) and cooling rate (1-2160 °C/s) assessed.

Table R3 The values of β calculated from Eqs. (R4-R6) by simultaneous iteration of ΔT and D_s

X_0 (at.%)	β									
	Mg-Sn	Mg-Pb	Mg-Al	Mg-Zn	Fe-C	Al-Cu	Ti-Cr	Cu-Sn	Zo-Mo	Al-Mg
0.01	0.194	0.194	0.194	0.195	0.964	0.197	0.194	0.197	0.196	0.196
0.1	0.195	0.194	0.194	0.197	0.981	0.200	0.197	0.202	0.199	0.199
0.2	0.197	0.195	0.195	0.200	0.990	0.207	0.201	0.209	0.204	0.205
0.3	0.199	0.196	0.196	0.203	0.994	0.213	0.204	0.216	0.209	0.211
Mean	0.196	0.195	0.195	0.199	0.982	0.204	0.199	0.206	0.202	0.203
STDEV	0.00293	0.00135	0.00148	0.00424	0.01050	0.00843	0.00413	0.00965	0.00596	0.00740

Table R4 Effect of cooling rate (\dot{T}) on the secondary dendrite arm spacing (λ), actual solidification gap ΔT , and value of β for different alloys

Alloy (at.%)	SDAS (λ , μm)	Cooling rate (\dot{T} , K/s)	ΔT (K)	β	Mean value of β	STDEV of β	Ref.
Mg-0.63Sn	41.6 ± 14.8	~8	15.1	0.204	0.205	0.0015	*
Mg-0.63Sn	18.2 ± 5.4	~54	15.0	0.207			
Mg-0.63Sn	11.1 ± 3.7	~166	15.0	0.206			
Mg-0.63Sn	4.0 ± 1.4	~1690	15.1	0.203			
Mg-2.67Al	40.1	6	93.3	0.212	0.206	0.0045	36
Mg-2.67Al	13.5	80	95.1	0.206			
Mg-2.67Al	8.8	220	95.4	0.205			
Mg-2.67Al	3.4	2160	96.6	0.201			
Al-0.41Cu	48.5	12	67.6	0.195	0.195	0.0005	37
Al-0.41Cu	24.2	58	67.6	0.195			
Al-0.41Cu	15.7	155	67.6	0.195			
Al-0.41Cu	5.2	1860	67.8	0.194			
Fe-0.0095C ⁹	137.4	1	47.37	Data from Won and Thomas [9]			9
Fe-0.0095C ⁹	44.1	10	47.44				
Fe-0.0095C ⁹	14.2	100	47.52				

*: this study, experimentally measured

Fig. R5 Comparisons of accurate numerical solutions to Eqs. (R4-R6) by iteration of both ΔT and the ΔT -related D_s , referred to as *variable β* , with those obtained from Eq. (R4) by simply applying a mean value of β from Table R3, referred to as *constant β* , with respect to 10 dilute alloy systems.

References (for the response only)

1. Gulliver, G. H. The quantitative effect of rapid cooling upon the constitution of binary alloys. *J. Inst. Met.* **9**, 120 (1913).
2. Scheil, E. Bemerkungen zur Schichtkristallbildung. *Zeitschrift Für Met.* **34**, 70–72 (1942).
3. Brody, H. D. & Flemings, M. C. Solute redistribution in dendritic solidification, *Trans. AIME* **236**, 615-624 (1966)
4. Brody, H. D. Solute redistribution in dendritic solidification. *Ph.D. Thesis*, (Massachusetts Institute of Technology, Cambridge, MA, 1965).
5. Clyne, T. W. & Kurz, W. Solute redistribution during solidification with rapid solid-state diffusion. *Metall. Mater. Trans. A.* **12A**, 965-971(1981).
6. Ohnaka, I. Mathematical analysis of solute redistribution solidification with diffusion in solid phase. *Trans. ISIJ*, **26**, 1046-1050 (1986).
7. Voller, V. R. A semi-analytical model of microsegregation and coarsening in a binary alloy. *J. Cryst. Growth.* **197**, 333-340 (1999)
8. Voller, V. R. and Beckermann, C. A Unified model of microsegregation and coarsening. *Metall. Mater. Trans. A.* **30**, 2183–2189 (1999).
9. Won, Y. M. & Thomas, B. G. Metall. Simple model of microsegregation during solidification of steels. *Mater. Trans. A.* **32**, 1755–1767 (2001).
10. Kurz, W., Fisher, D. J. & Trivedi, R. Progress in modelling solidification microstructures in metals and alloys: dendrites and cells from 1700 to 2000. *Int. Mater. Rev.* **64**, 311–354 (2019).
11. Kurz, W., Rappaza, M., Trivedi, R. Progress in modelling solidification microstructures in metals and alloys. Part II: dendrites from 2001 to 2018. *Int. Mater. Rev.* **66**, 30–76 (2021).
12. Battle, T. P. Mathematical modelling of solute segregation in solidifying materials. *Int. Mater. Rev.* **37**, 249–270 (1992).
13. T. Kraft and Y.A. Chang. Predicting microstructure and microsegregation in multicomponent alloys. *JOM*, **49**, 20-28 (1997).
14. Kurz, W. & Fisher, D. J. *Fundamentals of Solidification*. 4th edn, Chapter 3 and Chapter 6, 130-134 (Trans Tech Publications, Switzerland, 1986).
15. Glicksman, M. E. *Principles of Solidification: An Introduction to Modern Casting and Crystal Growth Concepts*. Chapter 2 and Chapter 5 (Springer, 2010).
16. Dantzig, J. A. & Rappaz, M. *Solidification*. 1st edn, 401-407 (EPFL Press, Lausanne, Switzerland, 2009).
17. Stefanescu, D.M., *Science and Engineering of Casting Solidification*. Chapter 4. (Springer, 2015).

-
18. Gong, T., Chen, Y., Li, S., Cao, Y., Li, D., Chen, X., Reinhart, G. & Nguyen-Thi, H. Revisiting dynamics and models of microsegregation during polycrystalline solidification of binary alloy. *J. Mater. Sci. Technol.* **74**, 155–167 (2021).
 19. Wotczyriski, W., Krajewski, W., Ebner, R. & Kloch, J. The use of equilibrium phase diagram for the calculation of non-equilibrium precipitates in dendritic solidification. Theory. *Calphad* **25**, 401-408 (2001).
 20. Zhong, W. *Measurement of Diffusion Coefficients of Nine Elements in Magnesium and Establishment of a Comprehensive Mobility Database for Lightweight Magnesium Alloys*. Ph.D. Thesis, 96 (The Ohio State University, Columbus, Ohio, 2019).
 21. Zhou, B., Shang, S., Wang, Y. & Liu, Z. Diffusion coefficients of alloying elements in dilute Mg alloys: A comprehensive first-principles study. *Acta Mater.* **103**, 573–586 (2016).
 22. Du, Y., Chang, Y. A., Huang, B., Gong, W., Jin, Z., Xu, H., Yuan, Z, Liu, Y, He, Y. & Xie, F. Y. Diffusion coefficients of some solutes in fcc and liquid Al: critical evaluation and correlation. *Mater. Sci. Eng.* **A363**, 140-151 (2003).
 23. Paliwal, M. & Jung, I. The evolution of the growth morphology in Mg–Al alloys depending on the cooling rate during solidification. *Acta Mater.* **61**, 4848–4860 (2013).
 24. Talamantes-Silva, M. A., Rodríguez, A., Talamantes-Silva, J., Valtierra, S. & Colás, R. Characterization of an Al–Cu cast alloy. *Mater. Charact.* **59**, 1434 – 1439 (2008).
 25. Neumann, G. & Tuijn, C. *Self-Diffusion and Impurity Diffusion in Pure Metals: Handbook of Experimental Data*. 52, 168 (Oxford, Elsevier, 2009)
 26. Kumoto, E. A., Alhadeff, R. O. & Martorano, M. A. Microsegregation and dendrite arm coarsening in tin bronze. *Mater. Sci. Tech.* **18**, 1001-1006 (2002).
 27. Zhu, L., Zhang, Q., Chen, Z., Wei, C., Cai, G., Jiang, L., Jin, Z. & Zhao, J. Measurement of interdiffusion and impurity diffusion coefficients in the bcc phase of the Ti-X (X = Cr, Hf, Mo, Nb, V, Zr) binary systems using diffusion multiples. *J Mater. Sci.* **52**, 3255-3268 (2017).
 28. Tedman-Jones, S. N., McDonald, S. D., Bermingham, M. J., StJohn, D. H. & Dargusch, M. S. Investigating the morphological effects of solute on the β -phase in as-cast titanium alloys. *J. Alloys and Compd.* **778**, 204-214 (2019).

-
29. Suyalatu, Nomura, N., Oya, K., Tanaka, Y., Kondo, R., Doi, H., Tsutsumi, Y. & Hanawa, T. Microstructure and magnetic susceptibility of as-cast Zr–Mo alloys. *Acta Biomater.* **6**,1033-1038 (2010).
 30. Rappaz, M., Jacot, A. & Boettinger, W. J. Last-stage solidification of alloys: theoretical model of dendrite-arm and grain coalescence. *Metall. Mater. Trans. A* **34**, 467-479 (2003).
 31. Mathier, V., Jacot, A. & Rappaz, M. Coalescence of equiaxed grains during solidification *Modelling Simul. Mater. Sci. Eng.* **12**, 479–490 (2004).
 32. Wang, N., Mokadem, S. Rappaz, M. & Kurz, W. Solidification cracking of superalloy single- and bi-crystals. *Acta Mater.* **52**, 3173–3182 (2004).
 33. Boettinger, W. J., Coriell, S. R., Greer, A. L., Karma, A., Kurz, W., Rappaz, M. & Trivedi, R. Solidification microstructures: recent developments, future directions. *Acta mater.* **48**, 43-70 (2000).
 34. Du, Q. & Jacot, A. A two-dimensional microsegregation model for the description of microstructure formation during solidification in multicomponent alloys: Formulation and behaviour of the model. *Acta Mater.* **53**, 3479–3493 (2005).
 35. Dutta, B., & Rettenmayr, M. Effect of cooling rate on the solidification behaviour of Al–Fe–Si alloys. *Mater. Sci. Eng. A* **283**, 218–224 (2000).
 36. Dev, A. & Paliwal, M. Influence of solute elements (Sn and Al) on microstructure evolution of Mg alloys: An experimental and simulation study. *J. Cryst. Growth.* **503**, 28–35 (2018)
 37. Eskin, D., Du, Q., Ruvalcaba, D. & Katgerman, L. Experimental study of structure formation in binary Al–Cu alloys at different cooling rates. *Mater. Sci. Eng. A* **405**, 1–10 (2005).

3) The energy barrier factor $(\sigma_0 - \sigma) / \sigma_0$ in Eq. (9) is very much dependent of the f_L value. In table 2 and Fig. 7, the calculated $(\sigma_0 - \sigma) / \sigma_0$ values are based on $f_L = 0,025$. First, this is a very arbitrary value. I don't see the reason why the grain boundaries have to form at this liquid fraction, instead of other fractions. Secondly, one can calculate that the X_L value of Mg-1.07at%Sn alloy (at $f_L = 0,025$, $k=0,39$) is about 47 wt% Sn. If we select $f_L = 0,01$, the X_L value will become 82.2 wt.%, then the inhibiting (energy barrier) factor term $(\sigma_0 - \sigma) / \sigma_0$ will be much higher than that shown in Table 2. As a result, there may not be enough driving force for SGBM. It has to be mentioned, with either f_L values, the X_L value is much higher than the eutectic point (36.9%Sn) and thus not possible. It is impossible to have so high a concentration of Sn in liquid without forming Mg₂Sn particles. Lastly, the replacement of $X_i(\varphi)$ with X_L , and the f_L values used by the authors have significantly overestimated the energy barrier.

Response: These comments are all perfectly valid due to the unrealistic prediction using the Scheil model in our last revision. With our response to Comment 2, we are ultimately able to address these significant questions by replacing the Scheil model with the six-generation Brody-Flemings model. Thank you for these tough but very rewarding questions.

As summarized earlier, the formation of a SGB results from the solidification of a last liquid film with the thickness of ~ 1 nm and solute concentration $X_{L(f_s \rightarrow 0)}$ ¹⁻⁵. For ease of description, we reproduce Eq. (R4) below

$$X_L = X_0 [1 - (1 - \beta k) f_s]^{\frac{k-1}{1-\beta k}} \quad (\text{R4})$$

Also, we reproduce Eq. (5) and Eq. (6) in our revised manuscript as follows

$$\frac{d\sigma}{dX_i} \approx - \frac{RT}{X_i} \frac{n_i^\phi}{A} \quad (5)$$

$$\Gamma_i = \frac{n_i^\phi}{A} = \Delta\rho X_i^\phi \quad (6)$$

Substituting X_L from Eq. (R4) as X_i^ϕ into Eq. (6) and then into Eq. (5) by noting that X_i stands for X_0 in Eq. (5) (see pages 13-14 of our manuscript), we obtain

$$\frac{d\sigma}{dX_i} \approx - RT \Delta\rho [1 - (1 - \beta k) f_s]^{\frac{k-1}{1-\beta k}} \quad (\text{R8})$$

Rearranging and integrating $d\sigma$ from σ_0 to σ_1 and dX_i from 0 to X_0 yields

$$\int_{\sigma_0}^{\sigma_1} d\sigma \approx \int_0^{X_0} - RT \Delta\rho [1 - (1 - \beta k) f_s]^{\frac{k-1}{1-\beta k}} dX_i \quad (\text{R9})$$

For the sake of simplicity, following Ref. [6], ρ can be taken as the solvent atomic density for dilute binary alloys (see Supplementary Section 4 for justification). In addition, since β can be regarded as a constant for a dilute alloy system (see response to Comment 2), it follows that

$$\sigma_1 - \sigma_0 \approx - RT \Delta\rho X_0 [1 - (1 - \beta k) f_s]^{\frac{k-1}{1-\beta k}} \quad (\text{R10})$$

Eq. (R10) provides an estimate of the specific SGB energy (σ) in a dilute binary alloy when f_s approaches 1.

Fig. R6(a-d) plots $(\sigma_0 - \sigma_1)/\sigma_0$ versus X_0 for the 10 binary dilute alloy systems assessed in this work up to 0.3at.%, with f_s being varied from 0.999 to 1, where the value of σ_0 as a function of temperature is calculated using an established model (see Supplementary Section 5) and β as a constant for each alloy system is taken from Table R3.

By comparing Fig. R6(a-d) with the experimental results listed in Table 2, our model, in which $(\sigma_0 - \sigma_I)/\sigma_0$ is regarded as an inhibiting factor, works well for each dilute Mg-X alloy investigated (see Discussion in page 19 of our revised manuscript), **even if the last liquid film thickness is set to zero**. In addition, the predictions agree well with experimental observations from six additional dilute alloys assessed (see Discussion in page 20 of our revised manuscript). This systematic validation of the model with observations from 10 different dilute alloy systems can be regarded as a strong justification of the rationales for our model.

Fig. R6 Calculated values of $(\sigma_0 - \sigma_I)/\sigma_0$ for 10 dilute binary alloy systems using Eq. (R10) with f_s being varied from 0.999 to 1, where the value of σ_0 as a function of temperature is calculated using an established model (see Supplementary Section 5) and β as a constant for each dilute alloy system is taken from Table R3.

In response to Comments 2 and 3, we have revised our manuscript as follows (most details have been placed in Supplementary Materials)

Since the formation of a SGB results from the solidification of a last liquid film (~ 1 nm thick) with solute composition $X_{L(f_s \rightarrow 0)}$ ³⁻⁷, we have $X_i^\Phi = X_{L(f_s \rightarrow 0)}$ immediately after solidification. The next step is to identify this last liquid film solute concentration $X_{L(f_s \rightarrow 0)}$.

For equilibrium solidification (complete diffusion in both liquid and solid), X_L is defined by the Lever Rule or phase diagram, while for non-equilibrium solidification that involves no solute back-diffusion into the solid from the remaining liquid (still complete diffusion in the liquid), X_L can be described by the Scheil-Gulliver model (breaking down when $f_s \rightarrow 1$)^{90,91}. Both models describe extreme cases deviating from reality. Consequently, numerous models⁹⁰⁻⁹⁸ have been proposed to predict the last-stage solidification since Scheil⁹⁰ introduced the Gulliver model⁹¹ in 1942. Among these, the Brody-Flemings model (1966)^{92,93} and its five subsequent variations⁹⁴⁻⁹⁸ have been assessed in detail for various alloy systems. The Won-Thomas version⁹⁸, which may be referred to as the six-generation Brody-Flemings model, considered both solute back-diffusion and dendrite coarsening based on previous efforts by Clyne and Kurz⁹⁴, Ohnaka⁹⁵, Voller⁹⁶, and Voller and Beckermann⁹⁷. The model, which retains the basic form of the Brody-Flemings model, is given by

$$X_L = X_0 [1 - (1 - \beta k) f_s]^{\frac{k-1}{1-\beta k}} \quad (7)$$

$$\beta = 2\alpha^+ \left[1 - \exp\left(-\frac{1}{\alpha^+}\right) \right] - \exp\left(-\frac{1}{2\alpha^+}\right) \quad (8)$$

$$\alpha^+ = \alpha + 0.1 = \frac{4D_s \Delta T}{\lambda^2 \dot{T}} + 0.1 \quad (9)$$

where X_L (remaining liquid composition), λ (SDAS) and \dot{T} (cooling rate) have the same definitions as defined earlier, X_0 is the alloy composition, k solute partition coefficient, f_s solid fraction, ΔT actual solidification temperature range (X_L related), and D_s solute diffusion coefficient in the solid (ΔT -related). Eq. (7) reduces to the Lever Rule if $\beta = 1$ and the Scheil-Gulliver model if $\beta = 0$.

Substituting $X_{L(f_s \rightarrow 1)}$ from Eq. (7) as X_i^Φ into Eq. (6) and further into Eq. (5) by noting that X_i stands for X_0 in Eq. (5), and then integrating the resulting Eq. (5) from σ_0 to σ_1 for $d\sigma$ and 0 to X_0 for dX_i , we obtain for $f_s \rightarrow 1$

$$\int_{\sigma_0}^{\sigma_1} d\sigma \approx \int_0^{X_0} -RT\Delta\rho[1 - (1 - \beta k)f_s]^{\frac{k-1}{1-\beta k}} dX_i \quad (10)$$

As detailed in Supplementary Section 5, our systematic assessments of the dependence of β on X_0 , k , \dot{T} , λ , ΔT , and D_s for 10 different dilute alloy systems ($k = 0.06 - 0.78$) over a wide range of cooling rate (up to 2160 °C/s) have uncovered that β can be treated as a constant for a dilute alloy system. In addition, for the sake of simplicity, following Ref. [88], ρ is taken as the solvent atomic density for a dilute alloy (see Supplementary Section 4 for justification). It follows that

$$\sigma_0 - \sigma_1 \approx RT\Delta\rho X_0 [1 - (1 - \beta k)f_s]^{\frac{k-1}{1-\beta k}} \quad (11)$$

where $f_s \rightarrow 1$. Eq. (11) provides an estimate of the specific SGB energy (σ) in a dilute binary alloy, where the value of β is listed in Table S4 for each dilute alloy system assessed in this research, which can be similarly determined for any new dilute alloy system of interest.

Pages 19-20 Discussion

As observed from Fig. 7, once $f_s \geq 0.9999$, the plots look identical. The underlying reason is that the last liquid film composition X_L remains little changed once $f_s \geq 0.9999$ (Fig. S4), consistent with the atomistic theory proposed for SGB formation³.

References (for the response only)

1. Rappaz, M., Jacot, A. & Boettinger, W. J. Last-stage solidification of alloys: theoretical model of dendrite-arm and grain coalescence. *Metall. Mater. Trans. A* **34**, 467-479 (2003).
2. Mathier, V., Jacot, A. & Rappaz, M. Coalescence of equiaxed grains during solidification *Modelling Simul. Mater. Sci. Eng.* **12**, 479–490 (2004).
3. Wang, N., Mokadem, S. Rappaz, M. & Kurz, W. Solidification cracking of superalloy single- and bi-crystals. *Acta Mater.* **52**, 3173–3182 (2004).
4. Boettinger, W. J., Coriell, S. R., Greer, A. L., Karma, A., Kurz, W., Rappaz, M. & Trivedi, R. Solidification microstructures: recent developments, future directions. *Acta mater.* **48**, 43-70 (2000).
5. Du, Q. & Jacot, A. A two-dimensional microsegregation model for the description of microstructure formation during solidification in multicomponent alloys: Formulation and behaviour of the model. *Acta Mater.* **53**, 3479–3493 (2005).
6. Liu, F. & Kirchheim, R. Nano-scale growth inhibited by reducing grain boundary energy through solute segregation. *J. Cryst. Growth* **264**, 385-391 (2004).

4) The approach used by the author to determine the energy driving factor of SGBM is not reasonable (1). The calculated values of $(A_1-A_2)/A_1$ shown in Table 2 are based on the GBs after completion of SGBM (1). If we consider the real process of SGBM, the $(A_1-A_2)/A_1$ should be very small at the beginning of SGBM, thus $(A_1-A_2)/A_1 \ll (\sigma_0-\sigma)/\sigma_0$. In this case, SGBM can not happen, because there is not enough driving force (2). This is fatal to the theoretical model proposed in the manuscript. Since SGBM is a phenomenon known for decades, without a proper theoretical interpretation the novelty of this manuscript is much reduced. The manuscript cannot be accepted in the present form as it is.

Response: We concur that we did not articulate the driving factor and kinetic process for SGBM clearly in our last revision. To clarify these issues, we respond to Comment 4 below in terms of **(A)** the driving force for SGBM; **(B)** the kinetic process for SGBM; and **(C)** the beginning issue of SGBM related to the comment on “ $(A_1-A_2)/A_1 \ll (\sigma_0-\sigma)/\sigma_0$ ”.

A. The driving force for SGBM

As shown in Fig. 2 of our manuscript, after migration, the GB triple junctions approach a near $120^\circ-120^\circ-120^\circ$ configuration. Furthermore, the ratio of grain size d to grain side length a approaches $\sqrt{3}$ ($d/a \approx 1.71-1.72$, Table 2 of our manuscript). Clearly, SGBs migrate towards their equilibrium state, which has the lowest free energy and is their final state.

The *integrated* driving force (often called simply driving force¹) for GBM is clear. “The deviation from equilibrium at the interface provides the driving force for the interface migration”¹. It refers to “the difference in free energy (usually Gibbs free energy) of the initial and final states”² and “Like all other natural processes, grain boundary migration always results in a reduction in total free energy”³. Following these definitions, for a closed system, at constant pressure and temperature, the difference in free energy between the **initial** state (G_I) and its final or equilibrium state (G_E), i.e., $G_E - G_I$, provides the driving force ΔG_{DF} ²⁻⁴. For SGBM in dilute alloys, excluding nanoscale grains, based on our formulation, ΔG_{DF} can be written as

$$\Delta G_{DF} \approx \sigma_E A_E - \sigma_I A_I \quad (\text{R11})$$

where the subscripts E and I refer to the equilibrium (final) and initial states, respectively.

Eq. (R11) concurs with the current understanding that the main driving force for SGBM is the reduction in the total GB energy^{5,6}.

Since after migration the SGBs always end up inside the original grains, for a dilute alloy, σ_E

can be taken as that for the solvent metal grains, i.e., σ_0 . Rearranging Eq. (R11) with $\sigma_E \approx \sigma_0$ gives

$$\frac{\Delta G_{DF}}{\sigma_0 A_I} \approx - \frac{(A_I - A_E)}{A_I} + \frac{\sigma_0 - \sigma_I}{\sigma_0} \quad (\text{R12})$$

As a necessary condition, SGBM requires $\Delta G_{DF} < 0$, i.e., $(A_I - A_E)/A_I > (\sigma_0 - \sigma_I)/\sigma_0$, where $(A_I - A_E)/A_I$ may be regarded as the driving factor and $(\sigma_0 - \sigma_I)/\sigma_0$ as the inhibiting factor.

Note that Eq. (R12) applies only to the initial state (I) and the equilibrium (E) or final state by the definition of ΔG_{DF} . We used subscripts 1 and 2 previously, rather than I and E , without restricting their implications, which was potentially misleading (may be part of the reason leading to the question of “ $(A_1 - A_2)/A_1 \ll (\sigma_0 - \sigma)/\sigma_0$ ”).

A useful point to note is that, in the case of a thermally activated process, before G_I can decrease to G_E , the system must pass through a so-called *transition* or *activated* state with an activation free energy barrier ΔG^a above G_I ^{3,4} or even through a series of free energy barriers⁴. Consequently, prior to overcoming this ΔG^a , any move from the initial state towards the final state will result in an increase in free energy (i.e., $\Delta G > 0$), where no obvious connection has been established between ΔG_{DF} (i.e., $G_E - G_I$) and ΔG^a ⁴. However, this does not mean that the process will not occur – thermal activation can overcome ΔG^a under suitable conditions³. Nonetheless, an athermal process is distinctly different as will be discussed below for the SGBM observed.

B. The kinetic process for SGBM

Since a SGB results from the solidification of a last thin liquid film (~ 1 nm thick) with an enriched solute concentration X_L (for alloys of $k < 1$), the SGB is associated with rich solute atoms immediately after solidification. In this regard, it is equivalent to having solute clouds attached to it. Owing to the irregular peripheries of the abutting dendritic grains (Fig. 2a in the manuscript), a SGB is typically tortuous (A_I), deviating from its equilibrium or final state. Consequently, in principle, these tortuous SGBs always lean towards reaching their equilibrium or final state by migration. At low temperatures, the solute clouds will tend to move along with the GBs^{2,7-14}, resulting in solute dragging thereby affecting GBM. However, at high temperatures^{2,7-14}, when the driving force is sufficiently large^{7-10,12,13} (versus the solute dragging force, related to solute content and type^{11,12}), GBs cannot be held by solute atoms, leading to solute detachment or breakaway^{2,7-14} (solute re-attachment occurs below a certain temperature^{8,14}).

The breakaway frees the GBs and can result in athermal GB motion^{7,8,10,14,17} until solute reattachment occurs^{8,14}. Athermal motion has been investigated both theoretically and experimentally^{1-3,15-31}. It can be regarded as a barrierless process with negligible or no activation barrier^{15,16,18,26,28,30}. It does not depend on time^{2,16} as there is no need to wait for thermal activation and is diffusionless^{1,3,4}. It occurs through a “co-operative movement” mechanism^{1,3,4,17,19} or “collective shuffling mechanism”^{20-29,31}, by which many atoms can move co-operatively at the same time with a velocity approaching that of sound in solid^{2,3,31,21,25,27}. More specifically, atoms traverse just the interatomic spacing (e.g. ~ 1.33 Å in a $\Sigma 5$ [010] tilt boundary with $\alpha = 0^\circ$ at 500 K²⁰)^{1,3,4,17,19,21-23,25-29} to transfer themselves from one side of the grain boundary to the other side, by some form of deformation and/or rotation^{1,27,28}. In-situ observations of atomic-scale GBM in ultra-pure Au (99.9999%) films (<10 nm thick) at 0.5-0.74 T_m by high-resolution transmission electron microscopy have confirmed this collective shuffling GB motion mechanism²⁵⁻²⁸.

As shown in Fig. 4 (c, c') of our manuscript, noticeable SGBM occurred even at the cooling rate of 1690 °C/s when the molten alloy was cooled from 720 °C, which is substantially faster than water quenching^{32,33} (diffusionless). The SGBM observed does not depend on time over the cooling rate range from 8 °C/s to 1690 °C/s (related to grain size, Table 1), further confirming its diffusionless nature. SGBM occurs near the solidus temperature³⁴⁻³⁶. The migration distance observed in different Mg alloys in this study ranges from 1 μm to 87 μm (Table 1, Figs. 2-4, Fig. S3). The speed of sound in solid Mg is known to be 5770 m/s³⁷. The estimated SGBM time thus falls in the range of 170 ps to 15 ns, which is consistent with the athermal GB motion processes investigated using molecular dynamics simulations (nano or picoseconds)^{9,18,22,38,39-42}. Hence, we propose that the SGBM observed in this work occurred athermally.

C. The beginning of SGBM

As discussed above, athermal GB motion is diffusionless and independent of time^{3,16}. It occurs at the speed of sound in solid^{2,3,31,21,25,27}, which corresponds to 5770 m/s in solid Mg³⁷. It is therefore **not** a step-by-step diffusional process. Rather, considering the remarkable athermal migration speed (5.77×10^9 $\mu\text{m/s}$) and the migration distance observed (1-87 μm), the beginning is just the end.

Enlightened by this comment, we have examined the cases where SGBs have only slightly departed from their as-solidified positions. Fig. R7 displays a few such cases (see arrows). As observed, in each case, despite the limited migration, the associated grains have all reached a

near equilibrium state (120° - 120° - 120° triple junctions + $d/a \approx \sqrt{3}$). We have added Fig. R7 as Fig. S3 to emphasize that even for small SGBM, the associated SGBs have moved towards reaching a stable or a near equilibrium state.

Fig. R7 Examples of SGBM where the SGBs have only slightly departed from their as-solidified positions but they have clearly moved towards reaching a near equilibrium state (120° - 120° - 120° triple junctions + $d/a \approx \sqrt{3}$). (a) as-cast Mg-0.3at.%Pb and (b) as-cast Mg-0.3at.%Pb-0.5at.%Zr.

Owing to Comment 4, we have largely revised our Discussion with the following amendments:

Page 12 Discussion

A theoretical model for SGBM. The integrated driving force (often called simply driving force⁸²) for GBM is clear. “The deviation from equilibrium at the interface provides the driving force for the interface migration”⁸³⁻⁸⁵, which always results in a reduction in total free energy.⁸⁴ As observed in Fig. 2, Fig. S3 and Table 2, after migration, SGBs approach their equilibrium state (near 120° -triple junctions plus the ratio of grain size to grain side length close to $\sqrt{3}$).

Pages 12-13 Discussion

Applying Eq. (1) to the as-solidified initial SGB state (I) and its final or equilibrium state (E) allows for the determination of deviation in free energy from equilibrium, i.e., the driving force (DF) by definition⁸³⁻⁸⁵, denoted as $\Delta G_{DF} = G_E - G_I$. As pointed out earlier, G_B can be regarded as a constant (excluding nano-grains), we have

$$\Delta G_{DF} = \approx \sigma_E A_E - \sigma_I A_I \quad (2)$$

The formulation of Eq. (2) is consistent with the current understanding that the main driving force for SGBM is the reduction in the total GB energy.^{20,24}

Since after migration SGBs always end up inside the grains (Fig. 2), for a dilute alloy, σ_E can be taken as that for the solvent metal grains σ_0 . Rearranging Eq. (2) with $\sigma_E \approx \sigma_0$ gives

$$\frac{\Delta G_{DF}}{\sigma_0 A_I} \approx - \frac{(A_I - A_E)}{A_I} + \frac{\sigma_0 - \sigma_I}{\sigma_0} \quad (3)$$

As a necessary condition, SGBM requires $\Delta G_{DF} < 0$, i.e., $(A_I - A_E)/A_I > (\sigma_0 - \sigma_I)/\sigma_0$, where $(A_I - A_E)/A_I$ may be regarded as the driving factor and $(\sigma_0 - \sigma_I)/\sigma_0$ as the inhibiting factor. It is noteworthy that σ_0 can vary significantly, depending on the solvent system (Supplementary Tables S1 and S2). Accordingly, ΔG_{DF} may vary significantly for a similar A_I . Another point to note is that, in the case of a thermally activated process, before G_I can decrease to G_E , the system must pass through a so-called *transition* or *activated* state with an activation free energy barrier ΔG^a above G_I .^{84,85}

Pages 20-22 Discussion

Kinetics of SGBM. Immediately after solidification, a SGB is associated with solute atoms inherited from the last liquid film (~ 1 nm thick, $X_{L(f_s \rightarrow 1)}$). It is equivalent to having solute clouds attached to it. Owing to the irregular peripheries of the abutting dendritic grains (Fig. 2a), a SGB is typically tortuous (A_I), deviating from its equilibrium state. Consequently, these tortuous SGBs always lean towards migrating to their equilibrium or final state. At low temperatures, the solute clouds will tend to move along with the GBs,^{83,101-108} resulting in solute dragging affecting GBM. However, at high temperatures^{83,101-108}, when the driving force is sufficiently large^{101-104,107,108} (versus the solute dragging force, related to solute content and type^{83,105}), GBs cannot be held by solute atoms leading to solute detachment or breakaway^{83,101-108} (solute re-attachment occurs below a certain temperature^{102,108}).

The breakaway frees the GBs and can result in athermal GB motion^{101,102,104,108,111} until the solute reattachment occurs^{102,108}. Athermal motion has been investigated both theoretically and experimentally.^{36,83-86,109-124} It can be regarded as a barrierless process^{109,110,112,119,121,123} and does not depend on time^{84,110} as there is no need to wait for any thermal activation. It is diffusionless^{82,84,85} and occurs through a “co-operative movement”^{82-85,111,113} or

“collective shuffling” mechanism^{36,114-122,124}, by which many atoms can move cooperatively at the same time with a velocity approaching that of sound in solid^{83,84,124,36,116,118}. More specifically, atoms traverse just the interatomic spacing (e.g., 1.33 Å in a $\Sigma 5$ [010] tilt boundary with $\alpha = 0^\circ$ at 500 K¹¹⁵)^{36,83,84,111,113,115,116,118-122} to transfer themselves from one side of the GB to the other side, by some form of deformation and/or rotation to realize significant collective movement^{82,120,121}. In-situ observations of atomic-scale GBM in ultra-pure Au (99.9999%) films (<10 nm thick) at 0.5-0.74T_m by high-resolution transmission electron microscopy have confirmed this collective shuffling GB motion mechanism¹¹⁸⁻¹²¹.

As shown in Figs. 4 (c, c’), noticeable SGBM occurred even at the cooling rate of 1690 °C/s when the molten alloy was cooled from 720 °C, which is substantially faster than water quenching (diffusionless)^{125,126}. The SGBM observed does not depend on time over the cooling rate range from 8 °C/s to 1690 °C/s (related to grain size, Table 1), further confirming its diffusionless nature. SGBM occurs near the solidus temperature.^{9,24,25} The SGBM distance observed in different Mg alloys in this study ranges from 1 μm to 87 μm (Table 1, Figs. 2-4, Fig. S3). Applying the speed of sound in solid Mg (5770 m/s)¹²⁷, the estimated migration time thus falls in the range of 170 ps to 15 ns, in line with the athermal GB motion processes investigated using molecular dynamics simulations (nano or picoseconds)^{102,112,115,128-132}. Hence, we propose that the SGBM observed in this work occurred athermally.

References (for the response only)

1. M. Hillert, *Phase Equilibria, Phase Diagrams and Phase Transformations*. 57, 391-399 (Cambridge University Press, Cambridge,1998).
2. Christian, J.W. *The theory of Transformations in Metals and Alloys*. 4, 850, 977-978 (Elsevier, Oxford, 2002).
3. Porter, D. A., Easterling, K. E., & Sherif, M. Y. *Phase Transformations in Metals and Alloys*. 3rd. Edn. 59, 60, 176, 178, 383-416, 463 (CRC Press, Taylor & Francis Group, Boca Raton, 2009).
4. Mittemeijer E. J. *Fundamentals of Materials Science, the Microstructure–Property Relationship Using Metals as Model Systems*. 307, 373, 410(Heidelberg, Springer Verlag; 2010).
5. Shibata, S.& Watanabe, T. The effect of the surface on grain boundary migration in austenitic stainless steel weld metal, *Mater. Trans. A* **32A**, 1453-1458 (2001).
6. Matsuda, F., Nakagawa, H., Ogata, S. & Katayama, S. Fractographic investigation on

-
- solidification crack in the vareststraint test of fully austenitic stainless steel: studies on fractography of welded zone (III). *Trans. of JWRI* **7**, 59-70 (1978).
7. Mendeleev, M. I. & Srolovitz, D. J. Impurity effects on grain boundary migration. *Modelling Simul. Mater. Sci. Eng.* **10**, R79–R109 (2002).
 8. Sursaeva, V. & Zieba, P. Diffusion Impurity Drag of Twin Grain Boundaries and Triple Junctions Motion in Zinc. *Def. Diff. Forum* **237 – 240**, 578-583 (2005).
 9. Rickman, J., Phillpot, S., Wolf, D., Woodraska, D., & Yip, S. On the mechanism of grain-boundary migration in metals: A molecular dynamics study. *J. Mater. Res.* **6**, 2291-2304 (1991).
 10. Sursaeva, V. Athermal motion of Grain Boundaries, Twins and Triple Junctions in Zn. *Mater. Sci. Forum* **467-470**, 801-806 (2004).
 11. Aust, K. T. & Rutter, J. W., Grain boundary migration in high-purity lead and dilute lead-tin alloys. *Trans. AIME* **215**, 119 (1959).
 12. Hersent, E., Marthinsen, K. & Nes, E. The effect of solute atoms on grain boundary migration: a solute pinning approach. *Metall. Mater. Trans. A* **44**, 3364–3375 (2013).
 13. Cahn, J. W. The impurity-drag effect in grain boundary motion. *Acta Metall.* **10**, 789-798 (1962).
 14. Sursaeva, V. & Zieba, P. Shape of Moving Grain Boundary and its Influence on Grain Boundary. *Def. Diff. Forum* **249**, 183-188 (2006).
 15. Estrin, E. I. Athermal processes in solids. *Bull. Russ. Acad. Sci. Phys.* **71**, 1649–1655 (2007).
 16. Laughlin, D. E., Jones, N. J., Schwartz, A. J. & Massalski, T. B. Thermally activated martensite: its relationship to non-thermally activated (athermal) martensite. *In: Proceedings of the international conference on martensitic transformations*, (Eds. by Olson, G. B., Lieberman, D. S. & Saxena, A), 141-144 (Santa Fe, NM USA, 2008)
 17. Kopetsky, Ch. V., Shvindlerman, L. S. & Sursayeva, V. G. Effect of athermal motion of grain boundaries. *Scr. Mater.* **12**, 953-956 (1978).
 18. Homer, E. R., Holm, E. A., Foiles, S. M. & Olmsted, D. L. Trends in Grain Boundary Mobility: Survey of Motion Mechanisms. *JOM* **66**, 114–120 (2014).
 19. Gutkin, M. Y., Mikaelyan, K. N. & Ovid'ko, I. A. Athermal grain growth through cooperative migration of grain boundaries in deformed nanomaterials. *Scr. Mater.* **58**, 850-853 (2008).
 20. Chesser, I., Holm, E. & Bunnell. Optimal transportation of grain boundaries: A forward model for predicting migration mechanisms. *Acta Mater.* **210**, 116823 1-14 (2021).
 21. Gottstein, G. & Shvindlerman, L. S. Grain boundary migration in metals: thermodynamics, kinetics, applications. 2nd edn, 250-251 (CRC Press, Taylor & Francis Group, Boca Raton, 2010).

-
22. Zhang, H., Du, D. & Srolovitz, D. J. Effects of boundary inclination and boundary type on shear-driven grain boundary migration. *Philos. Mag.* **88**, 243–256 (2008).
 23. Babcock, S. E. & Balluffi, R. W. Grain boundary kinetics—II. In situ observations of the role of grain boundary dislocations in high-angle boundary migration. *Acta metall.* **37**, 2367-2376 (1989).
 24. Jhan, R. J. & Bristowe, P. D. A molecular dynamics study of grain boundary migration without the participation of secondary grain boundary dislocations. *Scr. Metall.* **24**, 1313-1318 (1990).
 25. Merkle, K. L., Thompson, L. J. & Phillipp, F. Dynamics of grain boundary motion at the atomic level. *Mat. Res. Soc. Symp. Proc.* **819**, N6.1.1- N6.1.12(2004).
 26. Merkle, K. L., Thompson, L. J. & Phillipp, F. High-Resolution Electron Microscopy of Grain Boundary Migration. *Mat. Res. Soc. Symp. Proc.* **652**, Y2.4.1- Y2.4.12 (2000).
 27. Merkle, K. L., Thompson, L. J. & Phillipp, F. In-Situ HREM studies of grain boundary migration. *Interface Sci.* **12**, 277–292 (2004).
 28. Merkle, K. L., Thompson, L. J. & Phillipp, F. Collective effects in grain boundary migration. *Phys. Rev. Lett.* **88**, 225501 1-225501 4 (2002).
 29. Wei, J., Feng, B., Ishikawa, R., Yokoi, T., Matsunaga, K., Shibata, N. & Ikuhara, Y. Direct imaging of atomistic grain boundary migration. *Nat. Mater.* 2021 (DOI: [10.1038/s41563-020-00879-z](https://doi.org/10.1038/s41563-020-00879-z)).
 30. Olmsted, D. L., Holm, E. A. & Foiles, S. M. Survey of computed grain boundary properties in face-centered cubic metals—II: Grain boundary mobility. *Acta Mater.* **57**, 3704–3713 (2009).
 31. Banerjee S, Mukhopadhyay P. Chapter 4 Martensitic Transformation in *Phase transformation: Example from Titanium and Zirconium alloys*. 362 (Oxford: Elsevier Science, 2007).
 32. Paliwal, M. & Jung, I. The evolution of the growth morphology in Mg–Al alloys depending on the cooling rate during solidification. *Acta Mater.* **61**, 4848-4860 (2013).
 33. Sun, M., StJohn, D. H., Easton, M. A., Wang, K. & Ni, J. Effect of cooling rate on the grain refinement of Mg-Y-Zr Alloys. *Metall Mater Trans A* **51**, 482–496 (2020).
 34. Pigenko, A. A., Chernyshova, T. A., Shorshorov, M. K. & Dontsova, A. Y. Structure formation and in cast and deposited maraging steel N18K8M3TYu. *Met. Sci. Heat Treat.* **17**, 751-754 (1975).
 35. Matsuda, F., Nakagawa, H., Uehara, T., Katayama, S. & Arata, Y. A new explanation for role of delta-ferrite improving weld solidification crack susceptibility in austenitic stainless steel. *Trans. of JWRI* **8**, 105-112 (1979).
 36. Katayama, S., Fujimoto, T., & Matsunawa, A. Correlation among solidification process,

-
- microstructure, microsegregation and solidification cracking susceptibility in stainless steel weld metals. *Trans. of JWRI* **14**, 123-138 (1985).
37. Solid and metals – Speed of sound. https://www.engineeringtoolbox.com/sound-speed-solids-d_713.html.
38. Ulomek, F & Mohles, V. Molecular dynamics simulations of grain boundary mobility in Al, Cu and γ -Fe using a symmetrical driving force. *Modelling Simul. Mater. Sci. Eng.* **22**, 055011:1-18 (2014).
39. Zhang, H., Srolovitz, D. J., Douglas, J. F. & Warren, J. A. Characterization of atomic motion governing grain boundary migration. *Phys. Rev. B* **74**, 115404:1-10 (2006).
40. Peng, H. R., Liu, W., Hou, H. Y. & Liu, F. Pinning effect of coherent particles on moving planar grain boundary: Theoretical models and molecular dynamics simulations. *Materialia*. **5**,100225: 1-11 (2019).
41. Schönfelder, B., Gottstein, G. & Shvindlerman, L. S. Comparative study of grain-boundary migration and grain-boundary self-diffusion of [001] twist-grain boundaries in copper by atomistic simulations. *Acta Mater.* **53**, 1597–1560 (2005).
42. Trautt, Z.T. & Y. Mishin. Grain boundary migration and grain rotation studied by molecular dynamics. *Acta Mater.* **60**, 2407-2424 (2012).

Reviewers' comments:

Reviewer #2 (Remarks to the Author):

In comparison to the previous version, the authors have improved the solidification model by considering the back diffusion in the solid grain, which can give a better estimation of the chemical concentrations of the last liquid film and therefore the solute concentration at the GB. However, this is not sufficient to make their model more predictive. It is because the concern by reviewer 2, "different solute elements have different GB segregation energy", is still not addressed in the revised manuscript. Although the SGB concentration is not directly dependent of GB segregation energy of elements, the specific GB energy will be strongly influenced by the solute segregation energy, which is alloying element dependent. GBs containing different solute elements of the same concentration level, will have different GB energies. However, equation 11 in the revised manuscript just tells us the driving factor of $(\sigma_0 - \sigma)$ is only dependent of the final concentration of the last liquid film (about 1nm thick) during solidification, independent of solute type. The only conclusion we can get from their model is that a dilute alloy is easier to get SGBM than a denser alloy for the same alloy system. If for different alloys systems, the alloys with less solute enrichment at the grain boundary (less solidification microsegregation) are easier to get SGBM while the type of alloying elements has no effect at all. Frankly, I would not consider this model as predictive.

The major problem is: when deducing Equation 4 and 5, the authors dropped down the term of $\mu_i(\phi)$ which is relating to the grain boundary concentration of different solute elements. Therefore, the effect of solute concentration of different elements in GB on the specific GB energy is not properly addressed. In the revised manuscript, it is suggested that SGBM is an athermal process, which is independent of time. Then it means that all SGBM could complete during solidification independent of cooling rate. But this is contradictory to their experimental results shown in Fig. 5. A clear migration of GB from the solute rich zone can be observed in the lower part of the GB (to the left of mark 4 in Fig. 5b). This may imply that SGBM is similar to the GB migration during post solidification annealing treatment.

More importantly, since 1960's there have been extensive research efforts, of both experimental and theoretical, on interface (including grain boundary) migration problems in materials. A classic example is chemically induced grain-boundary migration (CIGM). There can be different driving forces for GB migration to happen, including capillary force, chemical force, elastic force and frictional force. Cahn, Hillert and Purdy etc have developed models to describe the mechanisms and kinetics behind. As a special example of grain boundary migration problems, SGBM is happening during solidification process. As reviewed in the manuscript, SGBM occurs near the solidus temperature. But the authors did not clarify if SGBM is happening before or after the solidification is completed (above or below solidus temperature). If for the former case, the last liquid film in between grains should have played important roles during the SGBM, while for the latter, the SGBM is a pure solid state migration process. In any case, SGBM should be comparable to CIGM. However, in this manuscript the authors just simply attributed SGBM to the capillarity force as the driving force without considering other possible effects like elastic strain energy. In comparison to the previous classical work, the manuscript, unfortunately, did not bring any significantly new insights of the thermodynamics or kinetics of the SGBM.

Appeal against Reviewer #2's third round of comments - NCOMMS-19-11145C

A snapshot of the work and previous reviews

Solidification grain boundary migration (**SGBM**) occurs in metals and alloys during casting, welding, and 3D metal printing, covering both industry alloys and emerging high entropy alloys. It is the final stage of the grain formation process through solidification and is important for solidification cracking or hot tearing, intergranular fracture, and grain boundary corrosion.

Since **SGBM** was reported in 1961, no theory had been able to predict it until our work. We quantitatively predicted the dependence of **SGBM** on **solute type and content**, validated with remarkable consistency in **10 different** alloy systems under various solidification conditions.

Reviewer #1 recommended publication after we addressed his/her 17 insightful questions. **Reviewer #3** rated the work as “**appealing and interesting**” and a “**valuable contribution**”, and unfortunately could not continue to serve as a reviewer after the first round of review.

Reviewer #2 appreciated our experimental work (“**interesting and valuable**”) and provided three rounds of review, for which we are **very grateful**. In his/her 3rd review, **Reviewer #2** made **four** major criticisms and **one** concluding comment, which led to the rejection of this work. After examining these criticisms, we decided to appeal the validity of each of them because

- The **first three** criticisms all **contradict** the basic facts or data presented and discussed in our work (the first three criticisms are fundamentally the same).
- Most notably, Reviewer #2 has **finally** admitted and accepted that his/her previous insistence that the **SGB** composition depends on the **GB** segregation energy of solutes is **incorrect**. This insistence has been the **central criticism** of our work from Reviewer #2, but from the outset, it **contradicts** both solidification science and the correct opinions or suggestions of **Reviewer #1**.
- The fourth criticism **contradicts** Reviewer #2's own previous full confirmation of the same hypothesis we discussed (no objection to it from Reviewers #1 and #3).
- Reviewer #2's concluding comment that “**In any case, SGBM should be comparable to CIGM**” **contradicts** the established basic facts (detailed in **Table R3, page 9**), where **CIGM** refers to **chemically induced grain-boundary migration**. This is the first mention of **CIGM** by Reviewer #2 but explains the origin of his/her insistence just described.

In the following, we present our point-to-point response to each criticism and further clarify two already clarified issues mentioned by Reviewer #2.

CRITICISM 1: “It is because the concern by reviewer 2, “different solute elements have different GB segregation energy”, is still not addressed in the revised manuscript. Although **the SGB concentration is not directly dependent of GB segregation energy of elements**, the specific GB energy will be strongly influenced by the solute segregation energy, which is alloying element dependent. **GBs containing different solute elements of the same concentration level, will have different GB energies.**”

We respond to this criticism in the following three points.

Point 1: “Although the SGB concentration is not directly dependent of GB segregation energy of elements”

After two rounds of detailed response from us, Reviewer #2 has finally admitted and accepted that “**the SGB concentration is not directly dependent of GB segregation energy of elements**”. Reviewer #2 had relentlessly insisted that the SGB concentration depends on GB segregation energy – the following is duplicated from Reviewer #2's second round of review:

“The authors **should be aware** that different solute elements have different GB segregation energy and therefore different concentrations at GB even though they have the same matrix concentration.”

Note that the term **GB** used by Reviewer #2 refers to SGB, as our entire work is about SGB, which results from the solidification of the last liquid film. Our Table R1 on page 3 details the fundamental differences between **alloy solidification microsegregation** and **solid-state GB segregation**. Reviewer #2 has overlooked the distinct fundamental differences between them, and therefore incorrectly insists that the SGB concentration depends on the GB segregation energy of the solute.

The concentration of a SGB has **nothing to do with** the GB segregation energy of solutes (ΔG_{seg}). As we have already articulated in our previous response, SGB results from the solidification of the last liquid film (~ 1 nm thick) and inherits the solute concentration of this last film [Rappaz et al., Last-stage solidification of alloys: theoretical model of dendrite-arm and grain coalescence. Metall. Mater. Trans. A 34, 467-479 (2003)]. It is fundamentally different from the formation of a GB through a solid-state process.

The above **incorrect** understanding has been **Reviewer #2's central criticism** of our work from the beginning. Even after Reviewer #2 had ultimately admitted that the SGB concentration does not depend on ΔG_{seg} , he/she still **insisted** that we should include ΔG_{seg} in our model. This

insistence is incorrect and contradicts the alloy solidification science established to date.

Table R1 Alloy solidification microsegregation versus solid-state GB segregation

	Alloy solidification microsegregation	Solid-state GB segregation
Process	Rejection of solutes from the solidifying liquid into the remaining liquid ($k < 1$)	Diffusion of solute atoms from the bulk grain to the GB
Cause	Difference in the solubility of solute atoms between liquid and solid	Difference in the chemical potential of solute atoms between bulk grain and GB
Driving force	Reduction in Gibbs free energy per unit volume from the liquid to the solid	Reduction in Gibbs free energy of the system after solute redistribution
Solute type	All ($k < 1$ enriched; $k > 1$ depleted)	Selective
Basic Equation (totally different) [1, 2]	$X_L = X_0 [1 - (1 - \beta k) f_s]^{\frac{k-1}{1-\beta k}} \quad (1)$ $\beta = 2\alpha \left[1 - \exp\left(-\frac{1}{\alpha}\right) \right] - \exp\left(-\frac{1}{2\alpha}\right) \quad (2)$ $\alpha = \frac{4D_s \Delta T}{\lambda^2 \dot{T}} \quad (3)$  X_L: solute composition in the remaining liquid X_0: overall solute concentration of the alloy k: phase diagram solute partition coefficient f_s: solid fraction by solidification D_s: solute diffusion coefficient in the solid λ: secondary dendrite arm spacing ΔT: actual solidification temperature range \dot{T}: cooling rate during solidification 	$\frac{X_I^\theta}{1-X_I^\theta} = \frac{X_I^B}{1-X_I^B} \exp\left(-\frac{\Delta G_{seg}}{RT}\right) \quad (1)$ $\Delta G_{seg} = \Delta G_I^0 + \Delta G_I^E \quad (2)$ (for dilute binary alloys)  X_I^θ: GB solute composition in molar fraction X_I^B: bulk grain solute composition ΔG_{seg}: GB segregation energy ΔG_I^0: segregation energy for the case of an ideal solution ΔG_I^E: excess segregation energy. 
Characteristic parameters	Solute partition coefficient k and solute diffusion coefficient in the solid D_s	GB segregation energy ΔG_{seg} , GB structure and temperature

[1] Clyne, T. W. & Kurz, W. Solute redistribution during solidification with rapid solid-state diffusion. *Metall. Mater. Trans. A*. **12A**, 965-971 (1981).

[2] Lejček, P. *Grain Boundary Segregation in Metals*, 59, 67 (Heidelberg Univ., Heidelberg, 2010).

Point 2: “GBs containing different solute elements of the same concentration level, will have different GB energies.”

Reviewer #2 criticized that our model only predicts the dependence of SGB energy on solute concentration, but not on solute type. This criticism **contradicts** the basic data and facts presented in our work – our model precisely predicts that “GBs containing different solute elements of the same concentration level, will have different GB energies.”

Table R2 was compiled from our last submission (columns 1 and 5 from our Table 2, while columns 2-4 from Supplementary Tables S3 and S4). These results **accentuate** that **SGBs containing different solute elements** (Zn, Al, Sn, Pb in Mg, column 1 in Table R2) **of the same concentration** (0.3at.%) **have different GB energy values of σ_1** (column 6 in Table R2).

Furthermore, Figure 7 of our last submission (reproduced below) highlights the same fact, that is, **both solute type and solute content** affect the SGB energies (σ_1), using **10 different alloy systems**. Our Figure 6 (a, b) in our manuscript **also conveys** the same concept. **This is the basic concept of our model** and has been discussed in detail in our submission. Note that this is the **third review** of the same sets of data by Reviewer #2.

Table R2 Different solute elements of the same concentration (0.3at.%) lead to different SGB energies (σ_1) by our model.

Alloy (at.%)	T (K)	k	β	$(\sigma_0 - \sigma_1)/\sigma_0$ (%)	σ_1 (J/m ²)	$(\sigma_0 - \sigma_1)$ (J/m ²)
Mg-0.30Zn	884	0.06	0.199	20.51	0.4134	0.1066
Mg-0.30Al	921	0.36	0.196	2.26	0.5082	0.0118
Mg-0.30Sn	915	0.31	0.195	2.47	0.5272	0.0128
Mg-0.30Pb	916	0.41	0.195	1.59	0.5117	0.0083

Figure 7 (reproduced from our last submission) **Different solute elements of the same concentration** lead to different SGB energies (σ_1) in different alloy systems.

Point 3: From the beginning, Reviewer #2's suggestion to adopt the concept of GB segregation energy ran counter to the comments of Reviewer #1 and the solidification science.

Reviewer #1 (first review): “(7) For the dilute compositions discussed in this paper, the authors are assuming equilibrium solidification when they take the last solid to form to have composition X_0 and for grain boundaries to form at the solidus temperature. Surely, limited diffusion in the solid at your cooling rates will cause the final composition of the alpha-Mg to be higher and, for most compositions, to be the solid solubility at the eutectic tie line? (i.e. solidification will be closer to Scheil than Lever). This would result in a bigger difference in composition between the GB and the interior. Does assuming **Scheil conditions instead of Lever rule** make a significant difference to your analysis?”

In these comments, Reviewer #1 has clearly suggested that we should also consider using the Scheil model, not just the Lever rule, to determine the SGB composition, which affects the SGB specific energy (σ_1). The important fact is that whether we use the Scheil equation or the Lever rule, the resulting SGB composition is **not related to the solute GB segregation energy ΔG_{seg}** . Following the comments/suggestions of Reviewers #1 and #3, we improved our SGBM model, which shows remarkable consistency with observations in 10 different alloy systems. Review #2's suggestion contradicts the comments of Reviewer #1 and the solidification science represented by either the Lever rule or the Scheil equation.

CRITICISM 2: “However, equation 11 in the revised manuscript just tells us **the driving factor of $(\sigma_0 - \sigma)$ is only dependent of the final concentration of the last liquid film (about 1nm thick) during solidification, independent of solute type**. The only conclusion we can get from their model is that a dilute alloy is easier to get SGBM than a denser alloy for the same alloy system. If for different alloys systems, the alloys with less solute enrichment at the grain boundary (less solidification microsegregation) are easier to get SGBM **while the type of alloying elements has no effect at all**. Frankly, I would not consider this model as predictive.”

This criticism is essentially the same as Criticism 1. Reviewer #2 criticizes us again by emphasizing that the quantity $(\sigma_0 - \sigma)$ predicted by our model (Eq. 11) “**is only dependent of the final concentration of the last liquid film (about 1nm thick) during solidification, independent of solute type**”, where σ is the specific SGB energy (the SGB contains solutes) and σ_0 is specific solvent metal SGB energy (free of solutes).

As highlighted above in the last column of our Table R2, the values of $(\sigma_0 - \sigma_1)$ are **directly dependent on solute type** (Zn, Al, Sn, Pb in Mg) at the same solute concentration (0.3at.%).

For example, changing the solute type from Pb to Zn at the same solute composition of 0.3at.% increases $(\sigma_0 - \sigma_1)$ from 0.0083 J/m² to 0.1066 J/m², **an increase of 1284%! Reviewer #2's criticism that “while the type of alloying elements has no effect at all” again simply contradicts** the basic data or fact presented and discussed in our work.

Furthermore, as shown in our Fig. 7 above, our model highlights the combined effect of **solute type and solute content** on $(\sigma_0 - \sigma_1)$ for all 10 different alloy systems studied, where the influence of **solute type** can far exceed that of solute content as just discussed.

CRITICISM 3: “The major problem is: when deducing Equation 4 and 5, the authors **dropped down the term of $\mu_i(\phi)$** which is relating to the grain boundary concentration of different solute elements. **Therefore, the effect of solute concentration of different elements in GB on the specific GB energy is not properly addressed.”**

This criticism is a different version of Criticisms 1 and 2. As regards the term of **$\mu_i(\phi)$** , we **did not drop down** it. First, let us **duplicate** Lines 240-256 of our 3rd submission as follows:

“... The basic Gibbs equation that connects the specific interfacial energy (σ) between two phases with their interfacial composition, at constant temperature and pressure, is given by^{80,83}

$$\sum_{i=1}^m n_i^\phi d\mu_i + A d\sigma = 0 \quad (4)$$

where n_i^ϕ means the same as above, i.e., the number of moles of component i in the interface (ϕ).

... while for the solute (X_i) we can write $\mu_i \approx \mu_i^0 + RT \ln X_i$, where μ_i^0 refers to the Henrian standard state^{86*}. Eq. (4) can be written as

$$\frac{d\sigma}{dX_i} \approx - \frac{RT}{X_i} \frac{n_i^\phi}{A} \quad (5)$$

which is effectively the same as Eq. (3) of Ref. [88] used by Liu and Kirchheim in their formulation of σ for solute-segregated non-solidification GBs. The quantity n_i^ϕ/A , often denoted as Γ_i ⁸⁶, can be expressed through the GB thickness (Δ), GB atomic density (moles/m³) and GB solute composition X_i^ϕ as follows^{88, 89}

$$\Gamma_i = \frac{n_i^\phi}{A} = \Delta \rho X_i^\phi \quad (6)$$

As highlighted in the line above Eq. (5), the term μ_i is substituted by $\mu_i^0 + RT \ln X_i$, where

μ_i is related to the SGB composition X_i^ϕ through Eqs. (5) and (6), denoted as $\mu_i(\phi)$. We **did not drop down $\mu_i(\phi)$ at all**. Conversely, Reviewer #2 **has overlooked** these details.

The repeated comment of Reviewer #2, "Therefore, the effect of the solute concentration of different elements in the GB on the energy of a specific GB is not properly addressed", is the **same** as Criticisms 1 and 2 just addressed above, namely Reviewer #2 criticizes us again for not considering the effect of solute type on SGB energy σ_1 . We have explicitly shown through our Table R2 and Fig. 7 above that the specific SGB energy σ_1 or the quantity $(\sigma_0 - \sigma_1)$ depends on both **solute type and solute content** and that the influence of solute type (e.g., increased by 1284%) can far exceed the influence of solute content. Furthermore, all calculated values of σ_1 using our model (e.g., see Table R2 above) are in line with the widely accepted GB energy values ($< 1 \text{ J/m}^2$ or around 1 J/m^2 , see our Table S1 and Table S2). All of these results were presented in our manuscript reviewed by Reviewer #2, but they were ignored.

CRITICISM 4: "In the **revised** manuscript, **it is suggested** that SGBM is an athermal process, which is independent of time. Then it means that all SGBM could complete during solidification independent of cooling rate. **But this is contradictory to their experimental results shown in Fig. 5.** A clear migration of GB from the solute rich zone can be observed in the lower part of the GB (to the left of mark 4 in Fig. 5b). This may imply that SGBM is similar to the GB migration during post solidification annealing treatment."

First of all, the hypothesis that SGBM is potentially athermal was made in our **1st submission**, **not** in the 3rd submission. It was reviewed by all **three** reviewers (reviewed twice by Reviewer #2 prior to this review). Let us duplicate lines 355-357 from our **2nd** submission as follows:

"In other words, it tends to fall in the regime of athermal motion of GBs, which can traverse at the speed of sound [89], compatible with the rapid SGBM observed at the cooling rate of $1690 \text{ }^\circ\text{C/s}$ in this work."

Secondly, Reviewer #2, when asked to act on behalf of Reviewer #3, fully **confirmed** our above hypothesis. Please see Reviewer #2's original comments below on the **same** hypothesis:

"About the diffusion mechanism which controls the GB migration, the authors have revised the manuscript and discussed the mechanism in a proper way".

Reviewer #2 seems to have forgotten this previous confirmation (due to the long interval).

Thirdly, this hypothesis was proposed by us as a possible explanation for the profound SGBM

observed at the cooling rate of **1690 °C/s** (**7.5 times** the cooling rate of brine quenching), which is **diffusionless** (it is not the core part of our work). There was no objection from all reviewers after the 2nd review. As quoted above, it was confirmed by Reviewer #2 as well. The language we used was “It tends to occur athermally” (Abstract).

Finally, this hypothesis does not **contradict** but **fully supports** our experimental results in Fig. 5. As discussed in our work, SGBs need to break away from the enriched solutes before athermal SGBM occurs. This breakaway requires a certain driving force. We have repeatedly emphasized that **only** those SGBs that have a **sufficient driving force** for SGBM can migrate (many SGBs do not exhibit migration due to the insufficient driving force for SGBM). Our model has predicted both types of cases, validated with remarkable consistency in 10 different alloy systems containing different types of solutes with different compositions.

Furthermore, we have **systematically** shown that after pronounced migration, SGBs approach their equilibrium state (nearly hexagonal with $d/a \approx \sqrt{3}$, Table 2 in our manuscript), while without migration, those tortuous SGBs mostly remain irregular. During subsequent annealing, these **irregular SGBs** tend to evolve to their near equilibrium state to reduce the total GB energy. **Further migration** is one way to achieve it, as predicted by J. N. DuPont et al [*Welding metallurgy and weldability of nickel-base alloys*, p. 59, Wiley, 2009]. Conversely, those **close-to-equilibrium SGBs** due to **SGBM** will be much more stable during subsequent annealing.

All these expectations are perfectly confirmed in our Fig. 5 (annealed 48 hours at $0.7T_{\text{solidus}}$), **fully supporting our model**. We have now used Arrow 5 to **highlight** the migration mentioned by Reviewer #2 in our Fig. 5 to **further** support our model.

Reviewer #2's two additional comments and concluding comment

Comment 1: “As reviewed in the manuscript, SGBM occurs near the solidus temperature. But the authors **did not clarify** if SGBM is happening before or after the solidification is completed (above or below solidus temperature). If for the former case, the last liquid film in between grains should have played important roles during the SGBM, while for the latter, the SGBM is a pure solid state migration process. **In any case, SGBM should be comparable to CIGM**”.

First of all, a GB **does not exist** in the liquid state. It **must** exist in the solid state. We articulated the definition of GBs in our Introduction and emphasized that each SGB results from the **solidification** (disappearance) of a **last liquid film**. We have made it clear that SGBM occurs after solidification near T_{solidus} . **It is also clear to Reviewers #1 and #3**. Let us duplicate the original descriptions in our 3rd submission just reviewed by Reviewer #2:

Page 20 Line 385:

“Kinetics of SGBM. Immediately after solidification, a SGB is associated with solute ...”

Page 14 Lines 257-259:

“Since the formation of a SGB results from the solidification of a last liquid film ... we have $X_i^\phi = X_{L(f_s \rightarrow 1)}$ immediately after solidification.” (X_i^ϕ is the SGB composition).

The term “immediately after solidification” should be clear enough, which means solid state.

Reviewer #2's concluding comment: “In any case, SGBM should be comparable to CIGM”

The facts listed in Table R3 below highlight that this concluding comment made by Reviewer #2 contradicts the basic facts and is incorrect.

As mentioned in the beginning, CIGM stands for chemically induced grain-boundary migration, usually referred to as diffusion induced grain-boundary migration (DIGM) [International Materials Reviews, 1995:40;149-179]. This is the first mention of CIGM by Reviewer #2. Table R3 summarises the fundamental differences between SGBM and CIGM. Reviewer #2's concluding comment contradicts the basic facts and is incorrect.

Table R3 Distinct fundamental differences between SGBM and CIGM or DIGM

	SGBM	CIGM or DIGM
Temperature	Near solidus temperature or melting point (T_m) [1-3]	0.3-0.5 T_m (in Kelvin) [4-7]
Kinetics	Diffusionless – occurs at the cooling rate of 1690°C/s (this work)	Diffusion controlled [4-12] time dependent
GB curvature	Decrease [1-3, 13, this work]	Increase [4, 7, 8,10]
Driving force	Reduction in the total SGB energy [1-3, 13, this work]; not due to grain growth [13, this work])	Diffusional coherency strain energy [4-12] but other driving forces might also operate [4, 5, 9, 11, 12]

(The references listed below include seminal contributions of Cahn, Hillert and Purdy mentioned by Reviewer #2)

1. Pigenko, A. A., Chernyshova, T. A., Shorshorov, M. K. & Dontsova, A. Y. Structure formation and in cast and deposited maraging steel N18K8M3TYu. *Met. Sci. Heat Treat.* **17**, 751-754 (1975).
2. Matsuda, F., Nakagawa, H., Ogata, S. & Katayama, S. Fractographic investigation on solidification crack in the vareststraint test of fully austenitic stainless steel: studies on fractography of welded zone (III). *Trans. of JWRI* **7**, 59-70 (1978).
3. Matsuda, F., Nakagawa, H., Uehara, T., Katayama, S. & Arata, Y. A new explanation for role of

- delta-ferrite improving weld solidification crack susceptibility in austenitic stainless steel. *Trans. of JWRI* **8**, 105-112 (1979).
4. Beke, D. L., Kaganovskii, Y. & Katona, G. L. Interdiffusion along grain boundaries–Diffusion induced grain boundary migration, low temperature homogenization and reactions in nanostructured thin films. *Prog. Mater. Sci.*, **98**, 625-674 (2018).
 5. King, A. H. Diffusion induced grain boundary migration. *Int. Mater. Rev.*, **32**, (173-189) 1987.
 6. Balluffi, R. W. & Cahn, J. W. Mechanism for diffusion induced grain boundary migration. *Acta Metall.* **29**, 493-500 (1981).
 7. Cahn, J. W., Pan, J. D. & Balluffi, R. W. Diffusion induced grain boundary migration. *Scripta Metall.* **13**, 503-509 (1979).
 8. Yoon, D. Y. Theories and observations of chemically induced interface migration. *Int. Mater. Rev.* **40**, 149-179 (1995).
 9. Hillert, M. Solute drag, solute trapping and diffusional dissipation of Gibbs energy. *Acta Mater.* **47**, 4481-4505, 1999.
 10. Ma, C. Y., Rabkin, E., Gust, W. & Hsu, S, E. On the kinetic behavior and driving force of diffusion induced grain boundary migration. *Acta Metall. Mater.* **43**, 3113-3124 (1995).
 11. Hillert, M. & Purdy, C. R. Chemically induced grain boundary migration. *Acta Metall.* **26**, 333-340 (1978).
 12. Broeder, F. J. A. D. Interface reaction and a special form of grain boundary diffusion in the Cr-W system. *Acta Metall.* **20**, 319-332 (1972).
 13. Shibata, S & Watanabe, T. Grain boundary behavior in the weld metal of austenitic stainless steel - II. *Koon Gakkai-Shi (Journal of High Temperature Society) (Japan)*, **20**, 69-78 (1994).

Comment 2: “However, in this manuscript the authors just simply attributed SGBM to the capillarity force as the driving force without considering other possible effects like **elastic strain energy**”.

We appreciate this comment, but we have already **clarified** the issue of elastic strain energy in our 1st to 3rd submissions, based on experiments specifically designed by two influential researchers in the field (Ref. [22] in our manuscript). Our clarification was highlighted in the Introduction of all three submissions as **point (ii)** on page 4:

Page 4 Lines 70-72 (3rd submission): “(ii) the **strain** or stress generated during cooling after solidification has minor or negligible influence based on systematic assessments²², **which clarifies an important concern about the underlying mechanism;**”

Furthermore, the elastic strain energy (G_{elastic}) as a source of the driving force for GBM is known to be usually much smaller than the GB energy (G_{GB}) for micron-sized and nano-sized grains [1-3], that is, G_{elastic} is usually negligible when G_{GB} operates. This is in line with Ref. [22] above. To further clarify this issue, let us consider an alloy, which has an average grain size of 100 μm , specific SGB energy of 0.5 J/m^2 near T_{solidus} (usually 0.52 – 0.90 J/m^2 at T_{solidus} ,

Table S2), and elastic modulus of 100 GPa at T_{solidus} (high for a normal alloy). We then apply a stress of 10 MPa to the alloy at T_{solidus} . Note that the strength of most alloys approaches zero at T_{solidus} (< 1 MPa) [4-6]. The applied stress of 10 MPa at T_{solidus} is thus substantial. Using these parameters, Gottstein and Shvindlerman compared G_{elastic} and G_{GB} as driving forces for GBM in their seminal monograph on GBM [1]. G_{elastic} is two orders of magnitude smaller than G_{GB} , i.e., it is negligible.

The formulation of our driving force model is thus appropriate. It is also consistent with previous studies on the driving force for SGBM (Refs. 20, 24, 30, 44 in our manuscript). Therefore, we **did not** “just simply attribute SGBM to the capillarity force as the driving force”. Our model is supported by its precise application to alloy design for SGBM and by its remarkable agreement with observations of SGBM in **10** representative alloy systems. This is **not** a coincidence.

For greater clarity, we have added the above discussion as Section 4 Elastic strain energy as a source of the driving force for SGBM near the T_{solidus} to our Supplementary Information and the following sentence to Page 12 of our revised manuscript:

The elastic strain energy as a source of the DF for SGBM is negligible near the T_{solidus} (see Supplementary Section 4).

1. Gottstein, G. & Shvindlerman, L. S. *Grain boundary migration in metals: thermodynamics, kinetics, applications. 2nd edn*, (ed. Ralph, B.), p144 (CRC Press, Taylor & Francis Group, Boca Raton, 2010).
2. Saxena, R., Cho, W., Rodriguez, O., Gill, W. N. & Plawsky, J. L. Stability of thin copper films on mesoporous dielectrics. *J. Non-cryst Solids* **350**, 14-22 (2004)
3. Gianola, D. S., Petegem S. V., Legros, M., Brandstetter, S., Swygenhoven, H. V. & Hemker, K. J. Stress-assisted discontinuous grain growth and its effect on the deformation behavior of nanocrystalline aluminum thin films. *Acta Mater.* **54**, 2253–2263 (2006).
4. Eskin, D. G., Suyitno & Katgerman, L. Mechanical properties in the semi-solid state and hot tearing of aluminium alloys. *Prog. Mater. Sci.*, **49**, 629–711 (2004)
5. Summers, P. T., Chen, Y., Rippe, C. M., Allen, B., Mouritz, A. P., Case, S. W. & Lattimer, B. Y. Overview of aluminum alloy mechanical properties during and after fires. *Fire Sci. Rev.* 4:3 (2015).
6. Xi, X., Li, S., Yang, S., Li, J. & Zhao, M. Effect of adding yttrium on precipitation behaviors of inclusions in E690 ultra high strength offshore platform steel. *High Temp. Mat. PR-ISR* **39**, 510-519 (2020).

Summary

In his/her 3rd and last review, Reviewer #2 made **four** major criticisms and **one** concluding comment (“In any case, SGBM should be comparable to CIGM”). The first three criticisms all contradict the basic facts or data presented and discussed in our work, while the fourth criticism contradicts Reviewer #2's own previous full confirmation of the same hypothesis (athermal) we proposed. It is not the core part of our work and just offers a possible explanation of the diffusionless nature of the SGBM experimentally observed. We are happy to revise it if required (Reviewers #1 and 3 have both reviewed it and had no objection).

From the beginning, Reviewer #2's **central criticism** of our work was our resistance to relating the SGB composition to GB segregation energy of solutes (ΔG_{seg}). After our two rounds of response, Reviewer #2 ultimately admitted and accepted that his/her insistence that the SGB composition depends on the GB segregation energy of solutes is incorrect or at least redundant. This is because a SGB results from the solidification of a last liquid film (~ 1 nm thick) and inherits the last liquid film composition. It is fundamentally different from the formation of a GB through a solid-state process. The SGB composition thus has nothing to do with ΔG_{seg} .

Reviewer #2's concluding comment that “In any case, SGBM should be comparable to CIGM” contradicts the basic facts as highlighted in Table R3 and is incorrect. SGBM and CIGM should be treated differently.

Based on the clear evidence presented above, we have properly addressed all the issues raised by Reviewer #2 in his/her second review. Reviewer #1 offered a thorough review of this work with 17 insightful questions. From the questions he/she asked, Reviewer #1 is likely to be a top expert in solidification science. Also considering the appreciation of our work by Reviewer #3, we propose that our further revised manuscript deserves to be re-considered by NC. It is a milestone development in this topic since 1961, validated with 10 different alloy systems covering a wide range of solidification cases.

We have made some changes to the manuscript to further clarify all the issues that Reviewer #2 commented on in his/her third review. All these changes are highlighted **in red** in the further revised manuscript and listed in order in the accompanying file entitled "**Further revisions to the manuscript**".

Thank you so much for taking the time to review or comment on this work and to evaluate our arguments.

REVIEWERS' COMMENTS

Reviewer #4 (Remarks to the Author):

With respect to the disagreement between Reviewer #2 and the authors, I found that the authors' arguments must be valid. The reasons are as follows.

In the case of non-solidification grain boundary (GB), the concentration at GB (X in Eq. (6)) is generally determined by the GB segregation energy, and therefore the resulting GB energy explicitly depends on the segregation energy. However, the concentration at solidification GB (SGB), which is the main concern of this paper, is mainly determined by the microsegregation during solidification and, as a result, it is not related to the GB segregation energy. As the reviewer #2 points out, then, the SGB energy is determined only by the concentration of the last-solidified liquid film and is not explicitly dependent on solute type in the authors' model. This is a natural consequence of using the Gibbs adsorption theorem based on Henry's law. The fact that their experimental results can be explained by this model is, in my opinion, an important finding that indicates the validity of these laws for the SGBM problem.

On the other hand, as the reviewer #2 pointed out, the possibility that the chemically induced grain-boundary migration (CIGM) is involved in this process cannot be completely ruled out. Having said that, I conclude that the discussion about the possibility of SGBM being an athermal process is not a reason to reject this paper.

So, I found that this manuscript is publishable at a technical level.